# Chalcogen-bridged coordination polymer for the photocatalytic activation of aryl halides

Le Zeng [1,3], Tiexin Zhang [1,3] ✉, Renhai Liu[1,3], Wenming Tian [2], Kaifeng Wu [2], Jingyi Zhu [2], Zhonghe Wang[1], Cheng He[1], Jing Feng[1], Xiangyang Guo [2], Abdoulkader Ibro Douka [1] & Chunying Duan [1] ✉

The ability to deliver electrons is vital for dye-based photocatalysts. Conventionally, the aromatic stacking-based charge-transfer complex increases photogenerated electron accessibility but decreases the energy of excited-state dyes. To circumvent this dilemma, here we show a strategy by tuning the stacking mode of dyes. By decorating naphthalene diimide with S-bearing branches, the S···S contact-linked naphthalene diimide string is created in coordination polymer, thereby enhancing electron mobility while simultaneously preserving competent excited-state reducing power. This benefit, along with in situ assembly between naphthalene diimide strings and exogenous reagent/reactant, improves the accessibility of short-lived excited states during consecutive photon excitation, resulting in greater efficiency in photoinduced electron-transfer activation of inert bonds in comparison to other coordination polymers with different dye-stacking modes. This heterogeneous approach is successfully applied in the photoreduction of inert aryl halides and the successive formation of $C_{Ar}$–C/S/P/B bonds with potential pharmaceutical applications.

Photoredox catalysis has led to a paradigm shift in organic synthesis that incorporates substrate activation and provides access to heretofore elusive reaction pathways[1–4]. Photocatalysts undergo single electron transfer from their excited states to generate open-shell intermediates that participate in distinct single-electron activation modes that are complementary to transition-metal thermo-catalysed reactions[5–7]. Recently, consecutive photon excitation of photoresponsive dyes has been used to purchase the more elevated photoredox potentials[8–11], allowing for the thermodynamically demanding activation of inert bonds[12–15]. Nonetheless, the intrinsic instability and even shorter lifetime of the corresponding excited states hinder the efficiency of photoinduced electron transfer (PET) working with the excited-state dye and inert substrate under the typical paradigm of diffusion-limited electron transfer in homogeneous solution

(Supplementary Fig. 1a)[1–4]. In addition to the competent redox power, the efficient electron transfer between photocatalyst and substrate under photoirradiation, which can be influenced by the lifetime of the excited state, the electron delivery pathway, or the interaction model between electron donor and electron acceptor, is vital for the open-shell activation mode, besides the competent redox power[16–18]. Thus, enhancing the efficiency of PET by simultaneously attaining the competent electron delivery ability and adequate thermodynamic driving force of the excited state is highly desirable for photocatalytic conversion[16,17].

Thinking outside the box of a single dye molecule, the manner of dye-stacking might enhance the electron-delivering ability without relying on sophisticated decoration and tedious synthetic protocols of dye motifs[19,20]. Charge-transfer interactions between photosensitive

[1]State Key Laboratory of Fine Chemicals, School of Chemical Engineering, School of Chemistry, Dalian University of Technology, Dalian 116024, China. [2]Dalian Institute of Chemical Physics, Chinese Academy of Sciences, Dalian 116023, China. [3]These authors contributed equally: Le Zeng, Tiexin Zhang, Renhai Liu. ✉e-mail: zhangtiexin@dlut.edu.cn; cyduan@dlut.edu.cn

electron-deficient (A) and electron-rich (D) counterparts possess interchromophoric electronic coupling, which partially shares electrons in the ground state and reaches complete charge separation after being excited to achieve the high charge mobilities, and is also capable of directing the formation of various supramolecular D···A packing modes[19–21]. In particular, the columnar stacking of D and A not only facilitates the proximal excitation to improve PET efficacy between D···A pairs, but also enhances the long-range charge transfer through the string direction to form the long-lived photoinduced charge-separated state and alleviate the back-electron transfer[22–24], providing the additional possibility of exciton transport within the supramolecular wire-like assembly[25,26]. Thus, it was believed that the columnar arrangement of dye units coupled with charge-transfer interactions could enhance the electron delivery capability of short-lived excited-state dye species from an aggregation-state perspective.

Utilising porous coordination polymer to tune the long-range ordered dye stacking modes and arrange the non-covalent interaction sites at high density around the pores[27–29], we anticipated that connecting the dye motifs by specific charge-transfer interactions and forging them into the supramolecular wire-like[30–32] infinite columnar stacking within the coordination polymer would facilitate the spatial separation and transport of photogenerated charges. This heterogeneous strategy had the potential to overcome the limitations of single-molecule photocatalysis by simultaneously achieving competent electron delivery ability and excited-state thermodynamic driving force. However, spontaneous aromatic stacking typically dominates the assembling modes among aryl moieties of dyes, degrading the precious excited-state energy through the mutual interference of π-orbitals[22,33,34]. Few dye-incorporated coordination polymers possess non-aromatic stacking-bridged dye moiety connections[35–37]. In light of the role of chalcogen-bridged interactions in supramolecular conductors and other materials[38,39], we hypothesised that introducing chalcogen-linked interactions into coordination polymers might improve electronic communication between neighbouring dye moieties in the infinite columnar assembly without relying on π···π interactions, fulfilling the competent electron delivery ability along the dye strings and maintaining the intrinsic photoelectronic property of a single dye unit.

Recently, electron-deficient dyes, such as the aromatic diimide dyes[8,17,40] and analogues[10,11,41], have been subjected to consecutive photon excitation in order to generate radical anionic dye intermediates and utilise their super-strong excited-state reducing potentials in inert bond cleavages under visible light irradiation[8–11]. Among these, naphthalene diimide (NDI) is a proto-typed aromatic diimide dye commonly used in π-stacking molecular materials[42,43], and it has been reported that the excited state of the radical anion $(NDI^{\cdot-})^*$ possesses a high reduction potential of −2.1 V (vs. SCE). Yet, the relatively short lifetime of $(NDI^{\cdot-})^*$ in the range of 103−260 ps[44,45] and the dissipative excited-state dissipation during diffusion in solution usually ends up in a sluggish efficiency of intermolecular PET, thus precluding the practical application of NDIs for multiphoton photocatalysis in solution phase. Herein, we adopted NDI as a model fragment to decorate S-bearing branches (Fig. 1a, b) and obtained S···S-bridged non-aromatic NDI stacking in coordination polymer. The coordination-oriented structural coercions[28,36] were expected to overcome the strong tendency of inter-dye aromatic stacking to separate the cores of NDI, allowing the S-mediated non-covalent interactions to link the neighbouring NDI units (Fig. 1b, c). It was hypothesised that the inter-ligand S···S contacts bridged the charge transfer throughout the non-aromatic stacked NDI string with long-range order (Fig. 1d–f), without compromising the excited-state redox potential of a single NDI/NDI⁻ unit. The electron-donating S-branches may have also served as sites for in situ assembly with electron-deficient inert substrates[46] (Fig. 1f), which improved the

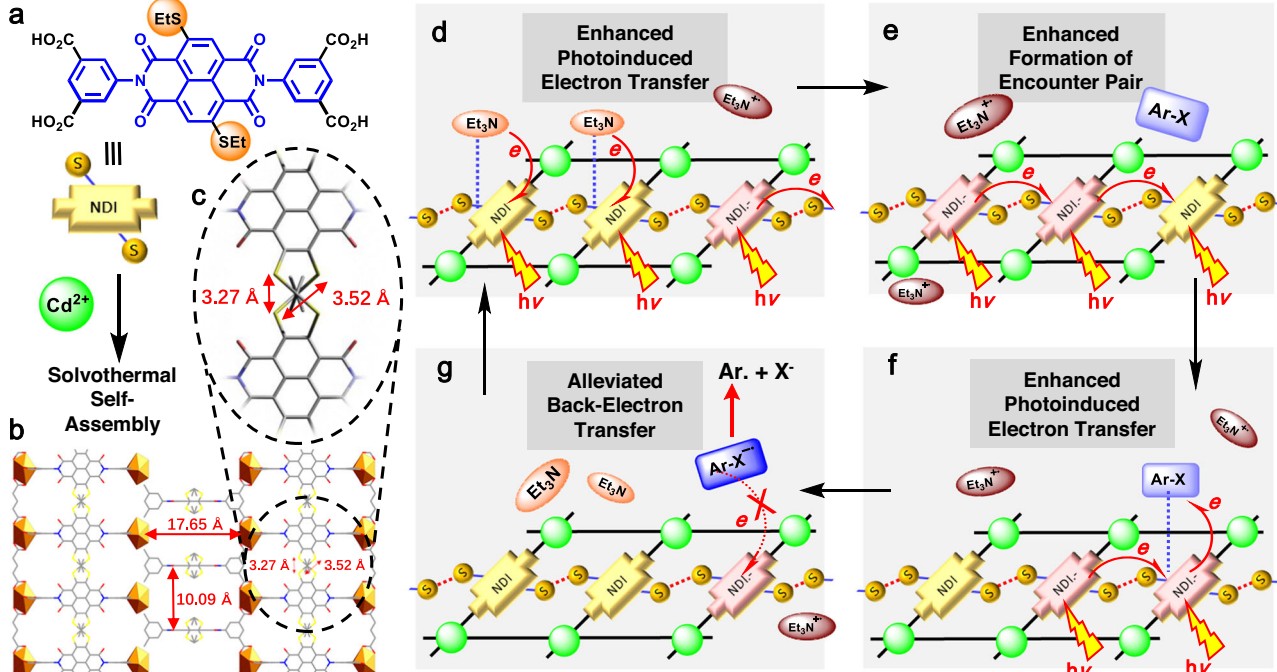

**Fig. 1 | Schematic illustrations of chalcogen-bridged non-aromatic stacking of NDIs to enhance the reactivity of short-lived (dye⁻)*. a, b** Structure of the ligand H₄SNDI and crystal Cd−SNDI showing the non-aromatic assembly of NDI units. **c** An enlarged view of the crystal structure displaying S···S contact between adjacent NDI units. Distances between atoms were shown in angstroms (Å). The disordered SEt groups were drawn from all possible positions. **d–g** Schematic illustrations of one catalytic cycle in Cd−SNDI. The neutral NDI unit and radical anionic NDI⁻ were shown by yellow and wine-red blocks, respectively. The neutral Et₃N and the radical cationic Et₃N⁺ were represented in orange and brown ellipses, respectively. The neutral substrate Ar−X and the one-electron reduced [Ar−X]⁻ were exhibited in light blue and navy blue rectangles, respectively. The "injection of light" symbol, the "hν" text mark, and the "electron transfer" curve arrow were employed at the same time to depict the excitation events of NDI or NDI⁻ and the concomitant electron transfer behaviours.

accessibility of proximally generated excited states toward substrate molecules (Fig. 1e, f).

The activation of inert bonds such as aryl halides through the single-electron transfer (SET) pathway was a cornerstone of synthetic chemistry[41,46,47], and the reductive cleavage of inert aryl halide bonds was widely used as the probe reaction by the photocatalytic system involving consecutive photon excitation[10,47], which was also employed as the proof-of-concept with added value for this study. In the comparative study, a series of NDI-based coordination polymers with alternative stacking modes of NDI moieties were also synthesised; these coordination polymers showed different tendencies for the in-situ association with exogenous reagent/reactant prior to each step of consecutive photoirradiation, as well as distinct accessibilities of short-lived excited states. The chalcogen-bridged coordinated polymer Cd−SNDI was effectively used in the photocleavage of inert aryl halide bonds and the successive formation of $C_{Ar}$−C/S/P/B bonds to prepare molecules with pharmaceutical interests. This supramolecular wire-inspired heterogeneous strategy provided programme-controlled electron transfer through the spatiotemporal order of in situ assembly between NDI-based strings and the guest molecules, which might bring distinctive perspectives to sustainable photocatalysis[48], photo-electronic devices[49], and solar energy conversions[9].

## Results

### Synthesis and characterisation of Cd−SNDI

By replacing the dibromo-substituents of NDI with SEt groups, the chalcogen-branched NDI ligand $H_4$SNDI was prepared from classical procedures[50,51]. UV−vis spectrum of $H_4$SNDI solution in DMF exhibited a broad band in the range of 400−600 nm, which can be assigned to the intra-ligand charge-transfer (CT) band between the electron-donating SEt moiety and the electron-deficient NDI core[52].

The coordination polymer Cd−SNDI was prepared by the solvothermal reaction of a thioethyl-branched NDI-based ligand $H_4$SNDI and cadmium salt in a solvent mixture of DMF and $H_2O$ (Fig. 1a, b). By elemental analysis and powder X-ray diffraction (PXRD)

(Supplementary Fig. 2), the bulk phase purity of Cd−SNDI crystals was confirmed. Single-crystal X-ray structural analysis revealed that Cd−SNDI crystallised in space group P-42c. Each Cd(II) ion was coordinated by four bidentate carboxylates from four ligands, and each ligand connected four cadmium ions to consolidate the three-dimensional network (Supplementary Table 2 and Supplementary Figs. 3–4). The separation between the parallel NDI moieties (ca. 10.09 Å) (Fig. 1b) was well beyond the threshold of aromatic π-stacking to avoid mutual interference between the π-orbitals. The calculated S···S distance between S-substituents of neighbouring ligands was 3.40 Å, which was the mean of two conceivable configurations of disordered SEt groups with distances of 3.27 Å and 3.52 Å, respectively. This proximity indicated a remarkable inter-ligand S···S contact (Fig. 1c) to bridge the infinite strings of S−NDI−S···(S−NDI−S)$_n$···S−NDI−S[38,39], which might facilitate the charge transfer throughout Cd−SNDI (Fig. 1d−f).

### Photoelectronic property of Cd−SNDI and $H_4$SNDI

Compared to the free ligand $H_4$SNDI, the suspension of Cd−SNDI in DMF showed a more notable and wider charge-transfer band spanning the visible light region of 400−700 nm (Fig. 2a), indicating the presence of abundant charge-transfer interactions within the ground-state framework. The remarkably lower fluorescence intensity (Fig. 2c) and much shorter lifetime of photoluminescence emission (Fig. 2d) of Cd−SNDI compared to those of free ligand $H_4$SNDI suggested a more efficient inter-ligand emission quenching and a potential photo-induced charge separation along the framework[53]. For reducing NDI to NDI·⁻, the solid-state electrochemistry of Cd−SNDI exhibited a redox peak at −0.57 V (vs. SCE, Fig. 2b), which was similar to that of the free ligand $H_4$SNDI (−0.54 V vs. SCE). And the oxidative potential of excited-state Cd−SNDI was determined to be 1.64 V based on a free energy change of 2.21 eV, which was comparable to the case of ($H_4$SNDI)* (1.60 V calculated from $E^0$ = 2.14 eV, Fig. 2a). Theoretically, this oxidising power was sufficient for the excited-state (NDI)* core to extract electrons from an electron-donating reagent such as $Et_3$N (0.87 V vs.

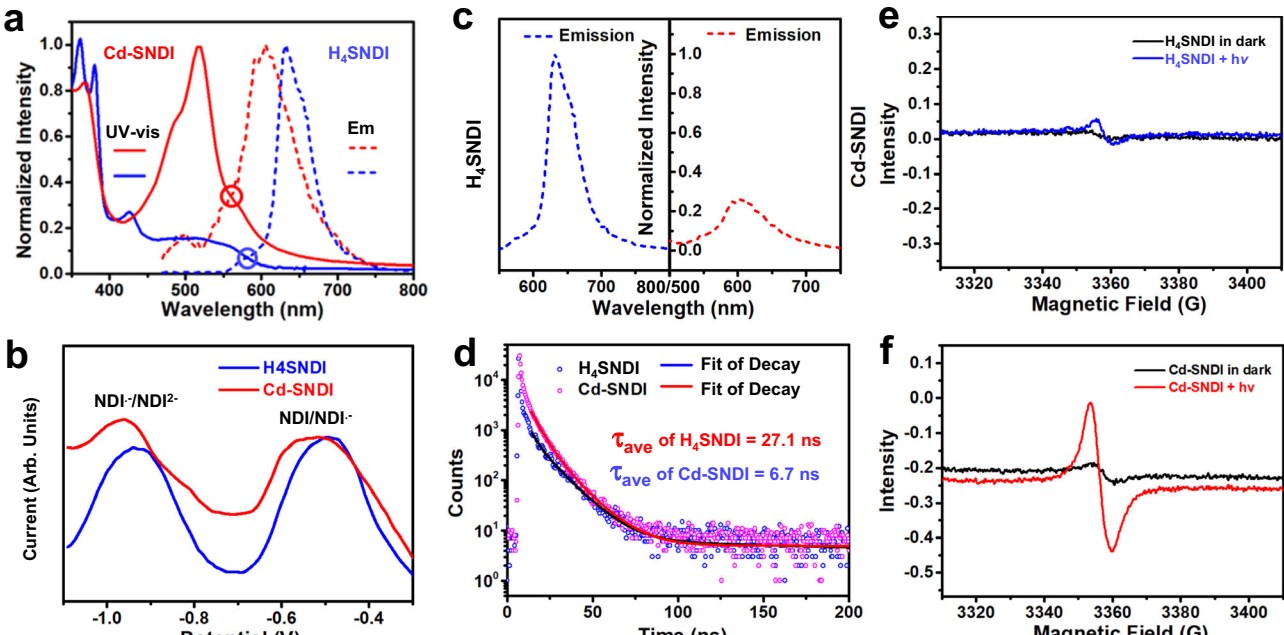

**Fig. 2 | Comparison of the photo- and electronic properties of Cd−SNDI and $H_4$SNDI to reveal the charge transfer along the chalcogen-linked NDI string.** **a** Normalised absorption and emission spectra of the suspension of Cd−SNDI and the solution of $H_4$SNDI. **b** Differential pulse voltammetry (DPV) curves of Cd−SNDI and $H_4$SNDI. **c** Fluorescence emission spectra and **d** lifetime decay for the suspension of Cd−SNDI and the solution of $H_4$SNDI. The fluorescence intensities in (**c**) are normalised by the maximal emission intensity of $H_4$SNDI, which was set to 1.0. Electron paramagnetic resonance (EPR) spectra of solid samples of (**e**), $H_4$SNDI and (**f**), Cd−SNDI in the absence or presence of irradiation.

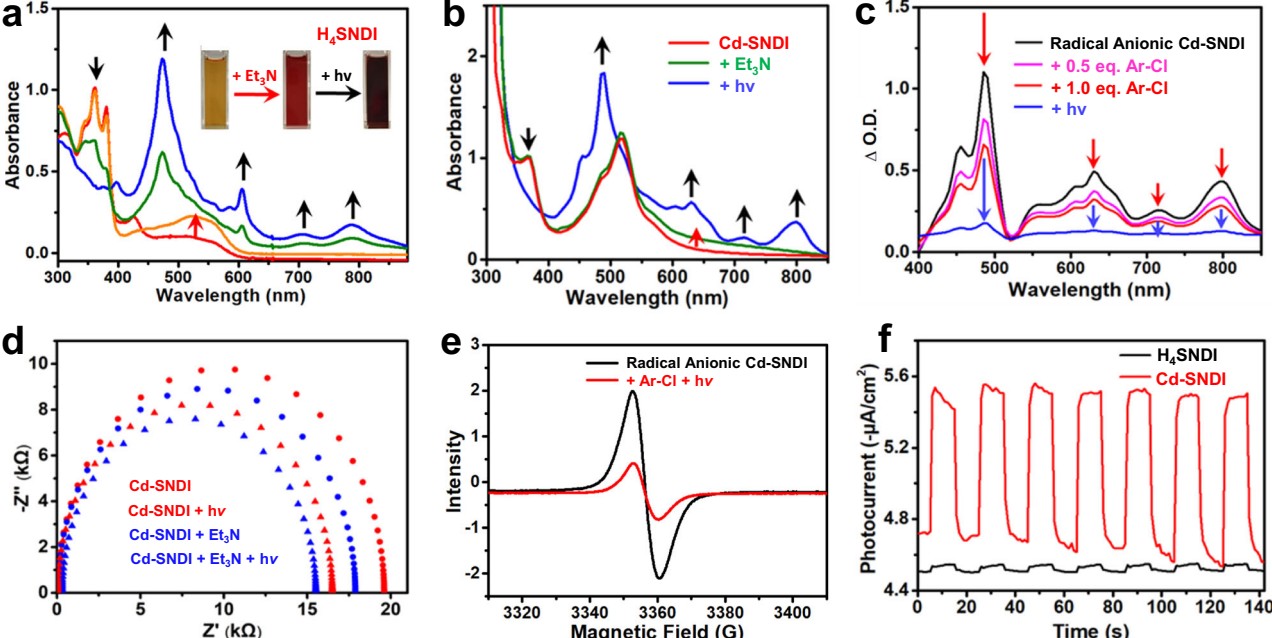

**Fig. 3 | The generation of radical anion of Cd−SNDI/H₄SNDI and their interactions with guest molecules.** UV−visible absorption spectra of (**a**) H₄SNDI solution or (**b**) Cd−SNDI suspension after Et₃N addition (red arrow) and subsequent irradiation (black arrow). The inset pictures in (**a**) show naked eye-detectable colour changes. **c** The absorbance difference spectra of radical anionic Cd−SNDI suspension upon the addition of Ar−Cl (red arrow) and subsequent irradiation (blue arrow). The spectrum for subtraction is the mixture of neutral Cd−SNDI and Et₃N without irradiation. **d** Electrochemical impedance spectra (EIS) of Cd−SNDI (red) or Cd−SNDI with Et₃N addition (blue), respectively, before (circles) and after (triangles) 455 nm LED irradiation. **e** EPR spectra of solid radical anion Cd−SNDI and its response to Ar−Cl addition and light irradiation. **f** Transient photocurrent responses of Cd−SNDI (red) and H₄SNDI (black) on the cathode using 4'-chloroacetophenone as an electron sink.

SCE) in order to achieve photoinduced charge separations. Evidently, the S···S bridged stacking manner in coordination polymer Cd−SNDI had no impact on the ground- and excited-state redox abilities of NDI moieties. The electron paramagnetic resonance (EPR) analysis of a solid sample of free ligand revealed a negligible NDI[•−] signal in the absence of LED irradiation (Fig. 2e), indicating a slight charge separation under the weak photoirradiation from ambient daylight. In comparison, the crystals of coordination polymer Cd−SNDI exhibited a more pronounced NDI[•−] signal in the EPR spectrum (Fig. 2f). Under the photoirradiation of a 455 nm LED, the NDI[•−] signal of Cd−SNDI intensified significantly more than that of H₄SNDI (Fig. 2e, f). These phenomena possibly implied the photoinduced inter-ligand charge separation in Cd−SNDI. As shown in Fig. 3d, electrochemical impedance spectroscopy (EIS) revealed a decrease in the electrical impedance of Cd−SNDI in response to light irradiation, demonstrating the presence of photoseparated charge pairs[54]. On the basis of these findings, it was believed that the close S···S contact of Cd−SNDI might provide non-aromatic *p*-orbital coupling, thus facilitating the photogeneration of charge carriers and their inter-ligand transfer[38,39] (Fig. 2f), which was desirable for the photocatalytic application.

## Consecutive photon excitation of Cd−SNDI and H₄SNDI
With the addition of the electron donor Et₃N to the DMF solution of H₄SNDI, the intensification of the broad intra-ligand charge-transfer (CT) band centred at *ca.* 520 nm was detected, indicating the S-mediated partial charge transfer from Et₃N to the NDI moiety (Fig. 3a). When Cd−SNDI was treated with Et₃N, a broadened absorption band centred at *ca.* 650 nm was detected (Fig. 3b), implying a possible charge-transfer interaction between the electron-donating guest molecule and the electron-accepting host framework[39]. Upon irradiating the suspension of Cd−SNDI with Et₃N using a 455 nm LED, the typical doublet-state transition peaks of NDI[•−] were observed near 485, 630, 715, and 800 nm, respectively, confirming the accumulation

of net negative charges (Fig. 3b)[9,55]. The overlap between the broad charge-transfer band of neutral Cd−SNDI and the characteristic peaks of its radical anionic form favoured the consecutive utilisation of monochromatic LED light. Similar treatment of the solution of ligand H₄SNDI resulted in the appearance of UV−visible absorption bands at the aforementioned four radical anionic sites, accompanied by a colour change visible to the naked eye (Fig. 3a). Importantly, the $D_0 \rightarrow D_1$ transition wavelength of H₄SNDI was located at 788 nm. In this case, a 12 nm red-shift for the $D_0 \rightarrow D_1$ transition of Cd−SNDI than H₄SNDI was detected (Fig. 3a, b), suggesting the possible effect of inter-ligand charge sharing in Cd−SNDI[44,56]. The EPR spectra of both Cd−SNDI and free ligand H₄SNDI exhibited the signals of NDI[•−] after successive treatment with Et₃N and visible light (455 nm) (Supplementary Fig. 8c, d), demonstrating that the first stage of consecutive PET was accomplished. The EIS investigation revealed that the addition of Et₃N decreased the electrical resistance of Cd−SNDI, and that subsequent photoirradiation further reduced the impedance of the material (Fig. 3d), indicating the improved electronic conductance of NDI strings in the presence of additional charges.

After the above-mentioned addition of Et₃N and successive 455 nm LED photoirradiation, the C_Ar−Cl model substrate 4'-chloroacetophenone (**1a**) was added into this in situ generated radical anionic Cd−SNDI suspension, and the four fingerprint peaks of NDI[•−] were found to attenuate (Fig. 3c), reflecting the possible interactions and weak associations between the electron-enriched radical anionic NDI[•−] moieties and the electron-deficient substrate **1a**. Similar treatment of radical anionic ligand by substrate **1a** also revealed the evidence for Ar−Cl···H₄SNDI[•−] interaction (Supplementary Fig. 18). On the contrary, if the model substrate **1a** was added directly into a DMF suspension of neutral Cd−SNDI, only a tiny attenuation of the broad band centred at *ca.* 520 nm was observed (Supplementary Fig. 17b). These results ruled out noticeable interaction or pre-association between neutral Cd−SNDI and Ar−Cl substrate **1a**. As depicted in

Fig. 1d–g, interactions and weak associations between exogenous reagents/substrates and the NDI strings of Cd−SNDI might contribute to the in situ assembly between the host framework and the guest molecule during each step of consecutive excitation. Furthermore, considering the high charge transfer efficiency of the S-bridged supramolecular systems[38,39] and Cd−SNDI with close S···S contacts, the collision and association of Et$_3$N (or Ar−Cl) with an arbitrary site of neutral (or negatively charged) NDI string could form the encounter pair for the PET process, and the separated charges would be carried away for the next rounds of encountering and PET events.

Owing to the competitive nonradiative decay of (NDI$^{•-}$)$^*$, it was impractical to access the free energy change ($E^0$) between the ground state and the vibrationally related excited state utilising fluorescence technology. Alternatively, based on the potential of NDI/NDI$^{•-}$ ($E_{1/2}$) and the doublet transition energy ($D_0 \rightarrow D_1$) of NDI$^{•-}$, the excited-state reducing potentials of radical anionic Cd−SNDI and radical anionic H$_4$SNDI were estimated according to the literature method[57,58] as the similar values of ca. −2.12 V and ca. −2.21 V, respectively, which possessed sufficient driving forces for electron transfer to the substrate, allowing the cleavage of inert C$_{Ar}$−Halide bonds (e.g., -1.90 V for C$_{Ar}$−Cl of 4'-chloroacetophenone)[59]. These results indicated that the S···S-bridged assembly mode of NDI motifs in Cd−SNDI did not degrade the thermodynamic properties of (NDI$^{•-}$)$^*$. After shining light of 455 nm LED to the pre-mixed equivalent amount of radical anionic Cd−SNDI and **1a**, the typical NDI$^{•-}$ absorption peaks eventually diminished (Fig. 3c), validating the accessibility of proximal excited-state radical anion towards substrate in PET process. Furthermore, when employing an excess amount of 4'-chloroacetophenone as the electron sink under a N$_2$ atmosphere, the transient photocurrent of Cd−SNDI at the cathode under the photoirradiation of 455 nm LED was much more remarkable than in the case of using ligand H$_4$SNDI (Fig. 3f), demonstrating the superior PET efficiency from (NDI$^{•-}$)$^*$ of Cd−SNDI to electron sink **1a**[60].

## Comparative photocatalytic analysis of Cd−SNDI and H$_4$SNDI

Photoreductive cleavage of aryl halides began with 4'-chloroacetophenone. A 10 mol% Cd−SNDI loading enabled a 74% yield in the presence of Et$_3$N after 4 h light irradiation of 455 nm LED (Table 1, entry 1). Among the various electron donors screened, dibutylamine outperformed the other candidates and gave rise to a yield of 90% for C$_{Ar}$−Cl cleavage (Table 1, entry 6). Furthermore, a series of control experiments were performed to investigate the structure-activity relationship of this photocatalytic system. No reactions were detected without photocatalyst, electron donor, or photoirradiation (entries 8, 12, and 13). The use of cadmium salt, the ligand H$_4$SNDI or a mixture of cadmium salt with the ligand resulted in significantly inferior activities (entries 9, 10, and 11), proving the necessity of the integrated structure of Cd−SNDI for the reaction. The photocleavage by Cd−SNDI did not occur when exposed to an aerobic atmosphere (entry 14). With the addition of the radical scavenger 2,2,6,6-tetramethyl-1-piperidinyloxy (TEMPO), the reaction could be blocked with a conversion of less than 10%, indicating a radical process (entry 15)[8]. The in situ generated aryl radical was trapped by TEMPO, as confirmed by high-resolution mass spectrometry (HRMS) (Fig. 4f)[8]. Moreover, the conversion of **1a** terminated immediately after the hot filtration of catalyst particles, reflecting the heterogeneous nature of this reaction (entry 16). After photocatalysis, the coordination polymer was easily isolated from the reaction mixture by centrifugation and could be reused at least three times without significant loss of reactivity (Fig. 4a). The PXRD pattern of the recovered catalyst indicated its structural integrity (Fig. 4b).

When 4'-bromoacetophenone (**1b**) and 4'-iodoacetophenone (**1c**) were used, the photocatalytic system gave nearly quantitative yields at a reduced photocatalyst dose of 5 mol% (Fig. 4c). Under the same

reaction conditions, the free ligand H$_4$SNDI exhibited decreased efficiencies in photocleaving C$_{Ar}$−Br (44%) or C$_{Ar}$−I (61%) bonds, and yielded less than 10% when photocleaving the C$_{Ar}$−Cl bond (Fig. 4c and Supplementary Fig. 30). Considering the competent thermodynamic driving forces of both Cd−SNDI and H$_4$SNDI for the reducing potentials of 4'-haloacetophenones and these photocatalytic results, we can conclude that the sharp contrast of photoinduced C$_{Ar}$−Cl inert bond dissociation by coordination polymer versus ligand (Supplementary Fig. 1) might due to the electron-transfer efficiency rather than the photoreducing power of (NDI$^{•-}$)$^*$[44,45,61].

As depicted in Fig. 4e, further control experiments demonstrated that the photoreducing efficiency of inert bonds, such as C$_{Ar}$−Br, was not directly correlated to the redox potential of electron donors, and that the highest conversion (96%) was achieved with dibutylamine. When aromatic fragment-containing electron donors were added, conversions dropped markedly. We speculated that the aryl group in the backbone of the electron donor interacted with the NDI moiety of Cd−SNDI to interfere with the π-systems, thereby limiting the photoreducing power and electron feeding ability.

## Femtosecond transient absorption analysis of photocatalysts

To gain a deeper understanding, femtosecond transient absorption (fs-TA) experiments were performed on radical anionic samples of H$_4$SNDI and Cd−SNDI in order to compare the results. Under a laser wavelength of 630 nm, the characteristic peaks of NDI$^{•-}$ of radical anionic ligand or radical anionic coordination polymer can be excited, while the excitation of neutral species is hampered. After shining radical anionic Cd−SNDI for ca. 1 ps (Fig. 5c), the excited state absorption (ESA) band covering a broad range of 400–550 nm was observed, of which the decay lifetime ($\tau = 164$ ps, probed at 480 nm; Fig. 5d) was close to the previously reported data of (NDI$^{•-}$)$^*$ (such as $\tau = 142$ ps)[45]. In contrast, the ESA band of radical anionic H$_4$SNDI was blue-shifted and exhibited a narrower shape (Fig. 5a), with a shorter decay lifetime ($\tau = 93$ ps, measured at 445 nm; Fig. 5b)[62]. For radical anionic Cd−SNDI, the ground state bleach (GSB) of the intra-ligand charge-transfer band at ca. 425 nm (Fig. 2a) was detected after 80 ps, implying the possible competition from inter-ligand electronic communication along the S···S-bridged network. In comparison, no bleaching of the intra-ligand charge-transfer band was observed with radical anionic H$_4$SNDI (Fig. 5a). As a supplement to these 630 nm experiments, the fs-TA of radical anionic H$_4$SNDI and Cd−SNDI were also compared under the laser irradiation of 480 nm that was closer to the 455 nm wavelength of LED in practical photocatalysis, which further revealed the complicated events and species evolutions after the excitation of NDI$^{•-}$, especially exhibiting the much longer decay lifetimes to disclose the strong tendency of inter-ligand charge transfer within the framework (see Supplementary Information for detailed analyses).

Based upon the pioneering works[44,63,64] and the above-mentioned comparative fs-TA study, the merged benefits of the prolonged excited-state lifetime of (NDI$^{•-}$)$^*$, the long-lived inter-ligand charge-separation state along the NDI string, and the in situ assembled Ar−Cl···NDI$^{•-}$ association within Cd−SNDI host framework, were believed to improve the accessibility of (NDI$^{•-}$)$^*$ towards the substrate and alleviate the consumptive back-electron transfer (Fig. 1d–g). These advantages might help to circumvent the diffusion-limited PET between the short-lived excited-state dye$^{•-}$ and substrate in the solution phase, which was crucial to the photocleavage of the C$_{Ar}$−Cl bond due to the stepwise dissociation mechanism and the reversibility of the first electron-accepting step[9,17,65].

## Substrate scope of Cd−SNDI catalysed photoreduction

Under the optimised conditions with dibutylamine as the electron donor, Cd−SNDI enabled the photocleavage of a series of substituted aryl halides with a high tolerance for functional groups (Fig. 6, **1–6, 10**).

**Table 1 | Optimisation of reaction conditions and control experiments for photoreductive $C_{Ar}$–Cl cleavage[a]**

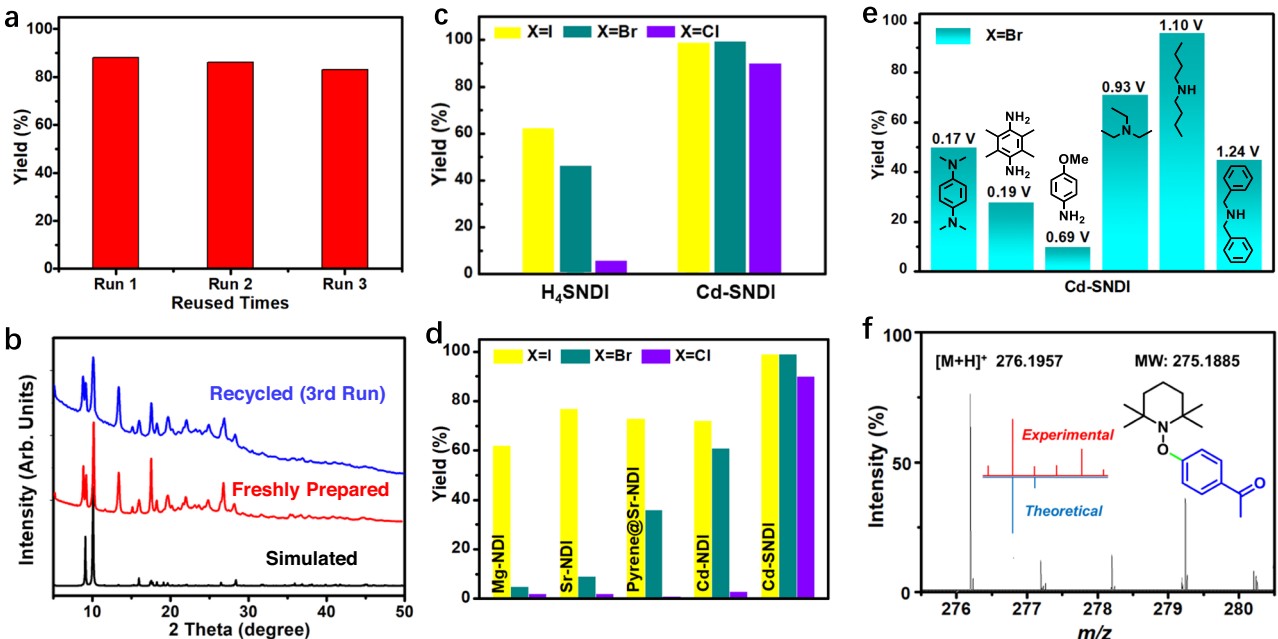

| Entry | Photocatalyst | Electron Donor | Yield (%)[b] |
|---|---|---|---|
| 1 | Cd–SNDI | $Et_3N$ | 74 |
| 2 | Cd–NDI | $Et_3N$ | <10 |
| 3 | Mg–NDI | $Et_3N$ | <5 |
| 4 | Sr–NDI | $Et_3N$ | <5 |
| 5 | Pyrene@Sr–NDI | $Et_3N$ | trace |
| 6 | Cd–SNDI | dibutylamine | 90 |
| 7 | Cd–SNDI | dibenzylamine | 48 |
| Variants of the optimal conditions | | | |
| 8 | No catalyst | | ND |
| 9 | $Cd(NO_3)_2$ as catalyst | | ND |
| 10 | $H_4SNDI$ as catalyst | | <5 |
| 11 | $Cd(NO_3)_2$ + $H_4SNDI$ as catalyst | | <5 |
| 12 | No electron donor | | ND |
| 13 | Under darkness | | ND |
| 14 | Under air atmosphere | | ND |
| 15 | With TEMPO (1.2 equiv.) added | | <10 |
| 16 | Cd–SNDI filtered off after 1 h | | 39 |
| 17 | Reaction conditions of **1a** in ref. [8]. | | 76 |

*ND* not determined.

[a]Reaction conditions: Ar–Cl (0.05 mmol, 1 equiv.), catalyst (10 mol%), DMF (3 mL), electron donor (72 equiv.), $N_2$ atmosphere, 455 nm LED, 40 °C, 4 h.

[b]GC yields.

**Fig. 4 | Investigation on the heterogeneous photocatalytic cleavage of $C_{Ar}$–halides by Cd–SNDI. a** Column chart of Cd–SNDI reuse experiments. **b** PXRD patterns of freshly prepared Cd–SNDI crystals (red), the simulated pattern based on the single-crystal data (black), and the recycled solid after 3 catalytic cycles (blue). **c** Comparison of the photocatalytic cleavage of 4′-haloacetophenone by Cd–SNDI and $H_4SNDI$. **d** Photoreductive yields of 4′-haloacetophenone catalysed by various NDI coordination polymers. **e** Photoreduction of 4′-bromoacetophenone with the photocatalyst Cd–SNDI and different amines as electron donors. **f** HRMS evidence of TEMPO entrapment of aryl radical when photocleaving 4′-chloroacetophenone by Cd–SNDI.

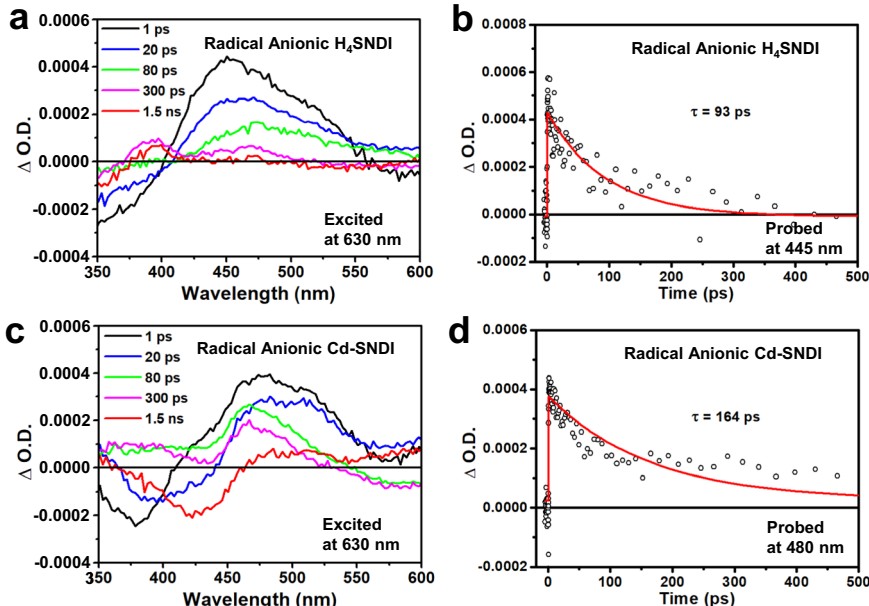

**Fig. 5 | The fs-TA analyses of radical anionic Cd–SNDI and H₄SNDI.** The 630 nm-laser excited femtosecond transient absorption (fs-TA) spectra of (**a**), radical anionic ligand H₄SNDI and (**c**), radical anionic coordination polymer Cd–SNDI at the indicated delay times. And the corresponding kinetic traces of ΔO.D. measured at (**b**), 445 nm for radical anionic H₄SNDI and (**d**), 480 nm for radical anionic Cd–SNDI. See Supplementary Information for the detailed experimental setups.

This heterogeneous photocatalytic method was applicable not only to substituted phenyl halides, but also to substrates containing carboaromatic fused rings (**8, 9**) and heteroaromatic cycles (**7, 11–13**). The bio-interesting molecules **12** and **13** were also involved in the photoreduction of the $C_{Ar}$–Cl/Br bond. Furthermore, pyrrole derivatives and styrene were suitable trapping reagents for the photogenerated aryl radical because of their high affinities to radical species, resulting in good to high yields of direct C–H arylation (**14–19**). Biologically important aryl sulfides and aryl phosphates can be obtained employing diaryl disulfide[66] and triethylphosphite[67] as trapping reagents, respectively (**20, 21**). In the presence of diboron pinacol ester[62], B₂(pin)₂, the corresponding arylated boron pinacol ester was successfully prepared, showing its value for accessing coupling precursors (**22**). This photocatalytic method also provided a one-step moderately yielding preparation of the antiepileptic drug perampanel[68] (**23**), demonstrating the tremendous potential of this approach for pharmaceutical applications. When employing the *n*-hexanyl branched perampanel precursor with a much bigger size of *ca.* 17.62 Å, the Cd–SNDI host framework did not accomplish the ingress/egress ability as well as when employing the normal-sized substrate (Supplementary Figs. 31 and 32). Not surprisingly, the corresponding photocatalytic reaction of a larger substrate gave a much lower yield (<10%) (**24**). These size-dependent experimental results suggested that the catalytic reaction occurred mainly within the pores or channels of the coordination polymer.

**Photocatalytic analysis of NDI-based coordination polymers**
A comparative study of heterogeneous photocleavage of aryl halide bonds was performed using coordination polymers with different NDI-stacking modes. To explore the structure-activity relationship of this heterogeneous photocatalytic approach, several NDI-based coordination polymers assembled from the non-decorated ligand H₄NDI were prepared (Supplementary Figs. 5, 11 and 12, and Supplementary Table 2)[69]. Cd–NDI exhibited three-fold interpenetrated networks; each cadmium ion was coordinated to four bidentate carboxylates from four deprotonated ligands, and two of the four carboxylate groups from one ligand were free (Supplementary Fig. 11). Multiple C=O⋯π and C–H⋯O=C interactions with distances between 3.08 and

3.62 Å were identified to stabilise this interpenetrated structure (Supplementary Figs. 5 and 11). Mg–NDI, which was prepared according to the literature method[69], possessed a similar three-dimensional network to that of Cd–SNDI, except for the absence of intra-ligand SEt groups and inter-ligand S⋯S contacts, and the discrete NDI units of Mg–NDI were parallel aligned and separated by 10.04 Å (Supplementary Fig. 5a). The solid-state absorption spectra of Mg–NDI and Cd–NDI showed visible light absorption bands at around 400 and 500 nm (Supplementary Fig. 14).

Solid-state differential pulse voltammetry (Fig. 7a) revealed the small differences in $E_{1/2}$ of those NDI-based coordination polymers. Similar to Cd–SNDI (Fig. 3b), Mg–NDI and Cd–NDI showed characteristic fingerprint bands of doublet transitions of NDI˙⁻ (Fig. 7b) upon Et₃N addition and the subsequent light irradiation (455 nm LED) in N₂. These findings indicated that the thermodynamic reducing powers of NDI/(NDI˙⁻)* within Cd–NDI, Mg–NDI, and Cd–SNDI were comparable. Despite the fact that the excited-state radical anionic forms of both Mg–NDI and Cd–NDI had sufficient reducing potentials (estimated to be *ca.* -2.24 V and *ca.* -2.13 V, respectively, vs. SCE; Supplementary Table 3)[58] for the cleavage of $C_{Ar}$–Cl bonds, neither Cd–NDI nor Mg–NDI were capable of reducing aryl chlorides (Table 1, entries 2, 3).

UV–vis spectra for suspensions of both Mg–NDI and Cd–NDI exhibited no noticeable absorption changes upon addition of the electron donor Et₃N (Supplementary Fig. 15), implying negligible interaction between this electron donor and framework. Upon addition of **1a** to the in situ generated radical anionic Mg–NDI suspension in DMF, the typical four groups of radical anionic peaks exhibited negligible variations compared to the case of Cd–SNDI (Supplementary Fig. 19), reflecting the insignificant interactions between radical anionic Mg–NDI and the Ar–Cl model substrate **1a**. Moreover, the NDI˙⁻ peaks of radical anionic Mg–NDI did not change when further shining light on the above mixture (Supplementary Fig. 19), reflecting the minimal PET from (NDI˙⁻)* to Ar–Cl substrate **1a** within Mg–NDI. Those results also reflected the vital role of the S-branch in mediating the in situ association and charge transfer between the coordination polymer and the external guest molecule. The Nyquist curves of EIS examinations confirmed the electrochemical impedance of the NDI-based coordination polymer candidates in the order of Cd–SNDI < Cd–NDI << Mg–NDI

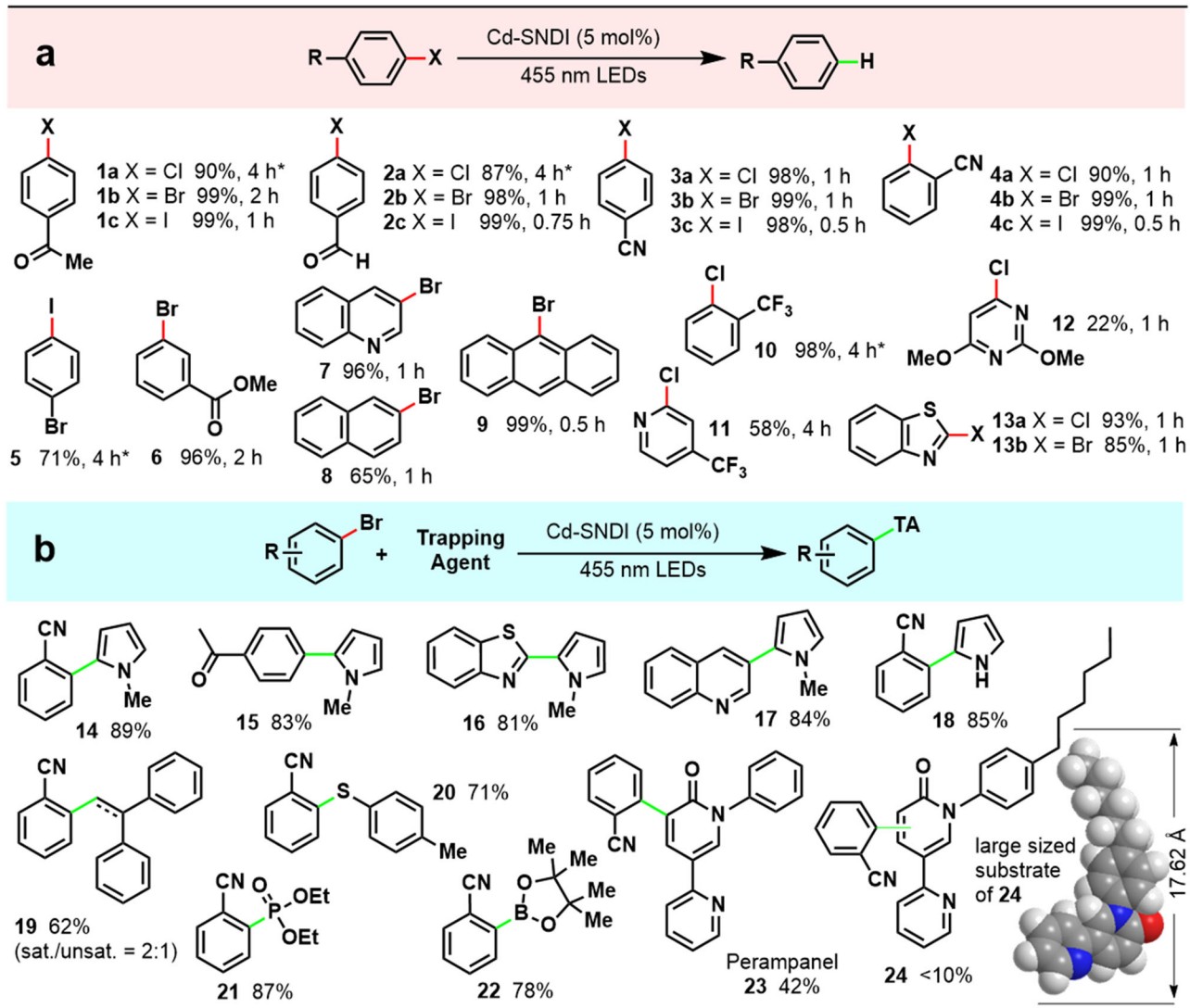

**Fig. 6 | Photoreductive cleavage of C_Ar–Halide bond and subsequent radical coupling with various trapping agents using photocatalyst Cd–SNDI.** **a** Reaction Conditions: substrate (0.05 mmol, 1 equiv.), Cd–SNDI (0.05 equiv.), dibutylamine (72 equiv.), DMF (3 mL), 40 °C, 455 nm LED, 4 h. GC yields (%). *10 mol % catalyst was used. **b** Reaction Conditions: substrate (0.1 mmol, 1 equiv.), Cd–SNDI (0.05 equiv.), dibutylamine (8 equiv.), trapping agent (25 equiv.), DMSO (1 mL), 40 °C, 455 nm LED, 4 h. Isolated yields (%).

(Fig. 7d). The positive correlation between charge-conducting abilities and photocatalytic performances of NDI-based coordination polymers were shown in Figs. 4d and 7d. The discrete arrangement of NDI moieties (Supplementary Fig. 5a) endowed Mg–NDI with sluggish charge conductance (Fig. 7d) and the lowest catalytic efficiency (Fig. 4d), and only the weakest C_Ar–I bond could be photocleaved by using Mg–NDI. In comparison, the n···π interaction-bridged NDI···NDI···NDI (A···A···A) string (Supplementary Fig. 5c) improved the charge-transfer ability in Cd–NDI (Fig. 7d). Cd–NDI successfully furnished the photocleavage of C_Ar–I and the thermodynamically more challenging C_Ar–Br bonds, but it proved to be insufficient for C_Ar–Cl cleavage, which was more dependent on electron injection compared with the case of breaking C_Ar–Br (Fig. 4d)[65]. In comparison to the cases of Mg–NDI or Cd–NDI, S···S-linked infinite NDI string in Cd–SNDI was akin to series-connected supramolecular wires (Fig. 1b–g)[38,39], improving the charge-transfer ability of material (Fig. 7d), which together with the in situ association of Ar–Cl···SEt–NDI·⁻ were believed to facilitate the proximal excitation and the effective electron feeding for splitting the inert C_Ar–Cl (Figs. 1d–g, 3c, d).

Within the two-fold interpenetrated structure of Sr–NDI prepared from ligand H₄NDI according to the literature method[69], the nearest NDI units were orthogonal to each other from two separate nets with the shortest interplanar C···C distance of 3.46 Å (Supplementary Fig. 5e). The orthogonal stacking pattern of two NDI cores and the spatial isolation of neighbouring NDI pairs (*ca.* 10.13 Å) hampered the inter-ligand electron transfer in Sr–NDI. When the electron-rich aromatic additive pyrene was intercalated into Sr–NDI to form the charge-transfer complex, a crystal pyrene@Sr–NDI was obtained with the same PXRD pattern as the original Sr–NDI (Supplementary Table 2 and Supplementary Fig. 16). X-ray single-crystal analysis revealed that two pyrene molecules were aligned in parallel within each void cavity of Sr–NDI, and the π···π distance between electron-donating pyrene (Pyr, D) and electron-withdrawing NDI cores (NDI, A) was *ca.* 3.32 Å, affording an infinite and twisted string of NDI···NDI···Pyr···Pyr···NDI···NDI (A···A···D···D···A···A) in a mixed aromatic stacking manner (Supplementary Fig. 5f). These structural characteristics might facilitate the inter-ligand charge transfer in the coordination polymer[70,71].

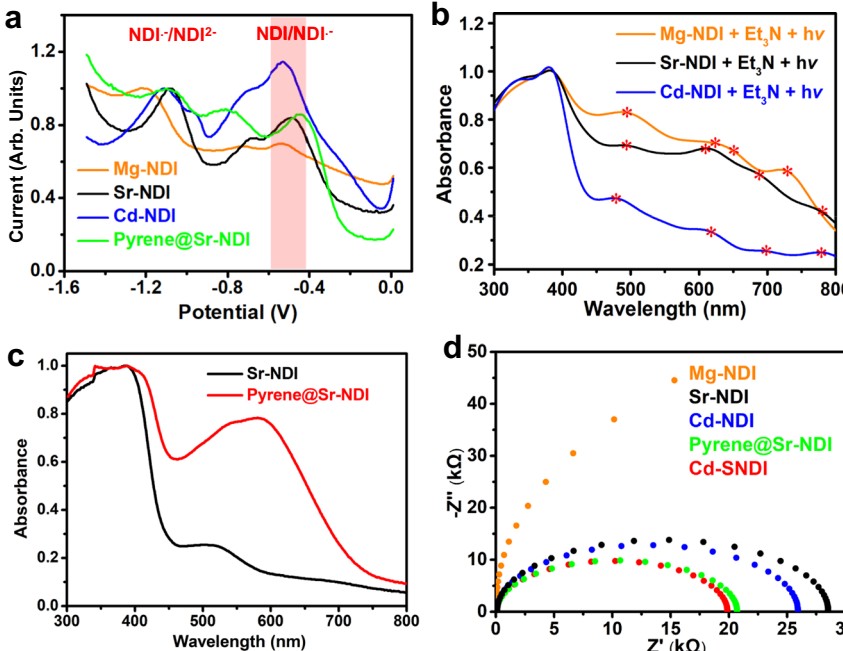

**Fig. 7 | The electro- and photo-properties of NDI-based coordination polymers. a** Solid-state DPV curves of Mg-NDI, Sr-NDI, pyrene@Sr-NDI and Cd-NDI. The pink shadow underlines the redox peak of NDI/NDI[.-]. **b** Solid-state UV–visible absorptions of Mg–NDI, Sr–NDI, and Cd–NDI with Et$_3$N addition followed by photoirradiation. **c** Solid-state UV–visible absorption spectra of Sr−NDI vs. pyrene@Sr−NDI. **d** EIS spectra of all the NDI-based coordination polymers in this work.

Incorporating pyrene in Sr−NDI gave rise to a broad absorption band centred at *ca*. 580 nm owing to the partial charge-transfer band (π → π*) between pyrene and NDI moieties (Fig. 7c)[21,71]. EIS examinations confirmed the lower electric resistance of pyrene@Sr−NDI compared to that of Sr−NDI (Fig. 7d). The aromatic Pyr...NDI (D···A) charge-transfer complex and the improved electrical conductivity of pyrene@Sr−NDI were believed to favour excited-state electron injection to the aryl halide substrate. As shown in Fig. 4d, Sr−NDI could only cleave the weakest C$_{Ar}$−I bond. In comparison, pyrene@Sr−NDI successfully photoreduced C$_{Ar}$−I and the more inert C$_{Ar}$−Br bond but could not split the much more challenging C$_{Ar}$−Cl bond. After irradiating the mixture of coordination polymers with Et$_3$N with a 455 nm LED, the solid-state UV−vis absorptions of radical anionic samples of pyrene@Sr−NDI and Sr−NDI were examined. Compared with the fingerprint NDI[.-] peaks (495 nm, 610 nm, 688 nm, 780 nm) of radical anionic Sr−NDI, the NDI[.-] peaks of radical anionic pyrene@Sr−NDI were merged into two broad peaks centred at 505 nm and 785 nm, respectively (Supplementary Fig. 20), which might be correlated with the mixed π···π stacking aggregation within pyrene@Sr−NDI (Supplementary Fig. 5f). Based upon the doublet-state $D_0 \rightarrow D_1$ transition peak at 780 nm (Sr−NDI, Fig. 7b) or 785 nm (pyrene@Sr−NDI, Supplementary Fig. 20), the photoreducing powers of excited-state radical anionic samples of Sr−NDI and pyrene@Sr−NDI were estimated to be enough to reduce the C$_{Ar}$−Cl model substrate such as **1a** (*ca*. -1.90 V, vs. SCE). However, as shown in Fig. 4d, none of them could complete the photoreduction of **1a**. Comparing the photoreduction performance of these two coordination polymers with that of Cd-SNDI, it was clear that not only the precious excited-state energy but also the electron delivery ability was vital for the photocatalytic reactivities of the short-lived (NDI[.-])*.

## Discussion

In summary, we incorporated chalcogen-containing moieties into a NDI and assembled S···S-bridged dye stacking in a coordination polymer. The coordination polymer with S···S-linked NDI dyes possessed significantly enhanced photoinduced charge separation and migration throughout the supramolecular wire-like framework when compared to NDI-based coordination polymers with different stacking modes. The non-aromatic nature of this dye stacking mode simultaneously fulfilled the competent excited-state reducing power of (NDI[.-])* by avoiding the aromatic stacking-induced energy dissipation. Moreover, the electron-donating S-branches contributed to the pre-association between the NDI-based framework and the guest molecules, such as electron donors or substrates. This heterogeneous approach provided a distinctive perspective for improving the kinetic aspect of PET, in contrast to the extensive effort on purchasing higher thermodynamic driving forces for SET-activation of inert bonds[11,14,72]. This strategy was validated to facilitate the formation of an encounter pair between (NDI[.-])* and an inert aryl halide substrate during consecutive photon excitation, which provided a possible solution for the long-pending issue of diffusion-limited electron transfer of the short-lived species in solution phase. The design of coordination polymer with S···S contact-linked dye stacking was further validated by probe reactions of consecutive photon excitation, which achieved the photocleavage of inert bonds such as aryl halides and the successive radical couplings to form the broad-scoped new bonds of C$_{Ar}$−C, C$_{Ar}$−S, C$_{Ar}$−P, and C$_{Ar}$−B with potential pharmaceutical applications. The results here unveiled a coordination polymer-based tool with the intrinsic advantage of supramolecular stacking for balancing the contradictory needs of the thermodynamic and kinetic factors of excited-state dyes in PET steps, paving the way toward future developments in green chemistry and efficient solar energy conversion.

## Methods
### Materials and measurements

1,4,5,8-naphthalene-tetracarboxylic acid dianhydride, acetic acid, 5-aminoisophthalic acid, NaSEt, Cd(NO$_3$)$_2$·4H$_2$O, Sr(NO$_3$)$_2$·4H$_2$O, pyrene, 4'-bromoacetophenone, 4'-chloroacetophenone, 4'-iodoacetophenone, 2-bromobenzonitrile, *N*-methyl pyrrole, triethylamine,

$N,N'$-tetramethylphenylenediamine, 2,3,5,6-tetramethylbenzene-1,4-diamine, $p$-Anisidine, dibutylamine, and dibenzylamine were chemically pure, dimethylformamide (DMF) and dimethyl sulfoxide (DMSO) were analytically pure. All of the above-mentioned chemicals were purchased from Sigma-Aldrich, TCI, J&K Scientific, and Energy Chemical, and used as received without further purification unless particularly pointed out. The ligand $N,N'$-bis(5-isophthalic acid) naphthalene diimide (H$_4$NDI) was synthesised according to the literature[69]. $^1$H-NMR spectra were measured on a Varian INOVA 400M spectrometer, and the solvent was CDCl$_3$ unless it was pointed out. Thermogravimetric analysis (TGA) was carried out at a ramp rate of 10 °C/min in a nitrogen flow with a Mettler-Toledo TGA/SDTA851 instrument. Products were purified by flash column chromatography on 200−300 mesh silica gel, SiO$_2$. The PXRD patterns were collected by Rigaku D/Max-2400 X-ray diffractometer with Cu $K\alpha$ radiation ($\lambda = 1.54056$ Å). Absorption spectra were recorded on an HP 8453 spectrometer. Solid-state differential pulse voltammetry (DPV) (Figs. 2b, 7a) was measured by preparing a carbon-paste working electrode: a well-ground mixture of sample and carbon paste (graphite and mineral oil) was set in the channel of a glass tube and connected to a copper wire. A platinum-wire counter electrode and a Ag/AgCl reference electrode were used. Measurements were performed by this three-electrode system in a 0.1 M KCl solution at a scan rate of 100 mV s$^{-1}$, in the scan range of −1.5 to 0 V. The EIS was performed on the Zahner Zennium electrochemical workstation in a standard three-electrode system with the photocatalyst-coated glassy carbon as the working electrode, a platinum-wire as the counter electrode, and a Ag/AgCl as a reference electrode. In the case of EIS measurements (Figs. 3d and 7d), the coordination polymer (2 mg) was dispersed into a mixed solution with 30 µL Nafion, 250 µL ethanol and 250 µL deionised water, and the working electrode was prepared by dropping the suspension (100 µL) onto the surface of the glassy carbon electrode. The working electrode was dried, and EIS measurement was performed with a bias cathode potential of 0.3 V.

### Synthesis of H$_4$NDI

1,4,5,8-naphthalene-tetracarboxylic acid dianhydride (6.70 g, 25.0 mmol) was dissolved in 125 mL acetic acid within a 250 mL round-bottom flask. The mixture was heated at 60 °C for 10 min before adding 5-aminoisophthalic acid (9.05 g, 50.0 mmol). After twelve hours of reflux at 120 °C, 100 mL of deionised water was added to the resulting mixture at room temperature. Off-white H$_4$NDI solid was obtained in a yield of 77% (12.0 g) after filtration and drying.

### Synthesis of H$_4$SNDI

The detailed synthesis steps of H$_4$SNDI are provided in the supplementary information.

### Synthesis of Cd−SNDI

H$_4$SNDI (14 mg, 0.020 mmol), Cd(NO$_3$)$_2$·4H$_2$O (12 mg, 0.040 mmol), 0.05 mL HCl (3 M) were mixed with 2.0 mL dimethylformamide (DMF) and 0.2 mL H$_2$O. The resulting mixture was heated in a 25 mL Teflon-lined autoclave at 100 °C for four days, then allowed to cool slowly to room temperature. Red to black crystals were obtained in 26% yield (based on H$_4$SNDI). Anal. Calcd (%) for Cd−SNDI (CdC$_{43.5}$N$_5$S$_2$O$_{15.5}$H$_{39}$): C, 49.46; H, 3.72; N, 6.63. Found: C, 48.80; H, 4.28; N, 7.09.

### Synthesis of Cd−NDI

A mixture of H$_4$NDI (24 mg, 0.040 mmol), Cd(NO$_3$)$_2$·4H$_2$O (12 mg, 0.040 mmol), and 0.08 mL HCl (3 M) were mixed in 2.0 mL DMF. The resulting mixture was heated in a 25 mL Teflon-lined autoclave at 100 °C for three days, then allowed to cool slowly to room temperature. Orange to red crystals were obtained and suitable for X-ray structural analysis. Yield: 70% (based on H$_4$NDI). Anal. Calcd (%) for Cd−NDI (C$_{77}$H$_{68}$CdN$_{11}$O$_{28.5}$): C, 53.90; H, 3.99; N, 8.98. Found: C, 53.78; H, 4.07; N, 9.14.

### Synthesis of pyrene@Sr−NDI

H$_4$NDI (21 mg, 0.035 mmol), Sr(NO$_3$)$_2$·4H$_2$O (24 mg, 0.085 mmol), pyrene (42 mg, 0.21 mmol), 0.2 mL HCl (3 M) were dissolved in 4.0 mL DMF. The resulting mixture was heated in a 25 mL Teflon-lined autoclave at 90 °C for 24 h, then allowed to cool slowly to room temperature. Red to black crystals were obtained in 70% yield (based on H$_4$NDI).

### Single crystal X-ray crystallography of Cd−SNDI, Cd−NDI, and pyrene@Sr−NDI

Intensities were collected on a Bruker SMART APEX CCD diffractometer with graphite monochromated Mo-K$\alpha$ ($\lambda = 0.71073$ Å) using the SMART and SAINT programmes[73,74]. The structure was solved by direct methods and refined on $F^2$ by full-matrix least-squares methods with SHELXTL version 5.1[75]. Non-hydrogen atoms of the ligand backbones were refined anisotropically. Hydrogen atoms within the ligand backbones were fixed geometrically at calculated positions and allowed to ride on the parent non-hydrogen atoms. Hydrogen atoms of the solvent molecules were found from the different Fourier MAP, but refined using the riding model with the thermal parameter fixed at 1.2 times of the oxygen atoms they attached. Several bond distance constraints were used to help the refinement of the solvent moiety. For Cd−SNDI, the atoms on the naphthalene ring were fixed at the same plane.

### Transient absorption (TA) spectra of radical anion of Cd−SNDI and H$_4$SNDI

The detailed setup of the femtosecond pump-probe TA measurements was similar to those described in prior studies[76]. Briefly, the laser source was a regenerative amplified Ti:sapphire laser system (Coherent; 800 nm, 70 fs, 6 mJ/pulse, 1 kHz repetition rate). The 800 nm output pulse was split into two parts with a 50% beam splitter. One part was used to pump an OPA, which can generate a wavelength-tunable laser pulse from 250 nm to 2.5 µm using a pump beam. Another part was attenuated with a neutral density filter and focused into a sapphire or CaF$_2$ crystal to generate a white light continuum for the probe beam. The delay between the pump and probe pulses was controlled by a motorised delay stage. The pump pulses were chopped by a synchronised chopper at 500 Hz, and the absorbance change was calculated with two adjacent probe pulses (pump-blocked and pump-unblocked). For all TA measurements, samples were filled in 1 mm airtight cuvettes prepared in a N$_2$-filled glove box and measured under ambient conditions.

### Typical procedure for the photoreductive cleavage of aryl halides

A glass tube was filled with aryl halide (0.05 mmol, 1 equiv.), Cd−SNDI (0.0025 mmol, 0.05 equiv.), a mini-stirrer, and dry DMF (3 mL) as solvent. The resulting mixture was degassed with N$_2$ bubbling for 20 min, and then sealed. An electron donor such as dibutylamine (3.6 mmol, 72 equiv.) was added during the bubbling process. The reaction mixture was irradiated at 40 °C by a 455 nm LED loop for 4 h. After the reaction, the reaction mixture was filtrated via a 0.22 µm filter, and the resulting clear solution was devoted to gas chromatography (GC) analysis.

### Typical procedure for photoreductive cleavage and successive C−C/Heteroatom bond formation

A glass tube was filled with aryl halide (0.1 mmol, 1 equiv.), Cd−SNDI (0.005 mmol, 0.05 equiv.), a mini-stirrer, DMSO (1 mL) as solvent, and radical trapping agent (2.5 mmol). The resulting mixture was degassed with N$_2$ bubbling for 20 min, the electron donor dibutylamine (0.8 mmol, 8 equiv.) was added during degassing, and the reaction tube was sealed. The reaction mixture was irradiated by a 455 nm LED loop. The reaction progress was monitored by GC analysis. For

workup, the reaction mixture was filtrated via a 0.22 µm filter, and the resulting clear solution was transferred into a separating funnel, and about 10 mL distilled water and 2 mL saturated brine were added. The obtained mixture was extracted three times with ethyl acetate ($3 \times 10$ mL). The combined organic layers were dried over $Na_2SO_4$, filtered, and concentrated under the vacuum. Purification of the crude product was achieved by flash column chromatography using petroleum ether/ethyl acetate as eluent on a silica gel column.

## Data availability

Materials and methods, and additional tables and figures related to coordination polymers and related intermediates (SXRD, PXRD, TGA, EPR, UV–vis, DPV, fs-TA) and photocatalytic details and products (NMR, GC) are available within the Supplementary Information. Crystallographic data for the structures reported in this article have been deposited at the Cambridge Crystallographic Data Centre, under deposition numbers CCDC 2246332 (Cd–SNDI), 1938743 (Cd–NDI), and 2076326 (pyrene@Sr–NDI). Copies of the data can be obtained free of charge via https://www.ccdc.cam.ac.uk/structures/. The other source data are available from the corresponding authors upon request. Source data are provided with this paper.

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

## Acknowledgements

The work was supported by the National Natural Science Foundation of China, No. 21971031 (T.Z.), No. 21820102001 (C.D.), 21890381 (C.D.), 21231003 (C.D.), and 21901033 (L.Z.). Special thanks are due to Dr. Rui Cai, Dr. Dan Wang, and Dr. Liyan Zhang at Instrumental Analysis Center and State Key Laboratory of Fine Chemicals, Dalian University of Technology, for the assistance with experiments and analyses on fluorescence, EPR, and ground-state UV/vis/NIR absorption, respectively.

## Author contributions

L.Z., T.Z., and R.L. conducted most of the experiments and ran most of the data analyses; T.Z. and L.Z. co-wrote the manuscript; Z.W. and A.I.D. ran a portion of reactions; W.T., J.Z., and R.L. ran the fs-TA experiments; T.Z. and J.F. analysed the fs-TA results with the help of K.W. and W.T.; C.H. contributed to the crystal structure analysis and discussion of experiments; X.G. ran a portion of the crystal structure analysis; C.D. and T.Z. conceived and supervised the project and revised the manuscript.

## Competing interests

The authors declare no competing interests.
