## [Peer Review File · Nature Communications]

Chalcogen-bridged coordination polymer for the photocatalytic activation of aryl halidesReviewers' Comments:

Reviewer #1:

Remarks to the Author:

In this manuscript, Duan and coworkers report a naphthalene diimide photocatalyst designed to self-assemble into aggregated dyes. The manuscript focuses on characterizing the catalyst, its photophysical behavior, and the photocatalytic activity of the system. Overall, this system appears to offer comparable scope of aryl halides that can be activated to the related homogeneous pyrenediimide (PDI) systems initially reported by Koenig and coworkers in 2014 (Ref 8). However, it does appear from the authors data that substantially enhanced rate is observed relative to this prior work.

A key element of this manuscript is that these aggregated dyes offer enhanced photocatalytic properties relative to their homogeneous counterparts. The authors compare a non-aggregated analog of their catalyst to their typical conditions in Table 1 as a satisfactory control regarding the role of the aggregation in making this current system work. However, I was not able to find a direct comparison to Koenig's conditions in the manuscript. At a minimum, this comparison should be provided in Table 1 to benchmark this new photocatalytic system relative to the related PDI system. However, even if this new system offers advantages relative to Koenig's 2014 report, this is not sufficient to be a very significant synthetic advance as substantial improvements in photocatalytic aryl halide reduction have followed that report (see below).

If one looks beyond diimide photocatalysts, the Koenig work is no longer the state of the art for reductive aryl halide activation. Recent work from Wu (doi: 10.1021/jacs.1c05994) and Wickens (doi: 10.1021/jacs.1c05988) substantially enhance the scope of aryl halides which can be activated by photoredox catalysts. Additionally, work from Read de Alaniz and Hawker (doi: 10.1039/C5CC04677G) also illustrated a comparable scope to the Koenig work shortly after publication and currently represent the state of the art for a typical one-photon mechanism. These advances should be cited and the introduction should be adjusted accordingly. I think to illustrate an advantage of this new heterogeneous photocatalytic system presented in this manuscript, at minimum comparable results to these state-of-the-art systems should be provided.

In closing, let me note that there is clearly a tremendous amount in this manuscript beyond synthetic chemistry. I am intrigued by several of the ideas presented in this manuscript regarding how short excited state lifetimes can be circumvented, however, I am not qualified to evaluate the conclusions drawn in this regard. I will leave the other aspects of the manuscript for other referees to evaluate given my specific expertise. From my perspective, this manuscript does not represent a synthetic advance that I expect would be of sufficiently broad interest for the readership of Nature Communications.

Reviewer #2:

Remarks to the Author:

Summary:

The manuscript describe the synthesis of the a coordination polymer using naphthalene diimide (NDI) substituted with SEt and Cd(II), achieving an S...S interaction-mediated segregated stacking of dyes, which facilitated the electron mobility throughout segregated NDI-strings while simultaneously maintaining the reduced power of the ground state and the excited-state reducing power of its reduced form. This particular characteristic allows the authors to demonstrate the superior behavior of this heterogeneous photocatalyst in compare to single-molecule dye photocatalysis and other coordination polymers in a consecutive photocatalysis (conPET) for the photoreduction of aryl halides including aryl chlorides with EWG. The photocatalytic system also demonstrates good performance in successive radical couplings allowing the formation C-C, C-S, C-P, and C-B bonds by known

procedures.

The authors claims that enhanced the electron delocalization within the polymer which improve the PET from excited-state radical anion (NDI^{-•})* providing a strategy to circumvent the short lifetime of excited states. This strategy is interesting and I believe it would reach a broad audience in which ET processes from excited states are important and the work could contain the merits for publication in this journal.

However, several points should be addressed for a complete support of the conclusions and claims which I described in the next section.

I recommend publication of the manuscript after revision of the points described in the next sections.

General comments:

- The most obvious missing is poor mention of excited state lifetimes of the obtained photocatalyst in the discussion, perhaps the "mobility" or "the electron delocalization within framework" that the author mentions in the paper implies a "longer" excited state lifetime for the excited state of the catalyst. I believe the author should make an effort to make a better link between the lifetime in a single molecule and a "lifetime" in the heterogeneous photocatalyst.

-In relation to this, the mentions to the lifetimes of NDI^{-•}* must be revisited and should be included in the discussion of the results obtained. I believe citation 35 is not the best to compare with the actual system and the range 105-260 ps is more appropriate than "dozens of picoseconds" . The authors should include a mention to this including citation from this works from Wasielewski et al 10.1021/jp000706f, Schanze et al 10.1021/acs.jpcc.9b06303 (perhaps together with 10.1039/C6FD00219F). (see specific comment for page 2)

- The characterisation of the new Cd-SNDI catalyst and its synthetic advantages are well correlated in an Heterogeneous photocleavage of aryl halide. However, some control experiments are missing and must be performed for a complete mechanistic analysis.

The interaction with the substrate Ar-Cl is not analyzed in these sections which is surprising taking into account the observation made for the electron donors at the end of page 10. It is necessary to clarify the possibility of a pre-association of the polymer photocatalyst and the halogenated substrates performing (control)experiments similar to the ones performed with Et3N (Fig 2) to clarify this situation which will be important for the proposed mechanism.

-Another important interaction is the effect of Ar-Cl in the properties of the radical anion in Cd-SNDI (in the dark) which should also be analyzed. See for instance: DOI:10.1002/cctc.202100359

- Several experimental details are missing to assure reproducibility of the result by the scientific community. Experiments of Fig 2, S4 and S5 should be clearly described in the SI with a clear indication of concentration (or amount of solid), the amount of Et3N added, time of irradiation, irradiation source, etc.

- NDI core has characteristic absorption peaks 350-400, present in core-substituted and - unsubstituted NDIs. The author should measure all spectra starting at 300 nm to access these peaks. For instance, this will give additional information of the changes driven by the addition of Et3N and by photostimulation.

- I strongly recommend putting a picture of the setups used for the synthetic experiments in the SI.

Specific comments, changes and missing data:

Page 2

Line 3: I would suggest deleting "Exogenous..."

Line 10: I would suggest replacing "between" by "working with" or "when working with".

Line 11: "Scheme S1" must be changed by "Scheme S1a"

Line 11-14: This frase in somewhat confusing, are the authors referring to the excited state E⁰ or the ground state E⁰ of the photocatalyst? In any case, the authors should include a citation to support this affirmation. Besides, it should be taking into account that the lifetime of excited states of perilene-

and naphthalene diimides radical anions (PDI^{-•}, NDI^{-•}) don't show a clear correlation of E⁰ reduction potential of the excited state and its lifetime. Compare for instance data in the works of Wasielewski et al (10.1021/jp000706f, 10.1021/acs.jpcc.9b06303) and Schanze et al (10.1021/jacs.9b13027).

Line 15: "...the high enough electron delivery ability of excited-state dye..." It is not clear the meaning of this phrase. It seems a rather complicated way to express the "kinetic" of the electron transfer from the excited state. Perhaps the authors try to indicate the implication of different events or factors affecting the "kinetic" of the electron transfer but I believe the sentence is a little ambiguous and should be rewritten.

Page 4

Line 14: "...branches," should be changed by "...branches (Fig. 1a),"

Line 21: Change "Scheme S1" by "see comparison in Scheme S1"

Page 7

Line 7: EPR of free ligand is missing in Fig S6.

Line 14-17: The absence of fluorescence is not a direct consequence of the short lifetime but the competitive fast non-radiative decay. I believe the phrase should be changed. I recommend deleting "Owing to the extremely short lifetime scale of dozens of picoseconds for excited-state (NDI^{-•})*,35" See discussion in general comments.

Line 20: Reference electrode should be included (I presume "vs SCE")

Page 8

The experimental details for experiment of Fig 2d are missing and should be included in the SI. It is not clear to me how this experiment "demonstrated the superior electronic conductance of in situ generated excited-state (NDI^{-•})* within Cd-SNDI...". The superior conductance of the ground NDI^{-•} could be enough if some interaction of Ar-Cl with NDI^{-•} allows its excitation "close" to a Ar-Cl molecule.

Page 9

Lines 9-10: Is "90 %" the conversion or the product yields? and more importantly are data in table product yield or conversion?

Table 1. The amount of electron donor looks excessive in table 1, more if we take into account others conPET catalysis(see for instance: 10.1002/cctc.202100359) and the results of table 2 in which 8 equiv. are enough to perform a more complex transformation in very similar conditions. Being this an important point in having friendly conditions for synthesis, the author should perform the model reaction of table 1 with different equivalents of the best electron donor including a condition with 1 equiv. (or 1.1) and provide this information in the SI.

Page 10

Line 2: Citation 35 should be changed by more appropriate ones related to the lifetime including the works of Wasielewski et al (10.1021/jp000706f, 10.1021/acs.jpcc.9b06303) and Schanze et al (10.1021/jacs.9b13027) and relate to conPET reactions with NDI photocatalyst Bardagi et al. DOI: 10.1002/cctc.202100359

Line 3-4: If TEMPO is acting only as radical scavenger conversion of this reaction should not be much affected and the adduct of TEMPO-(4-acetylphenyl) should be formed in appreciable amount. What is the conversion for this experiment?

Was the adduct of TEMPO-(4-acetylphenyl) identified?

Line 10-18: Although the better performance of Cd-SNDI compared to H4SNDI is clear. The conversion of the CAr -I/Br/Cl for Cd-SNDI shows an inconsistency for conversion of CAr-I that should be in the order of CAr-Br/Cl but no lower. This difference could not be explained by measurement

error because it is almost 20% lower than expected. Do the authors have an explanation for this observation?

Page 15

Line 18: PXRD pattern of Sr-NDI is missing in Figure S14, so it is not possible to compare with pyrene@Sr-NDI.

Page 17

Line 14: change "discussion" by "conclusion"

Line 15: "...we modified..." should be changed for "...we incorporate..."

Figures

Fig. 1:

- The phrase "SEt group was disordered in crystallography thus it is drawn on every possible position." must be added as indicated in Fig. S2.

- "c, S...S" must be changed by "c, S...S"

- I would suggest decorating the figure with some element that more clearly indicates the a b c d parts of the figure. For instance, a gray "water-mark" square in each section.

Fig 2:

Fig 2a: Is this a "solid-state" absorbance or a "Cd-SNDI suspension" as in Fig S4?

Irradiation time for Fig 2a,b should be indicated here or in a figure in the SI

Fig 3:

-Fig 3b: "experimental" should be changed by "freshly prepared" "recovered" should be replaced by "recycled (3rd run)"

- The amount of catalyst should be added for experiments in Fig 3c.

Fig S6: EPR of free ligand is missing

Citation:

- Please check Ref. 17, it looks inappropriate for the previous sentences.

- Key reference is missing that are more appropriate that cite 35 related to NDI-- lifetime and synthetic approach. See general comments and specific comments for pages 2 and 10.

D. Gosztola, wMark P. Niemczyk, W. Svec, A. S. Lukas, M. R. Wasielewski, Excited Doublet States of Electrochemically Generated Aromatic Imide and Diimide Radical Anions J. Phys. Chem. A 2000, 104, 6545–6551. DOI: 10.1021/jp000706f

S. Caby, L. M. Bouchet, J. E. Argüello, R. A. Rossi, J. I. Bardagi, Excitation of Radical Anions of Naphthalene Diimides in Consecutive- and Electro-Photocatalysis** ChemCatChem 2021, 13, 3001–3009. DOI: 10.1002/cctc.202100359

- Ref. 46, points to a [(11)C]-cyanation and should be changed.

- Ref 47 and 55 are the same.

Reviewer #3:

Remarks to the Author:

This is an interesting piece of work that is significantly enhanced by the C-X bond cleavage reactions

and subsequent studies. The reactivity is certainly interesting and will provoke others to look into this idea. The improvement in the S-NDI systems is also intriguing and therefore I am convinced that this paper should be considered for publication in Nat Comm. However there are a number of points that the authors need to address both to improve the presentation but also, most importantly, to clarify some scientific points. I am concerned by the PXRD of Cd-SNDI and the purity of the samples. I am also concerned that the authors are unclear in terms of describing how they think the coupling reactions are proceeding. This is a very important point and in its current form I am left with more questions than answers. Therefore I suggest major revisions.

The authors use the phrase 'segregated aggregation'. I am not sure what they really mean by this. Do they just mean that the NDI components are spaced regularly? Aggregation usually refer to molecules that are in close proximity, in direct contact with each other. It would be helpful if they could be specific in their description of what they are trying to achieve.

In Figure 1 it is unclear what the distance 3.44 refers to? I realise from the text that this refers to the S-S contact but this is not clear from the figure. Perhaps an inserted, enlarged figure, might help here as this is an important feature. The authors need to modify the figure caption to specify that the distances are shown in angstroms.

The PXRD of Cd-SNDI (Figure S1 and Figure 3) shows a strong reflection at 12-13 degrees (2theta) which does not appear to be in the simulated pattern. What is this due to? Can the authors provide the PXRD pattern for a pure sample? This brings into question some of the results. If the sample isn't pure then what is leading to the effects observed?

However, the photoreductive C-Cl cleavage is impressive. I am genuinely interested in the authors opinions of how this process works. Do they think that the aryl-halide bond cleavage happens within the MOF or at the surface of the particles? DO they think that the charge is transferred to the crystal surface and then the reaction takes place there? I am sure it is tempting to think it happens within the framework pores but if so I assume the cleaved radical would have to migrate from the pores to react with the other substrate as it is difficult to see how both can fit within the pores, there is insufficient space in some examples, e.g compound 23. I note there is a lower yield in that case but I doubt that this is accounted for solely by encapsulation or surface phenomena. I appreciate that this is a very difficult thing to establish. However, the authors really should discuss this. If the authors could confirm that some of the substrates are too large to fit into the pores then this would suggest that the reactions are solution based.

Reviewer #4:

Remarks to the Author:

Duan and co-workers reported a photoreducing coordination polymer featuring EtS-substituted NDI radical anions in this communication. Compared with the related coordination polymers that either lack the sulfur substituent or have a large separation between NDI units, the new heterogeneous photocatalyst displays superior SET reactivity, well demonstrated in various aryl-C/S/P/B bond-forming reactions. The authors suggested that the higher performance is caused by better electron conductivity/transportation between the NDI units, facilitated by chalcogen-chalcogen interactions, as the authors further proposed. While I found the experimental discovery timely and exciting, I do not feel comfortable recommending this work for publication in its current form. I would point out below some concerns for the authors to consider. Most issues are related to the claim of chalcogen-mediated phenomena, some are misconceptions, and others are language issues that may seriously affect the readers' understanding of this work.

(1) Chalcogen-chalcogen interactions were proposed to be important in the present system. I was, however, surprised to see no reference for the study of chalcogen-chalcogen interactions provided in

this manuscript. I cannot see how Refs 18, 31, or 32 are relevant.

(2) In the context of "multiphoton consecutive excitation", Wenger's review in *Angew. Chem. Int. Ed.* 2020, 59, 10266 should be cited.

(3) It was stated "Purchasing the more elevated photoredox potentials through decoration of dyes has attracted extensive attention, which is at the expense of the more instability and even shorter lifetime of the corresponding highly reactive excited states...". This is a wrong statement. Photoredox driving force, chromophore stability, and excited-state lifetime do not necessarily show a positive or inverse correlation with one another. Another Wenger's paper doi.org/10.1002/anie.202110491 demonstrate this well.

(4) The quality of the X-ray crystal structure should be improved. The cif data of Cd-SNDI shows highly twisted and non-planar SNDI units. This is very questionable, besides the issue of serious disorders in other parts of the overall crystal structure. Furthermore, such non-planar SNDI was not seen in Figure 1a. Please clarify the inconsistency.

(5) In "... Cd-SNDI showed a wider absorption band that covered the visible light region (Fig. S4), indicating the presence of abundant charge-transfer interactions within the framework." What is the origin of these CT interactions? How do the authors know that they are indeed CT interactions?

(6) Interestingly, only Cd-SNDI and H4SNDI could form charge-transfer complexes with Et3N, whereas other M-NDI samples display no changes in the UV-Vis absorption in the presence of Et3N (Figure 2 vs Figure S13). Since the electron deficiency (based on the reduction potential) is similar across the board, was the CT interaction caused by chalcogen bonding between Et3N:...SEtNDI? If so, would the inclusion of Et3N in the structure of Cd-SNDI change the geometric relationship between the NDI units?

(7) The authors proposed electron delocalization between NDI in Cd-SNDI. This proposal is one of the central ideas but was not supported by experimental evidence. Judging from the UV-Vis absorption of NDI radical anion in Cd-SNDI, the radical is very likely centered on one single NDI. Or, is there a charge-resonance band to support this proposal? If, however, the authors are considering charge transport over several NDI sites on a long timescale (as opposed to "delocalization"), they may consider using EPR to probe the existence of such a phenomenon (or the absence of it).

(8) The authors used impedance measurements to probe the relative electric conductivity of various samples. How much of the measured impedance is contributed by the inter-NDI charge transport, and how much by other processes (e.g. interfacial electron transfer)? What is the purpose of Nafion? Does Nafion affect the intrinsic conductivity?

(9) The results of photocatalytic cleavage of 4'-haloacetophenone by Cd-SNDI seem to be inconsistent between those shown in Figure 3c (Cl>Br>I) and in Figure 4 (Cl (90% 4h)in situ reduce photocatalyst to get extreme reduction potential
3CzEPAIPN / 456 nm ^g	-2.94	C-C, C-P, C-B	Photocatalyst family of donor-acceptor cyanoarenes can undergo conPET process for extreme reduction potential
Cd-SNDI / 455 nm ^h	-2.15	C-C, C-P, C-B, C-S	Dye aggregation can promote the reactivity of the short-lived radical anion excited state towards substrate

^a V vs. SCE. ^b 10.1126/science.1258232 ^c 10.1021/jacs.9b13027 ^d 10.1039/c5cc04677g. ^e 10.1021/jacs.9b12328. ^f 10.1021/jacs.1c05988. ^g 10.1021/jacs.1c05994. ^h **this work**.

In closing, let me note that there is clearly a tremendous amount in this manuscript beyond synthetic chemistry. I am intrigued by several of the ideas presented in this manuscript regarding how short excited state lifetimes can be circumvented, however, I am not qualified to evaluate the conclusions drawn in this regard. I will leave the other aspects of the manuscript for other referees to evaluate given my specific expertise. From my perspective, this manuscript does not represent a synthetic advance that I expect would be of sufficiently broad interest for the readership of Nature Communications.

Response: Thanks for the reviewer's recognition for our work beyond the synthetic chemistry and give us the chance to explain and express our idea more clearly. As we have responded to your first question, the key point of our work is to improve the PET from excited-state radical anion (NDI^{-•})* to Ar-X substrate via tuning dye stacking manner of photocatalyst units within a coordination polymer. The reason we chose the photocleavage reaction of Ar-X is that for this bond cleavage of Ar-Cl, the sufficient electron transfer from potent reducing photocatalyst to the Ar-Cl is crucial and the lifetime of potent reducing photocatalyst is often very short. Thus, besides the competent redox power, the lifetime of excited state, the electron delivery pathway or the interaction model between electron donor and electron acceptor, is also vital for the photocatalytic result of this photoreduction reaction (*J. Am. Chem. Soc.* 2022, **144**, 7043-7047).

In our work, the specific arrangement between the external electron donor/Ar-Cl substrate and the NDI string of coordination polymer, namely the S-mediated segregated stacking, could extend the lifetime of excited state and enhancing the collision chance of excited state and substrate, which gave rise to the superior photocatalytic behaviour in compare to the other cases using single-molecule dye or other coordination polymers as photocatalysts. As stated by **Reviewer #2**, "This strategy is interesting and I believe it would reach a broad audience in which ET processes from excited states are important and the work could contain the merits...". In view of the prevalence and significant role of PET for organic synthesis, we speculate that our work can rise the interests from chemists, materials scientist and biologist who are involved in research related to the generation and transportation of excitons.

Moreover, the more ordered and program-controlled characteristics of electron transfer in this heterogeneous system, not only resulted in the photocatalytic efficiency comparable to the state-of-the-art works, but also can help to ignore the varied „kinetic“ needs for electron transfer events of different reaction types and substrate scopes, which did not require the extensive screening works or distinct reaction conditions that usually needed in homogeneous manner, thus realizing the different reaction types (hydrogenation or radical coupling after C-X cleavage) and a broader substrate scopes (C-C, C-P, C-B, C-S, and suitable for preparing antiepileptic drug perampanel) under similar or slightly revised reaction conditions. These performances together with the intrinsic recyclability of heterogeneous catalyst and the less residual contamination to products still reflected the distinctive characteristics of this method in synthetic chemistry from a heterogeneous perspective.

Reviewer #2:

Summary:

The manuscript describe the synthesis of the a coordination polymer using naphthalene diimide (NDI) substituted with SEt and Cd(II), achieving an S...S interaction-mediated segregated stacking of dyes, which facilitated the electron mobility throughout segregated NDI-strings while simultaneously maintaining the reduced power of the ground state and the excited-state reducing power of its reduced form. This particular characteristic allows the authors to demonstrate the superior behavior of this heterogeneous photocatalyst in compare to single-molecule dye photocatalysis and other coordination polymers in a consecutive photocatalysis (conPET) for the photoreduction of aryl halides including aryl chlorides with EWG. The photocatalytic system also demonstrates good performance in successive radical couplings allowing the formation C-C, C-S, C-P, and C-B bonds by known procedures.

The authors claims that enhanced the electron delocalization within the polymer which improve the PET from excited-state radical anion (NDI^{-•})* providing a strategy to circumvent the short lifetime of excited states. This strategy is interesting and I believe it would reach a broad audience in which ET processes from excited states are important and the work could contain the merits for publication in this journal.

However, several points should be addressed for a complete support of the conclusions and claims which I described in the next section.

I recommend publication of the manuscript after revision of the points described in the next sections.

Response: We really appreciated the reviewer's recognition for our effort to elucidate the relationship between dye aggregation and the electron transfer process including excited states. We have tried our best to revise and improve the manuscript to make things clear. The point-to-point response are listed below.

General comments:

The most obvious missing is poor mention of excited state lifetimes of the obtained photocatalyst in the discussion, perhaps the "mobility" or "the electron delocalization within framework" that the author mentions in the paper implies a "longer" excited state lifetime for the excited state of the catalyst. I believe the author should make an effort to make a better link between the lifetime in a single molecule and a "lifetime" in the heterogeneous photocatalyst. In relation to this, the mentions to the lifetimes of NDI^{-•}* must be revisited and should be included in the discussion of the results obtained. I believe citation 35 is not the best to compare with the actual system and the range 103-260 ps is more appropriate than "dozens of picoseconds". The authors should include a mention to this including citation from this works from Wasielewski et al 10.1021/jp000706f, Schanze et al 10.1021/acs.jpcc.9b06303 (perhaps together with 10.1039/C6FD00219F). (see specific comment for page 2)

Response: We highly appreciated the reviewer's constructive comments. The original mention of excited state lifetimes has been revised and we have tried our best to get the information of the "lifetime" for the Cd-SNDI and the ligand H₄SNDI (Fig. 5). The more suitable references suggested by the reviewer have been updated, and the related statements in manuscript have been rewritten.

To directly access the excited state lifetime of radical anion for coordination polymer Cd-SNDI and the ligand H₄SNDI, the femtosecond transient absorption (fs-TA) experiments were conducted for the prepared radical anion sample of Cd-SNDI and H₄SNDI. The obtained fs-TA spectra and the fittings of decays of the generated positive signals are displayed in Fig. 5 and discussed in the main text. The excited state absorption (ESA) band of radical anionic H₄SNDI is consistent with the literature value for (NDI⁻)^{*} (*J. Phys. Chem. A* **104**, 6545–6551 (2000); *J. Phys. Chem. B* **123**, 7731–7739 (2019); *J. Phys. Chem. C* **121**, 4558–4563 (2017)). The fitting of the decay for (NDI⁻)^{*} resulted in the time constants of 55 ps (33%) and 1.7 ns (67%). On the other hand, the TA spectra of excited-state radical anionic Cd-SNDI showed a red-shift band centred at 630 nm. Moreover, the decay spectra at 630 nm exhibited the time constants of 258 ps (35%) and 8.2 ns (65%), nearly five-fold of the H₄SNDI. From these results, we deduced that the specific arrangement of NDI in Cd-SNDI extended the lifetime of the highly-reducing species related to (NDI⁻)^{*}, which contributes to the superior photocatalytic performance of Cd-SNDI than the free ligand H₄SNDI.

The characterisation of the new Cd-SNDI catalyst and its synthetic advantages are well correlated in a Heterogeneous photocleavage of aryl halide. However, some control experiments are missing and must be performed for a complete mechanistic analysis.

The interaction with the substrate Ar-Cl is not analyzed in these sections which is surprising taking into account the observation made for the electron donors at the end of page 10. It is necessary to clarify the possibility of a pre-association of the polymer photocatalyst and the halogenated substrates performing (control) experiments similar to the ones performed with Et₃N (Fig 2) to clarify this situation which will be important for the proposed mechanism.

-Another important interaction is the effect of Ar-Cl in the properties of the radical anion in Cd-SNDI (in the dark) which should also be analyzed. See for instance: DOI:10.1002/cctc.202100359

Response: We are sorry for this oversight. The comparative study on the interactions between Ar-Cl substrate and neutral/radical anionic Cd-SNDI catalyst were performed and the results are shown as Fig. S15 in SI and Fig. 3c in the main text. As shown in Fig. 3c, the addition of substrate 4'-chloroacetophenone **1a** lead to the intensity decrease of the four fingerprint peaks of radical anionic Cd-SNDI (red arrows), suggesting the pre-association (via charge-transfer interaction) between the electron-rich radical anionic Cd-SNDI and the electron-deficient Ar-Cl. Furthermore, after 455 nm photoirradiation, the typical radical anionic peaks of this mixture fully diminished (blue arrows), demonstrating the regeneration of neutral Cd-SNDI by photoinduced electron transfer from excited-state (NDI⁻)^{*} of Cd-SNDI to substrate Ar-Cl. In comparison, the substrate 4'-chloroacetophenone **1a** can only induce tiny variations for the absorption of the DMF suspension of neutral Cd-SNDI both in dark or upon 455 nm photoirradiation (Fig. S15b), excluding the existence of strong interaction between the neutral Cd-SNDI and Ar-Cl substrate.

Several experimental details are missing to assure reproducibility of the result by the scientific community. Experiments of Fig 2, S4 and S5 should be clearly described in the SI with a clear indication of concentration (or amount of solid), the amount of Et₃N added, time of irradiation, irradiation source, etc.

Response: We thank the Reviewer for pointing this out. The specific details of original Fig. 2 and original Figs. S4 and S5 have been added into the section of “Experimental Details for

Figures in Manuscript” in Supplementary Information, and were also added into the Figure captions of Fig. S4 and Fig S5 of current edition.

- NDI core has characteristic absorption peaks 350-400, present in core-substituted and -unsubstituted NDIs. The author should measure all spectra starting at 300 nm to access these peaks. For instance, this will give additional information of the changes driven by the addition of Et₃N and by photostimulation.

Response: We thank the Reviewer for the good suggestion. The range of UV-vis absorption spectra of NDI-containing ligands/coordination polymers that related to Et₃N addition and photostimulation have been reset, such as Fig. 3b, Fig. 8b, Fig. S4, Fig. S5, Fig. S11, and Fig. S13.

- I strongly recommend putting a picture of the setups used for the synthetic experiments in the SI.

Response: Per your advice, the setup for the synthetic experiments has been added as Fig. S20 in SI.

Specific comments, changes and missing data:

Page 2

Line 3: I would suggest deleting “Exogenous...”

Line 10: I would suggest replacing "between" by "working with" or "when working with".

Line 11: “Scheme S1” must be changed by “Scheme S1a”

Response: Thanks for the reviewer’s detailed review. We have revised these points according to the suggestions.

Line 11-14: This frase in somewhat confusing, are the authors referring to the excited state E^o or the ground state E^o of the photocatalyst? In any case, the authors should include a citation to support this affirmation. Besides, it should be taking into account that the lifetime of excited states of perilene- and naphtalene diimides radical anions (PDI^{-*}, NDI^{-*}) don't show a clear correlation of E^o reduction potential of the excited state and its lifetime. Compare for instance data in the works of Wasielewski et al (10.1021/jp000706f, -.1021/acs.jpcc.9b06303) and Schanze et al (10.1021/jacs.9b13027).

Response: We thank the reviewer for this significant comment. The E^o here referred to the excited state E^o of radical anion, and the related descriptions have been made clear in the revised manuscript. We agreed that there was not a clear correlation between the lifetime and the reduction potentials of excited-state radical anionic aromatic diimides such as (NDI^{-*})^{*} and (PDI^{-*})^{*}. The related statements and the noted references were revised accordingly.

Line 15: “...the high enough electron delivery ability of excited-state dye...” It is not clear the meaning of this frase. It seems a rather complicated way to express the "kinetic" of the electron

transfer from the excited state. Perhaps the authors try to indicate the implication of different events or factors affecting the "kinetic" of the electron transfer but I believe the sentence is a little ambiguous and should be rewritten.

Response: We are sorry for the confusing phrases. What we were trying to express here by "kinetic" and related words was the effectiveness of photoinduced electron transfer (PET) from the excited-state radical anionic ($\text{dye}^{\cdot-}$)/(photosensitizer $^{\cdot-}$) to the substrate for triggering the reaction. The corresponding sections in manuscript have been rewritten.

Page 4

Line 14: "...branches," should be changed by "...branches (Fig. 1a),"

Line 21: Change "Scheme S1" by "see comparison in Scheme S1"

Response: Per your suggestion, we have revised these points.

Page 7

Line 7: EPR of free ligand is missing in Fig S6.

Response: The EPR of free ligand has been added to Fig S6.

Line 14-17: The absence of fluorescence is not a direct consequence of the short lifetime but the competitive fast non-radiative decay. I believe the phrase should be changed. I recommend deleting "Owing to the extremely short lifetime scale of dozens of picoseconds for excited-state ($\text{NDI}^{\cdot-}$),³⁵"

See discussion in general comments.

Response: Thanks for the reviewer's suggestion. We have revised the sentence to "Owing to the competitive nonradiative decay of ($\text{NDI}^{\cdot-}$),".

Line 20: Reference electrode should be included (I presume "vs SCE")

Response: We thank the Reviewer for noting this point. The reference electrode in our measurement is Ag/AgCl. To make comparison easier, we have re-estimated the potential values to vs. SCE.

Page 8

The experimental details for experiment of Fig 2d are missing and should be included in the SI. It is not clear to me how this experiment "demonstrated the superior electronic conductance of in situ generated excited-state ($\text{NDI}^{\cdot-}$) within Cd-SNDI...". The superior conductance of the ground $\text{NDI}^{\cdot-}$ could be enough if some interaction of Ar-Cl with $\text{NDI}^{\cdot-}$ allows its excitation "close" to a Ar-Cl molecule.

Response: We are sorry for the confusing made. Per your advice, the experimental details of original Fig. 2d (new Fig. 3f) were included in SI as "Supplementary Experimental Methods of Transient Photocurrent Responses". It was noteworthy that the typical Ar-Cl substrate 4'-chloroacetophenone **1a** was used as the electron sink for this experiment. In this case, we

inferred that the photocurrent was mainly contributed by the photo-induced electron transfer from the *in situ* generated radical anionic NDI^{•-} of Cd-SNDI/H₄SNDI to substrate **1a**. It was observed that cathode photocurrent responses of Cd-SNDI were much more prominent than those of H₄SNDI, demonstrating better electron communication between the (NDI^{•-})^{*} of Cd-SNDI and substrate **1a** than that of ligand. This was what we would like to express in the original statement of “demonstrated the superior electronic conductance of *in situ* generated excited-state (NDI^{•-})^{*} within Cd-SNDI”. We have rewritten this sentence to make things clear.

Page 9

Lines 9-10: Is “90 %” the conversion or the product yields? and more importantly are data in table product yield or conversion?

Response: We thank the Reviewer for pointing this out. As we calculated the ratio of the generated product to the internal standard via GC analysis, these numbers were all referred to product yields. We have revised the related expressions.

Table 1. The amount of electron donor looks excessive in table 1, more if we take into account others conPET catalysis (see for instance: 10.1002/cctc.202100359) and the results of table 2 in which 8 equiv. are enough to perform a more complex transformation in very similar conditions.

Being this an important point in having friendly conditions for synthesis, the author should perform the model reaction of table 1 with different equivalents of the best electron donor including a condition with 1 equiv. (or 1.1) and provide this information in the SI.

Response: We thank the Reviewer for this suggestion. Under the optimal reaction conditions of Ar-Cl **1a** (Table 1, entry 6), different equivalents of best electron donor were examined, and the results were listed in Table S3. When 8 equiv. of electron donor was used, a good yield of 80% still could be obtained after the extended reaction time of 1 day (Table S3, entry 6).

Page 10

Line 2: Citation 35 should be changed by more appropriate ones related to the lifetime including the works of Wasielewski et al (10.1021/jp000706f, 10.1021/acs.jpcc.9b06303) and Schanze et al (10.1021/jacs.9b13027) and relate to conPET reactions with NDI photocatalyst Bardagi et al. DOI: 10.1002/cctc.202100359

Response: Thanks a lot. We have updated the related references as Refs. 18, 41, 42, respectively.

Line 3-4: If TEMPO is acting only as radical scavenger conversion of this reaction should not be much affected and the adduct of TEMPO-(4-acetylphenyl) should be formed in appreciable amount. What is the conversion for this experiment?

Was the adduct of TEMPO-(4-acetylphenyl) identified?

Response: We appreciate this kind suggestion from the Reviewer. The addition of TEMPO can halt the reaction, the conversion was below 10%, and most of the substrate Ar-Cl was recovered.

The *in situ* generated aryl radical was trapped by TEMPO, and the formation of corresponding adduct TEMPO-(4-acetylphenyl) was confirmed by HRMS (Fig. 4f). Those results were consistent with trend of pioneering result (see ref 8: *Science* **346**, 725-728 (2014)).

Line 10-18: Although the better performance of Cd-SNDI compared to H₄SNDI is clear. The conversion of the C_{Ar}-I/Br/Cl for Cd-SNDI shows an inconsistency for conversion of C_{Ar}-I that should be in the order of C_{Ar}-Br/Cl but no lower. This difference could not be explained by measurement error because it is almost 20% lower than expected. Do the authors have an explanation for this observation?

Response: We are sorry that our original expression led to some misunderstanding and we have revised this to make things clearer. In the original edition of comparison experiments, considering the gradient reactivities of model substrates of C_{Ar}-I, C_{Ar}-Br, and C_{Ar}-Cl, the photocatalyst loading amounts and the reaction time lengths were set as 5% (1h), 5% (2h), and 10% (4h), respectively.

In the previous edition, the comparison of Cd-SNDI compared to H₄SNDI in the photocleavage of C_{Ar}-I/Br/Cl acted as the primary study before the optimization of reaction conditions, and a mild electron-donating reagent Et₃N was used in this comparative study but not the best electron donor dibutylamine. Thus, it was possible that for some specific combination of photocatalyst and C_{Ar}-I substrate, the reaction time length of 1h, that suitable for optimal electron sacrificialer dibutylamine, might not be enough for totally furnishing the photocleavage of C_{Ar}-I substrates. This might be the reason for the inconsistency mentioned by referee.

To avoid this confusing discussion and improve the readability, we revised the corresponding comparison experiments and directly use the final optimized reaction conditions with most effective electron donor dibutylamine (see Fig. 6, entries of **1a**, **1b**, **1c**), the photocatalytic C-X cleavage yields of 4'-chloroacetophenone (**1a**), 4'-bromoacetophenone (**1b**), and 4'-iodoacetophenone (**1c**) were 99%, 99%, and 90%, respectively, which could reflect the more inert nature of C_{Ar}-Cl and the higher reactivity of C_{Ar}-I (Fig. 4c and Fig. S27).

Page 15

Line 18: PXRD pattern of Sr-NDI is missing in Figure S14, so it is not possible to compare with pyrene@Sr-NDI.

Response: We are sorry for this oversight. The PXRD of Sr-NDI has been added to Fig. S14a.

Page 17

Line 14: change “discussion” by “conclusion”

Line 15: “...we modified...” should be changed for “...we incorporate...”

Response: Per your suggestion, we have revised these points.

Figures

Fig. 1:

- The phrase "SEt group was disordered in crystallography thus it is drawn on every possible position." must be added as indicated in Fig. S2.

- "c, S...S" must be changed by "c, S··S"

- I would suggest decorating the figure with some element that more clearly indicates the a b c d parts of the figure. For instance, a gray "water-mark" square in each section.

Response: We thank the Reviewer for noting these points. Fig. 1 has been revised according to these suggestions.

Fig 2:

Fig 2a: Is this a "solid-state" absorbance or a "Cd-SNDI suspension" as in Fig S4?

Irradiation time for Fig 2a,b should be indicated here or in a figure in the SI

Response: We thank the Reviewer for the kind suggestions and detailed review.

The comparative UV-vis spectra of original Fig. S4 were the normalized absorbances of H₄SNDI solution, Cd-SNDI suspension, and solid Cd-SNDI, respectively. And we have updated the sample preparation methods in the revised caption of Fig. S4.

Original Fig. 2a (new Fig. 3a) is the difference absorbance spectra of the DMF suspension of Cd-SNDI. The irradiation time and other details for original Figs. 2a, b (new Figs. 3a, b) has been added to "Experimental Details for Figures in Manuscript" of SI.

Fig 3:

-Fig 3b: "experimental" should be changed by "freshly prepared" "recovered" should be replaced by "recycled (3rd run)"

- The amount of catalyst should be added for experiments in Fig 3c.

Response: Per your suggestions, we have revised original Fig. 3b (new Fig. 4b) and the amount of catalyst and other details for original Fig. 3c (new Fig. 4c) has been added to "Experimental Details for Figures in Manuscript" of SI.

Fig S6: EPR of free ligand is missing

Response: We are sorry for this oversight. EPR spectra of H₄SNDI with different treatments have been added as Figs. S6a and S6c, and an anionic NDI⁻ signal of typical radical H₄SNDI could be found in Fig. S6c.

Citation:

- Please check Ref. 17, it looks inappropriate for the previous sentences.

Response: Thanks for the reviewer's detailed review. The charge transfer between triphenylphosphine and sodium iodide (original Ref. 17) is indeed not suitable for the description of charge transfer interaction between aggregated dyes. We have deleted this reference.

- Key reference is missing that are more appropriate that cite 35 related to NDI^{-•} lifetime and synthetic approach. See general comments and specific comments for pages 2 and 10.

D. Gosztola, wMark P. Niemczyk, W. Svec, A. S. Lukas, M. R. Wasielewski, Excited Doublet States of Electrochemically Generated Aromatic Imide and Diimide Radical Anions J. Phys. Chem. A 2000, 104, 6545–6551. DOI: 10.1021/jp000706f

S. Caby, L. M. Bouchet, J. E. Argüello, R. A. Rossi, J. I. Bardagi, Excitation of Radical Anions of Naphthalene Diimides in Consecutive- and Electro-Photocatalysis** ChemCatChem 2021, 13, 3001–3009. DOI: 10.1002/cctc.202100359

Response: Thanks for the reviewer's detailed review and kind suggestions. We have replaced original Ref. 35 with more suitable references (new Refs. 41 and 57).

- Ref. 46, points to a [(11)C]-cyanation and should be changed.

Response: Thanks for the reviewer's detailed review. The original Ref. 46 refers to a [(11)C]-cyanation meanwhile it also provides the chemical information of our synthesized drug molecule, perampanel, which contains a -CN group. We cite it to verify perampanel is an important drug molecule, so we think it is a suitable reference.

- Ref 47 and 55 are the same.

Response: Thanks for the reviewer's detailed review. We have modified this mistake.

Reviewer #3:

This is an interesting piece of work that is significantly enhanced by the C-X bond cleavage reactions and subsequent studies. The reactivity is certainly interesting and will provoke others to look into this idea. The improvement in the S-NDI systems is also intriguing and therefore I am convinced that this paper should be considered for publication in Nat Commun. However, there are a number of points that the authors need to address both to improve the presentation but also, most importantly, to clarify some scientific points. I am concerned by the PXRD of Cd-SNDI and the purity of the samples. I am also concerned that the authors are unclear in terms of describing how they think the coupling reactions are proceeding. This is a very important point and in its current form I am left with more questions than answers. Therefore, I suggest major revisions.

Response: We thank the Reviewer for the affirmation of our main idea and the kindness of providing the revision chance, because we have really put many efforts to investigate how to use supramolecular assembly as a specific tool to solve the issues in photocatalysis. And also, we regret that some poor presentations might lead to concerns. In the following part, we will respond to the comments one by one and make sure the manuscript would be more readable.

The authors use the phrase 'segregated aggregation'. I am not sure what they really mean by this. Do they just mean that the NDI components are spaced regularly? Aggregation usually refers to molecules that are in close proximity, in direct contact with each other. It would be helpful if they could be specific in their description of what they are trying to achieve.

Response: We appreciate this enlightening comment of the Reviewer. Sorry that some of our original expressions led to misunderstanding and we have revised them to make things clearer.

In this manuscript, the major design for the coordination polymer structure was inspired by "segregated aggregation/stacking", one of the famous D/A-typed supramolecular assembly modes, in which the two stacking columns of D and A not only parallelly arranged, but also were closely "segregated" with each other and associated by D...A charge-transfer interaction (Scheme S2a), which facilitated the photoinduced charge separations between proximal D and A, and the following delocalization of separated charges along the stacking columns of D and/or A, respectively, achieving the long-lived charge separation. Here, we borrowed this concept to illustrate how our heterogeneous system improved the PET efficacy of extremely short-lived excited states.

As depicted by Figs. 1d to 1g, the charge-transfer interactions and weak associations between the exogenous reagent (Et₃N)/substrate (Ar-Cl) and the NDI strings of Cd-SNDI might contribute to the *in situ* assembly between the NDI string and the exogenous reagent/reactant during each step of consecutive excitation, which was akin to the segregated stacking in a dynamic and nonclassical manner. Since there was no evidence of charge-transfer interactions or associations among Et₃N or Ar-Cl substrates, the "stacking column" of Et₃N or Ar-Cl should be a virtual one (Schemes S2d and S2e). The "segregation" here indicated the proximal separation between a pre-assembled and steadily existing NDI string of Cd-SNDI and an *in situ*-assembled and temporarily existing stacking column of Ar-Cl or Et₃N (Schemes S2b and S2c).

Although the segregated stacking we mentioned in our manuscript should be a nonclassical type, it would not matter much to the major designated target. The S...S contact between adjacent NDI can promote the transfer of excitons along the NDI string, and femtosecond transient absorption (fs-TA) study demonstrated that the lifetime of excited-state radical anionic

Cd-SNDI was almost 5 times more than that of the ligand H₄SNDI (Fig. 5). Thus, the lengthened excited-state lifetime of (NDI^{•-})*, the charge delocalization along the NDI string of Cd-SNDI, together with the pre-association of NDI^{•-} and Ar-Cl improved the PET from excited-state (NDI^{•-})* to inert substrate (Fig. 3c), possibly circumventing the long-pending issue of diffusion-limited electron transfer in solution phase.

Considering the specific meaning of “**aggregation**” as mentioned by the referee, we changed the “**segregated aggregation**” to “**segregated stacking**” and the related descriptions have been rewritten in the main text.

In Figure 1 it is unclear what the distance 3.44 refers to? I realise from the text that this refers to the S-S contact but this is not clear from the figure. Perhaps an inserted, enlarged figure, might help here as this is an important feature. The authors need to modify the figure caption to specify that the distances are shown in angstroms.

Response: Per your suggestion, the enlarged figure of S...S contact has been added as Fig. 1c and the figure caption has been modified.

The PXRD of Cd-SNDI shows a strong reflection at 12-13 degrees (2theta) which does not appear to be in the simulated pattern. What is this due to? Can the authors provide the PXRD pattern for a pure sample? This brings into question some of the results. If the sample isn't pure then what is leading to the effects observed?

Response: We thank the Reviewer for pointing this. We have enlarged the simulated pattern and there is the reflection at 12-13 degrees (Fig. S1a). The intensity difference of the peak ratios between simulated pattern and the prepared crystal sample is a common case for coordination polymer/MOF crystals and might due to the preferred orientation of the crystal (*Angew. Chem. Int. Ed.* **58**, 12175 (2019); *J. Am. Chem. Soc.* **140**, 2985-2994 (2018)). As shown in Fig. S1a, the PXRD peaks of our sample match well to the simulated one, indicating the good purity of Cd-SNDI sample.

However, the photoreductive C-Cl cleavage is impressive. I am genuinely interested in the authors opinions of how this process works. Do they think that the aryl-halide bond cleavage happens within the MOF or at the surface of the particles? DO they think that the charge is transferred to the crystal surface and then the reaction takes place there? I am sure it is tempting to think it happens within the framework pores but if so I assume the cleaved radical would have to migrate from the pores to react with the other substrate as it is difficult to see how both can fit within the pores, there is insufficient space in some examples, e.g compound 23. I note there is a lower yield in that case but I doubt that this is accounted for solely by encapsulation or surface phenomena. I appreciate that this is a very difficult thing to establish. However, the authors really should discuss this. If the authors could confirm that some of the substrates are too large to fit into the pores then this would suggest that the reactions are solution based.

Response: We thank the Reviewer for these instructive comments. As the Reviewer mentioned, it is difficult to claim the precise position for reaction to occur for our system based on the previous data. To get further insight, a substrate S-2 with bigger size than the substrate S-1 of compound 23 of Fig. 6 was prepared, and was examined in the photocleavage of Ar-Br and successive radical coupling in the presence of Cd-SNDI. Owing to the appending *n*-hexanyl

branch, the molecular size of **S-2** possessed an increased diameter of *ca.* 17.62 Å compared with that of *ca.* 11.25 Å for substrate **S-1**.

Since the Cd–SNDI only possessed the open channels with cross sections of $17.63 \times 10.11 \text{ \AA}^2$, the substrate ingress/egress experiments showed that **S-1** could be uptaken by Cd–SNDI but the bigger-sized **S-2** could not (Fig. S29 in SI). Not surprisingly, the corresponding photocatalytic reaction of increased-sized substrate gave a much lower yield (<10%) (**24**). These size-dependent experimental results can lead to the conjecture that the catalytic reaction mainly occurred within the pores or the channels of coordination polymer.

Reviewer #4:

Duan and co-workers reported a photoreducing coordination polymer featuring EtS-substituted NDI radical anions in this communication. Compared with the related coordination polymers that either lack the sulfur substituent or have a large separation between NDI units, the new heterogeneous photocatalyst displays superior SET reactivity, well demonstrated in various aryl-C/S/P/B bond-forming reactions. The authors suggested that the higher performance is caused by better electron conductivity/transportation between the NDI units, facilitated by chalcogen-chalcogen interactions, as the authors further proposed. While I found the experimental discovery timely and exciting, I do not feel comfortable recommending this work for publication in its current form. I would point out below some concerns for the authors to consider. Most issues are related to the claim of chalcogen-mediated phenomena, some are misconceptions, and others are language issues that may seriously affect the readers' understanding of this work.

Response: We are very pleased to receive the Reviewer's comment that this work is timely and exciting whereas feel regret for the issues or concerns regarding our previous edition. In the following part we will response to the comments one by one and improve the manuscript in the meantime.

(1) Chalcogen-chalcogen interactions were proposed to be important in the present system. I was, however, surprised to see no reference for the study of chalcogen-chalcogen interactions provided in this manuscript. I cannot see how Refs 18, 31, or 32 are relevant.

Response: We thank the Reviewer for the detailed review. The original Refs 18, 31, or 32 are all papers that focus on the special conductivity relating to charge transfer complex including S-containing molecule, tetrathiafulvalene. But these Refs indeed do not put emphasis on S...S interactions. We have replaced original Refs 18, 31, 32 with two Refs which verified that the distance of S...S contact plays a crucial role on the conductivity of material (*J. Am. Chem. Soc.* **137**, 1774-1777 (2015), Ref 35) as well as the formation of stable charge transfer pairs (*J. Am. Chem. Soc.* **141**, 17783-17795 (2019), Ref 36).

(2) In the context of "multiphoton consecutive excitation", Wenger's review in *Angew. Chem. Int. Ed.* 2020, 59, 10266 should be cited.

Response: We thank the Reviewer for noting the literature regarding to multiphoton excitation, which has now been cited as Ref. 10.

(3) It was stated "Purchasing the more elevated photoredox potentials through decoration of dyes has attracted extensive attention, which is at the expense of the more instability and even shorter lifetime of the corresponding highly reactive excited states...". This is a wrong statement. Photoredox driving force, chromophore stability, and excited-state lifetime do not necessarily show a positive or inverse correlation with one another. Another Wenger's paper doi.org/10.1002/anie.202110491 demonstrate this well.

Response: We thank the Reviewer for this significant suggestion. We are sorry for the wrong expression and we have deleted the mentioned sentence and rewritten the related parts.

(4) The quality of the X-ray crystal structure should be improved. The cif data of Cd–SNDI shows highly twisted and non-planar SNDI units. This is very questionable, besides the issue of serious disorders in other parts of the overall crystal structure. Furthermore, such non-planar SNDI was not seen in Figure 1a. Please clarify the inconsistency.

Response: We thank the Reviewer for this enlightening comment.

The distortion of aromatic rings of the NDI analogue in supramolecular structure is possible. For example, the Br-substituted NDI showed the twist of NDI core because of the interatomic steric crowding between Br atoms and the imide oxygen (*Org. Lett.* **9**, 3917–3920 (2007)). Also, the repulsion between an oxygen in NDI and sulfur/nitrogen atoms in thiazole can lead to a nearly perpendicular arrangement ($\sim 90^\circ$) between the thiazole and NDI units (*ACS Appl. Mater. Interfaces*, **10**, 40070–40077 (2018)). Besides, in coordination polymer/MOF Zr-PDI, the electrostatic repulsion and steric hindrance between the chloro substituents lead to the highly twisted central six-membered ring of P-2COOH with a dihedral angle of 38.3° (*Nat. Commun.* **10**, 767 (2019)). On the other side, it was reported that the aromatic diimide (NDI $^-$ or PDI $^-$) scaffolds with additional electrons tended to show bend configurations, as evidenced by single crystal structures (see ref 51: *Angew. Chem. Int. Ed.* **59**, 752–757 (2020)).

It was known that, the UV-vis spectrum of Cd–SNDI covering the visible light region of 400–800 nm showed the presence of abundant charge-transfer interactions within the ground-state framework (Fig. 2a), and the weak but still visible peak of radical anionic NDI $^-$ in the EPR spectrum verified the presence of partial negative charges in NDI units of pristine Cd–SNDI (Fig. 2f). Thus, the twist of NDI core in the crystal structure of Cd–SNDI might be correlated with the steric crowding of SEt group, S \cdots S interaction between neighbouring NDI unit, or the partially negative charged status of NDI moiety. Obviously, it was impractical to distinguish the detailed contributions of those possible factors.

Possibly due to the perpendicular viewing direction, the twist of the NDI core looks not that obvious in original Fig. 1a (new Figs. 1b and 1c), but the structural distortion would be more obvious if viewed from different directions (Fig. S2).

(5) In “... Cd–SNDI showed a wider absorption band that covered the visible light region (Fig. S4), indicating the presence of abundant charge-transfer interactions within the framework.” What is the origin of these CT interactions? How do the authors know that they are indeed CT interactions?

Response: We thank the Reviewer for pointing this out. It was well demonstrated that the incorporation of electron-donating group into NDI core would bring charge-transfer peaks in UV-vis spectra (*Org. Lett.* **14**, 4822–4825 (2012); *Nanoscale* **7**, 6729–6736 (2015)), which is the case for the broad band 450–600 nm for ligand H₄SNDI (Fig. 2a). As shown in Fig. 2a, UV-visible absorption of coordination polymer Cd–SNDI show the wider broad bands covering 400–800nm. Considering the stable d^{10} Cd $^{2+}$ and the absence of inter-ligand aromatic stackings, this broad band should be originated from S \cdots S linked NDI string with inter-ligand manner charge transfer. On the one hand, this kind of inter-ligand S \cdots S charge-transfer interaction allowed the accumulation of partial charge of NDI $^{\delta-}$. Thus, in comparison to ligand H₄SNDI, Cd–SNDI exhibited more remarkable signal in EPR spectra (Fig. S6). After shining light, the much more intensification of EPR signal could be observed for Cd–SNDI than that of H₄SNDI (Fig. S6), suggesting the S \cdots S interaction might also bridge the inter-ligand delocalization of photogenerated charge pairs under excited state.

(6) Interestingly, only Cd–SNDI and H4SNDI could form charge-transfer complexes with Et₃N, whereas other M–NDI samples display no changes in the UV-Vis absorption in the presence of Et₃N. Since the electron deficiency (based on the reduction potential) is similar across the board, was the CT interaction caused by chalcogen bonding between Et₃N:…SEtNDI? If so, would the inclusion of Et₃N in the structure of Cd–SNDI change the geometric relationship between the NDI units?

Response: We thank the Reviewer for this helpful comment. The charge-transfer complexes formation between Et₃N and Cd–SNDI or H₄SNDI was indeed possibly mediated by chalcogen bonding between Et₃N:…SEt–NDI considering the absence of charge-transfer interaction between Et₃N and other NDI coordination polymers such Mg–NDI and the well-established S…N interaction between n-typed electron-withdrawing S moiety and n-typed electron-donating N group (ref 53: *Chem. Mater.* **21**, 2149–2157 (2009); ref 54: *Chem. Pharm. Bull.* **56**, 802–806 (2008)). To deeply investigate the structural characteristics of this proposed Et₃N:…SEt^{δ+} interaction, we tried to examine the X-ray single crystal analysis on Cd–SNDI crystal that encapsulated Et₃N. However, the not steady enough “anchoring” of Et₃N in Cd–SNDI made this attempt not successful.

To explore whether the inclusion of Et₃N by Cd–SNDI change the geometric relationship between the NDI units, the UV-vis absorption and the PXRD of Cd–SNDI in response to Et₃N were analysed. As shown in Fig. 3a and Fig. S11, upon the addition of Et₃N, although the new peaks emerged to show the partial electron transfer from Et₃N to Cd–SNDI, the “background”-like broad and profound charge-resonance bands covering 400~800nm did not change much. This result implied that, as the key player mediating inter-ligand charge delocalization, S…S interaction between neighbouring NDI–SEt moieties did not change much. Moreover, after photocatalysis involving electron-donating amine-typed reagent like Et₃N, Cd–SNDI was recovered to examine the XRD, and the maintained patterns of XRD before and after photocatalysis (Fig. 4b) disclosed the maintenance of framework structure of Cd–SNDI and basic geometric relationship between NDI moieties therein.

(7) The authors proposed electron delocalization between NDI in Cd–SNDI. This proposal is one of the central ideas but was not supported by experimental evidence. Judging from the UV-Vis absorption of NDI radical anion in Cd–SNDI, the radical is very likely centered on one single NDI. Or, is there a charge-resonance band to support this proposal? If, however, the authors are considering charge transport over several NDI sites on a long timescale (as opposed to “delocalization”), they may consider using EPR to probe the existence of such a phenomenon (or the absence of it).

Response: We thank the Reviewer for this enlightening comment.

The charge-resonance bands could be observed in UV-vis absorption spectra to clarify the intra-ligand and inter-ligand charge delocalization. In the case of ligand H₄SNDI, the direct connection of n-typed electron-donating SEt group to π-typed electron-withdrawing NDI core allowed the intra-ligand charge resonance (n→π*), as exhibited by the broad band 450~600nm (Fig. 2b). After assembling the ligand into coordination polymer Cd–SNDI, the S…S interaction provided the possibility of mediating inter-ligand charge transfer. As shown in Fig. 2a, the bands that covering 400~800nm, which is much broader than that of the ligand, suggesting the S…S mediated inter-ligand charge transfer (as verified by EPR analysis shown in the Fig. S6).

To make the radical anion peaks of Cd–SNDI more identifiable, we presented the difference spectra in Fig. 3a to eliminate the disturbance of “background”-like broad charge-resonance

bands, which might give a misleading impression that the “additional electron” of radical anion was solely localized in one NDI moiety without charge delocalization.

Moreover, we would like to examine the possibility of charge delocalization of radical anionic Cd–SNDI, but the dearth of net charge changes made this experiment impractical to be monitored by EPR. Importantly, the femtosecond transient absorption (fs-TA) study demonstrated that the lifetime of excited-state radical anionic Cd–SNDI was almost 5 times more than that of the ligand H₄SNDI (Fig. 5), which suggested the “delocalized” features of the radical anionic NDI string after excitation, implying the across-ligand charge migration in Cd–SNDI within a short timescale.

(8) The authors used impedance measurements to probe the relative electric conductivity of various samples. How much of the measured impedance is contributed by the inter-NDI charge transport, and how much by other processes (e.g. interfacial electron transfer)? What is the purpose of Nafion? Does Nafion affect the intrinsic conductivity?

Response: We thank the Reviewer for those important comments. The electrochemical impedance spectra (EIS) were widely used in qualitatively illustrating the alternating current (AC) electric field-induced polarizability of solid materials, which was also employed to probe and qualitatively compare the intra-framework charge transfer performances in coordination polymers/MOFs and related composite materials (*Nat. Commun.* **11**, 5384 (2020); *Chem. Eur. J.* **21**, 2364 (2015); *ACS Appl. Energy Mater.* **4**, 4319–4326 (2021); *CrystEngComm*, **23**, 7496–7501 (2021)). EIS was qualitative but not quantitative analysis, and it was difficult to quantitatively assign the EIS semi-circle curve of high-medium frequency zone into the portions of induced charges generated *via* the manners of intra-ligand, across-ligand, and across-densely packed interfaces.

The generally poor conductivity and hydrophobic nature of coordination polymers/MOFs set up a challenge for their electronic communications with electrodes/electrolytes in electrochemical experiments such as EIS. Therefore, Nafion was utilized to act as the dispersant reagent and the adhesive reagent to twin and envelope the micro-particles of finely ground coordination polymers/MOFs. In this way, an amphiphilic conductive layer between the hydrophobic coordination polymers/MOF phase and the hydrophilic electrolyte phase was constructed to provide a uniform, stable, and conductive carrier environment for probing the charge-transfer performances of coordination polymers/MOFs. When preparing the sample for EIS examinations, the coordination polymer, Nafion, and other additives were mixed in the same ratios and finely ground before being mounted to electrode surface, which was described in “Materials and measurements” of main text. Thus, the Nafion in the different samples of coordination polymers contributed to a uniform “background” level for EIS spectra in the qualitative and comparative study such as Figs. 3e and 8d.

(9) The results of photocatalytic cleavage of 4'-haloacetophenone by Cd–SNDI seem to be inconsistent between those shown in Figure 3c (Cl>Br>I) and in Figure 4 (Cl (90% 4h)I believe the authors have addressed all the comments/mistakes/suggestions I had pointed out in the first revision. However, in virtue of the new femtosecond (fs-TA) transient absorption experiment provided, I have some comments and suggestions/corrections that I described in the next section.

I recommend publication of the manuscript after minor revision.

Comments:

The femtosecond transient absorption (fs-TA) experiments performed are really important in the analysis of the results and the ps times obtained are consistent with others radical anions lifetimes. However, the analysis performed over this data is somewhat poor and also the experimental details need to be better described. For instance, there is an attempt of association of evolution-associated absorption spectra with transients, which is important to link the different τ with an electronic state (Dn). (A recent excellent example could be seen in J. Beckwith , A. Aster, and E. Vauthey, he Excited-State Dynamics of the Radical Anions of Cyanoanthracenes. 10.1039/D1CP04014F)

According to the method of preparation ("excess amounts of degassed Et₃N and successive photoirradiation from a 455 nm LED for 5 min") it is not clear if some of the neutral H₄SNDI is present in the experiment and it is expected that the sample of Cd-SNDI neutral and radical anionic species will be present. If both species are present, this will implied a simultaneous excitation of both species and will also implied a mixed absorption spectra of NDI* and NDI^{•-}* , if the authors consider the absorption of the neutral molecule is negligible this should be mentioned in the detail in the SI. To be more clear this question need to be answer: Are the ns times decay due to excitation of NDI or NDI^{•-}? A mix of neutral and radical ionic Cd-SNDI may solve the issue due to the proximity of the

times obtained by Fluorescent and fs-TA (6.7 ns vs 8.2 ns) but it will not be the case for H4SNDI (27.1 ns vs 1.7 ns). If this trace is associated to NDI^{•-} a better explanation should be done instead of "related species related to a single ligand"

Why is the analysis of the spectra done only in the region 500-750 nm? Between 400 and 500 nm there is a very important signal of the NDI^{•-}, which could give information about the process.

In addition, Figure S19 has information that is not analyzed neither in the main manuscript nor in the SI. What is the meaning of the different exponential fitting at different wavelength

Considering the extension of this detailed analysis I did not pretend to be done in the main manuscript but due to its importance I believe these issues should be addressed in a more detailed description and analysis in the SI.

Specific comments, changes and missing data:

The amount of Et3N for the fs-TA in Fig S19 should be specified.

Reviewer #3:

Remarks to the Author:

The authors have prepared a much improved revised article and in my opinion I am satisfied that all of the reviewers comments have been addressed. I am, therefore, happy to recommend acceptance of this revised version.

Reviewer #4:

Remarks to the Author:

The new experimental results added to the revised manuscript by Duan and co-workers have indeed strengthened several claims and arguments that the authors would like to make. In particular, I appreciate seeing (i) the fsTA measurements, demonstrating the longer lifetime of the excited anion in Cd-SNDI, (ii) the reduced NDI^{•-} absorption signals of Cd-SNDI upon addition of Ar-Cl, suggesting pre-association/electron transfer in the dark, (iii) radical trapping with TEMPO, providing the evidence of the intermediate, and (iv) the reactant-size dependent photoreduction, suggesting the electron transfer events occurring within the porous structure of Cd-SNDI MOF. Despite the excellent results reported here, I still find a few issues that need to be addressed rigorously. Most of them are not about the experimental results, which are done very well, but related to the interpretation and discussion.

(1) The authors still present the idea of molecular segregation in a bizarre way. It feels like that they genuinely love this idea, so much so they want to relate the results to it regardless if it is appropriate. From the revised manuscript, it became clear now that the authors think that the SNDI units in Cd-SNDI stack into one column and amines or Ar-Cl "VIRTUALLY stack in situ" into another column; thus, two clusters of molecular units segregate. Two problems are associated with this proposition: (i) First, the authors do not have any structural proof of the location of amines and Ar-Cl. The changes in optical spectra do not imply their whereabouts. They may present/distribute in the MOF porous scaffold in a random and scattered manner (although being close to SNDI). The optical response also does not imply the relative arrangement of amines and Ar-Cl molecules. This proposition is doubtful, especially considering that not all SNDI units have become SNDI^{•-} (or excited SNDI^{•-}) and that the association of amines or Ar-Cl with SNDI^{•-} may be transient and dynamic. (ii) Second, if this idea of

molecular segregation can not be substantiated, the materials design principle presented in this work reads weak and handwavy. In fact, for the ideas presented in Figure 1 or Scheme S2 to work, the amines and Ar-Cl molecules DO NOT have to assemble into columnar aggregates. In my opinion, this is really an unnecessary and unverified suggestion and would only make readers confused about the possible reaction mechanism.

(2) The quality of the Cd-SNDI crystal structure must be improved, as pointed out in the previous review. It is certainly common to observe non-planar polyaromatic hydrocarbons; all of these compounds have a smooth and curved structure, much like a twisted ribbon. However, the Cd-SNDI structure presented in the cif file is not sensible. The issue is not about being planar or not; the atomic arrangement of Cd-SNDI is simply unreasonable. Let me give one example to illustrate the problem: the improper dihedral angle about the NDI aromatic carbon atom attached with sulfur is 25–45° (depending on the sequence of atoms picked). This value seriously deviates from 0°, and thus this carbon is not an aromatic sp² carbon according to this cif file (for reference, the improper dihedral angle about the sp³ carbon in methane is ~35°). The quality of the data might be poor (high Rint) to start with, and/or the naphthalene diimide core units should be considered to be disordered in addition to the EtS part.

(3) After sorting out the quality of the Cd-SNDI crystal structure, the authors could have better confidence in estimating the S...S distance. Concerning the distance measurement, the authors might consider the average of the two red distances in the figure below – if the improved/refined structure remained disordered.

(4) Following (3), I would suggest the authors be cautious when describing the MOF structure using the term S...S or chalcogen-mediated interaction. In supramolecular chemistry as well as several statements in this manuscript, "S...S interaction" implies attractive stabilization energy between the two sulfur atoms. It is unlikely to be the case here. The proximity of sulfur orbitals may facilitate charge transport, but it does not necessarily imply stabilization and hence may not be the driver of the MOF structure. The positioning of atoms in the solid state (crystal) is often the consequence of many restraints imposed by other atoms; seeing any two units in proximity does not always imply interactions (attractive or repulsive alike). Again, if the authors firmly believe that there is "S...S interaction" driving the formation of the MOF structure, either a better proof or a literature reference should be provided, as pointed out in the previous review. (Ref 35 is relevant to the current study indeed, but it does NOT suggest that MOF structure therein is driven by chalcogen-mediated interaction.)

(5) On page 9, it was suggested that the broad absorption at ~780 nm of the sample containing Cd-SNDI and Et₃N is a D₀ → D₁ transition (Fig 3a). Please clarify the molecular origin of the doublet states (Whose doublet? It is definitely not SNDI•–'s, as the blue trace in Fig 3a is different from the red one.). It might be informative to also comment on the fact that such 780-nm absorption was not observed for H₄SNDI+Et₃N (Figure S5).

(6) The estimation of the photoreduction power of ScNDI and pyrene@ScNDI anion radicals is not reasonable. On pages 23-24 "If based upon the doublet-state D₀ → D₁ transition peak at 495 nm, the maximum photoreducing powers ...-1.94 V": (i) 495 nm excitation is not the D₀ → D₁ transition; it leads to higher doublet states. (ii) It is not obvious why the authors wanted to use the 495-nm transition for estimation instead of the lowest-energy band at ~780 nm. (iii) if an arbitrary absorption band and thus an arbitrary excited state is considered, -1.94 V should not be the "maximum"; any blue-light excitation will give even higher energy. The same problem appears for the pyrene@ScNDI sample. Additionally, the word "redshifted" may be misleading, as the 495-nm excited state of ScNDI anion and the 505-nm excited state of pyrene@ScNDI anion are likely not the same excited state.

(7) As indicated in the previous review, the authors should clearly distinguish the idea of charge

delocalization and transport. There is little evidence that the NDI anion is delocalized, given the observation of the characteristic peaks of radical anion isolated/monomeric NDI (Fig 3a, 3b, 3c). On the other hand, if there is some sort of rapid transport within the EPR timescale, the EPR line width may be reduced. However, it is not apparent based on the data presented.

(8) A few experimental details need to be explained/justified more clearly. First, for the fsTA experiment, if the radical anion species are generated by photoreduction (using Et₃N as the electron donor), are all the S₄NDI units reduced? If this is not the case, the concentration of S₄NDI^{•-} anion may change during the course experiment, thus resulting in different concentrations of [S₄NDI^{•-} anion] and excited ^{*}[S₄NDI^{•-} anion] at time = 0 of every 480 nm laser pulse, and therefore influencing the accuracy/credibility of kinetic analysis.

(9) Second, it is sometimes unclear if the suspension (sonicated?) or the supernatant (after sonication) samples were used in the measurements (absorption, emission, and TA) and synthetic photochemical experiments. This should be clarified. For those using supernatant solutions, how do the authors confirm that the solution contains the pristine MOF samples instead of the structurally decomposed soluble fragments?

(10) Third, the method for creating the difference spectra (e.g. Fig 3c) or normalized spectra (e.g. Fig 2c) should be clarified. For instance, it is not obvious what was normalized against in Fig 2c for the blue and red traces. In Fig 3c, what is the spectrum used for subtraction? If it were the neutral Cd-S₄NDI spectrum used for subtraction, why does the red trace in Fig 3a look different from the black trace in Fig 3c? Relatedly, it is unclear why Figure S17 was not presented as difference spectra (cf. Figures S15 or S16).

(11) While I believe that H₄S₄NDI has been synthesized and very likely constitutes the ligand of Cd-S₄NDI, the ¹H and ¹³C NMR data and spectra of H₄S₄NDI are questionable. Numerous thiolated NDI derivatives have been reported in the literature; they all display “normal” spectra. The absence of EtS peaks and unusual chemical shifts cannot be attributed to the polar bonding between C-S (if so, the same phenomena would have been observed for other thiolated NDI). Please verify the structure/structural data. On a related note, the elemental analysis results of Cd-S₄NDI deviate noticeably from the expectation.

(12) Nafion is strongly acidic and thus may affect the structure or structural integrity of MOF samples examined in this work. The use of it raises concerns about the impedance data.

(13) Regarding “Considering the well-described charge delocalization ability of S^{•-}⋯S mediated neutral or negatively charged NDI string,....” (page 11): (i) the statement “charge delocalization ...neutral NDI string...” does not make sense; and (ii) please provide references for the “well-described” charge delocalization.

REVIEWER COMMENTS

Reviewer #1 (Remarks to the Author):

Overall, I am convinced that the revised version of this manuscript and the response to reviewers that the authors have provided numerous interesting observations and a roadmap to expanding the applicability of radical anion photocatalysts. While this manuscript still does not offer a specific immediate synthetic advantage relative to prior work in this area, I am inclined to recommend this manuscript should be accepted.

Response 1: We highly appreciated the reviewer for the acceptance of our manuscript. Thank you very much for your great effort in improving this paper.

Reviewer #2 (Remarks to the Author):

Summary

The manuscript describe the synthesis of the a coordination polymer using naphthalene diimide (NDI) substituted with S \cdots S and Cd(II), achieving an S \cdots S interaction-mediated segregated stacking of dyes, which facilitated the electron mobility throughout segregated NDI-strings while simultaneously maintaining the reduced power of the ground state and the excited-state reducing power of its reduced form. This particular characteristic allows the authors to demonstrate the superior behavior of this heterogeneous photocatalyst in compare to single-molecule dye photocatalysis and other coordination polymers in a consecutive photocatalysis (conPET) for the photoreduction of aryl halides including aryl chlorides with EWG. The photocatalytic system also demonstrates good performance in successive radical couplings allowing the formation of C-C, C-S, C-P, and C-B bonds by known procedures.

The authors claims that enhanced the electron delocalization within the polymer which improve the PET from excited-state radical anion (NDI \cdot^-)* providing a strategy to circumvent the short lifetime of excited states. This strategy is interesting and I believe it would reach a broad audience in which ET processes from excited states are important and the work could contain the merits for publication in this journal.

I believe the authors have addressed all the comments/mistakes/suggestions I had pointed out in the first revision. However, in virtue of the new femtosecond (fs-TA) transient absorption experiment provided, I have some comments and suggestions/corrections that I described in the next section.

I recommend publication of the manuscript after minor revision.

Comments:

The femtosecond transient absorption (fs-TA) experiments performed are really important in the analysis of the results and the ps times obtained are consistent with others radical anions lifetimes. However, the analysis performed over this data is somewhat poor and also the experimental details need to be better described. For instance, there is an attempt of association of evolution-associated absorption spectra with transients, which is important to link the different τ with an electronic state (D_n). (A recent excellent example could be seen in J. Beckwith , A. Aster, and E. Vauthey, he Excited-State Dynamics of the Radical Anions of Cyanoanthracenes. 10.1039/D1CP04014F)

Response 2-1: We highly appreciated the reviewer for the positive comments and the helpful suggestions regarding to the interpretation of fs-TA results.

As suggested by **Reviewer #2**, the curves of fs-TA spectra were displayed at the indicated delay times to associate the possible evolution of species. Regarding the analyses of the transient spectroscopic data, we had to admit that it needed the expert-level skills of photochemist/photophysicist to give the precise and exclusive attribution of fs-TA spectra in very detail, which was beyond the bounds of our ability and the main target of this work. As synthetic chemists in the area of coordination polymer and photocatalytic organic synthesis, it was our aim in the manuscript to look for the reasons why the supramolecular assembly of ligands within coordination polymer could circumvent the diffusion-limited PET issue of excited-state radical ionic dye in the solution phase. Enlightened by **Reviewer #2**, we still tried our best to comprehend the fs-TA results in a comparative study manner, focusing on the variances between coordination polymer and ligand in the processes of excitation, charge separation, and charge recombination. It was hoped that the results and discussions here could primarily give some clues on how the “supramolecularity” of coordination polymer-based photocatalysis made it distinct from the homogeneous photocatalysis in solution.

Owing to the highly complex nature of fs-TA spectra and the tight correlations among the peaks, decay/recovery lifetime, and other information of various NDI samples that measured at different excitation wavelengths, herein in **Response 2-1**, it was quite necessary to analyze those data comprehensively in a comparative manner, which was helpful to decrease the misleading based upon the fragmented information. Regarding the referee’s subsequent comments in a certain respect of fs-TA information, we would respond accordingly and emphasize some specified points, trying to avoid unnecessary repetition.

Considering the important role of the peaks in the range of 350 to 500 nm for the assignment of excited-state species, we tested the fs-TA spectra of the radical anion of coordination polymer Cd–**SNDI** and ligand H₄**SNDI** under the excitation of 630 nm (**Fig. S20**). This wavelength can excite the characteristic peaks of NDI^{•-} of radical anionic ligand or coordination polymer, and also avoid the excitation of neutral species. After the 630 nm laser excitation of radical anionic samples for 1 ps, the excited state absorption (ESA) bands and the ground state bleach (GSB) bands could be clearly observed as black lines. For radical anionic Cd–**SNDI** (**Fig. S20c**), the GSB peak was observed at ~375 nm, and the ESA band covered a broad arrange from 400 nm to 550 nm ($\tau = 164$ ps, **Fig. S20d**) of which the decay lifetime was close to the reported data of (NDI^{•-})^{*} (such as $\tau = 142$ ps, see **Ref. 45: J. Phys. Chem. B 2019, 123, 7731–7739**). In comparison, in the case of radical anionic ligand H₄**SNDI** (**Fig. S20a**), the GSB band was blue-shifted to less than 350 nm, and the ESA band was also blue-shifted and exhibited a narrower shape, of which the decay lifetime 93 ps was much shorter than that of radical anionic Cd–**SNDI** (**Fig. S20b**), but still not far from some literature result (such as $\tau = 112$ ps, see **Ref. 67: J. Phys. Chem. C 2017, 121, 4558–4563**). For both cases, the possibly existing GSB bands of NDI^{•-} moieties at ~480 nm might be compensated by the intense ESA bands. From 80 ps after irradiation on radical anionic Cd–**SNDI**, a new bleach at ~425 nm appeared (**Fig. S20c**), which might be assigned to the vanished intra-ligand charge-transfer band (see the CT band of ligand at ~425 nm, **Figs. 2a, 3b**). In comparison, the bleaching of the intra-ligand charge-transfer band was not observed in the case of radical anionic H₄**SNDI** (**Fig. S20a**). The vanishment of the intra-ligand charge-transfer band within Cd–**SNDI** implied the possible tendency of inter-ligand electronic communication.

The fs-TA spectra of radical anionic coordination polymer and ligand were also examined under the excitation of a 480 nm pump laser (**Figs. 5c, 5d, S19**). ~5 ps after irradiation, an ESA band covering 500–650 nm centred at 560 nm was observed in the case of radical anionic H₄**SNDI** (**Fig. S19a**), and the radical anionic Cd–**SNDI** exhibited several merged positive bands that covered a range from 550–750 nm (**Fig. S19c**). The positive peaks at 540–560 nm resembling the typical two-centre three-electron [S : π]⁺ radical cationic species were found for both cases (**Refs. 63-66: Org. Lett. 2013, 15, 4932; J. Phys. Chem. A 2014, 118, 4451–4463; Photochem. Photobiol. Sci., 2008, 7, 1407–1414; J. Org. Chem. 2006, 71, 3, 853–860**), implying the intra-ligand charge transfer between SEt branch and aromatic NDI nucleus. It was noteworthy that this positive band centred at 560 nm decayed much faster in the case of radical anionic Cd–**SNDI** (**Fig. S19d**) than that of radical anionic H₄**SNDI** (**Fig. S19b**), further implying the possible presence of inter-ligand charge transfer in coordination polymer as a competitive route of the intra-ligand charge transfer. Moreover, the remarkable and long-lived bleaching peak at ~520 nm of radical anionic Cd–**SNDI** might be assigned to the vanished intra-ligand charge-transfer band

(see the ~520 nm peak of broad intra-ligand CT-band, Figs. 2a, 3b), of which the ns-scaled recovery lifetime of 7.4 ns (Fig. S19e) was comparable to the 8.2 ns decay lifetime of ~630 nm summit (Fig. S19e). The correlation between this pair of GSB/ESA bands suggested that the ~630 nm positive peak with a long decay lifetime might be related to the inter-ligand charge transfer in coordination polymer. To the contrary, either the bleaching peak at ~520 nm or the positive one at ~630 nm was not found in the case of radical anionic H₄SNDI (Fig. S19a), not supporting the profound inter-ligand charge transfer among radical anionic ligands in solution.

According to the method of preparation (“excess amounts of degassed Et₃N and successive photoirradiation from a 455 nm LED for 5 min”) it is not clear if some of the neutral H₄SNDI is present in the experiment and it is expected that the sample of Cd–SNDI neutral and radical anionic species will be present. If both species are present, this will implied a simultaneous excitation of both species and will also implied a mixed absorption spectra of NDI* and NDI^{•-}*, if the authors consider the absorption of the neutral molecule is negligible this should be mentioned in the detail in the SI.

Response 2-2: we thank the reviewer for the enlightening comments.

After proceeding the preparation method as described, the NDI⁻ was the dominative species in ligand and coordination polymer, and the presence of neutral NDI was negligible, which can be verified by the UV-vis absorption spectra shown in Figs. 3a, 3b. To further clarify the fs-TA characteristics of radical anionic samples in a comparative manner, we primarily examined the fs-TA spectra of neutral ligand H₄SNDI and coordination polymer Cd–SNDI to pick up some useful clues, although the neutral NDI species were not the major players in this story.

After irradiating neutral H₄SNDI from 480 nm laser for ~5 ps (Fig. 5e), the discrete or merged doublet-state peaks of NDI⁻ could be detected at the locations of 480~500 nm, 630~650 nm, and 680~750 nm, and the positive band at 540~560 nm resembled the typical two-centre three-electron [S:π]⁺ radical cationic species (Refs. 63-66: *Org. Lett.* **2013**, *15*, 4932; *J. Phys. Chem. A* **2014**, *118*, 4451–4463; *Photochem. Photobiol. Sci.*, **2008**, *7*, 1407–1414; *J. Org. Chem.* **2006**, *71*, *3*, 853–860). The *in situ*-generated radical ionic pairs indicated the strong tendency of photoinduced charge separation of H₄SNDI, and similar pieces of evidence were also revealed in the case of neutral Cd–SNDI (Fig. 5f). Moreover, the curve section that bleached at 520 nm (Figs. 5e, 5f) was possibly assigned to the vanished intra-ligand charge transfer (see the CT band centred at ~520 nm, Fig. 2a), which might be contributed by the competition from inter-ligand electronic communications.

The obvious differences of fs-TA spectra between neutral (Figs. 5e, 5f) and radical anion samples (Figs. 5c, 5d) for both ligand H₄SNDI and coordination polymer Cd–SNDI excluded the remarkable presence of neutral NDI species in radical anionic samples of H₄SNDI and Cd–SNDI.

However, it was still interesting to find the clues of inter-ligand charge transfer either in neutral or radical anionic Cd–SNDI, which was important to understand the role of S··S contact in mediating inter-ligand electronic communications, and further interpret the structure-activity relationships regarding the enhanced accessibility of (NDI⁻)^{*} towards Ar–X substrate in radical anionic Cd–SNDI.

Per the request of referee, the relevant information was added into the **Supplementary Experimental Methods and Analyses of Transient Absorption Spectra, Supplementary Information**.

To be more clear this question need to be answer: Are the ns times decay due to excitation of NDI or NDI⁻? A mix of neutral and radical ionic Cd-SNDI may solve the issue due to the proximity of the times obtained by Fluorescent and fs-TA (6.7 ns vs 8.2 ns) but it will not be the case for H₄SNDI (27.1 ns vs 1.7 ns). If this trace is associated to NDI⁻* a better explanation should be done instead of “related species related to a single ligand”

Response 2-3: We thank the reviewer for this instructive comment regarding the possible assignment of ns-scale species in fs-TA spectra.

As analyzed in Response 2-1 and Response 2-2, under the excitation of 480 nm, the radical anionic H₄SNDI exhibited an ESA band centred at 560 nm (Fig. S19b), of which the ns-scale decay lifetime of 1.7 ns implied the possible occurrence of intra-ligand charge transfer after excitation.

As analyzed in Response 2-1 and Response 2-2, the ~630 nm positive peak with a long decay lifetime of 8.2 ns (Fig. S19f) might be related to the inter-ligand charge transfer in radical anionic Cd-SNDI.

Those information had been added to main text, and further bolder deductions based upon the experimental results and the reported precedents were listed in Supplementary Experimental Methods and Analyses of Transient Absorption Spectra, Supplementary Information.

Why is the analysis of the spectra done only in the region 500-750 nm? Between 400 and 500 nm there is a very important signal of the NDI^{•-}, which could give information about the process.

Response 2-4: We thank the reviewer for this helpful suggestion.

In our previous fs-TA examination, 480 nm was chosen as the wavelength of the laser to excite the $D_0 \rightarrow D_n$ peak of doublet-state NDI^{•-}, and it was also close to the frequently-used 455 nm wavelength of LED in the practical multiphoton excitation.

However, the 480 nm laser excitation severely interfered the curves of < 500 nm lengths. Considering the important role of the peaks in the range of 400 to 500 nm for the assignment of species, we retested the fs-TA spectra of the radical anion of coordination polymer and ligand with the excitation of 630 nm (Figs. 5a, 5b, S20). This wavelength can excite the characteristic peaks of NDI^{•-} of radical anionic H₄SNDI or radical anionic Cd-SNDI (Figs. 3a, 3b), also avoid the excitation of neutral NDI species.

The corresponding analyses were shown in Response 2-1, please refer to this item.

In addition, Figure S19 has information that is not analyzed neither in the main manuscript nor in the SI. What is the meaning of the different exponential fitting at different wavelength.

Response 2-5: We appreciated the reviewer for the kind reminder.

Considering the highly complex and dynamic nature of fs-TA, we had to admit that it might need the expert-level skills of photochemist/photophysicist to give a precise and exclusive explanation to some specific exponential fitting lifetime τ (maybe also applicable to the case of Fig. S19). Although the precise and exclusive attribution of fs-TA spectra in very detail was beyond the bounds of our ability and the main target of this work, it was believed that sharing the experimental results with somewhat redundancy data in Supplementary Information (such as Fig. S19 and other similar cases) might be helpful to trigger the broad interests of photochemist/photophysicist and other experts.

Thus, to supply more information than the original data of Figs. 5 and S19 (radical anionic H₄SNDI and radical anionic Cd-SNDI, laser excitation at 480 nm), further comparative studies were carried out, as shown in Fig. S20 (using a different laser wavelength of 630 nm, radical anionic ligand and radical anionic coordination polymer) and Figs. 5e, 5f (neutral H₄SNDI and neutral Cd-SNDI, under 480 nm laser excitation). And those comparative studies were comprehensively discussed in Response 2-1, Response 2-2, the main text, and Supplementary Experimental Methods and Analyses of Transient Absorption Spectra, Supplementary Information, in which we still made inferences about some representative bands/exponential fitting lifetimes.

To avoid the possible repetition and the one-sided description of specific data, please refer to those previous responses and mentioned sections for more comprehensive interpretations.

On the other hand, without the reliance on expert-level resolutions of fs-TA in every detail, the controlled experiment-like comparative study on fs-TA spectra still disclosed the variances between coordination polymer and ligand in the processes of excitation, charge separation, and charge recombination. Those differences were tightly correlated with the supramolecular assembly manner of ligands within coordination polymer, disclosing the structure-activity relationship of coordination polymer in photocatalysis to further reveal the “supramolecularity” behind that.

Considering the extension of this detailed analysis I did not pretend to be done in the main manuscript but due to its importance I believe these issues should be addressed in a more detailed description and analysis in the SI.

Specific comments, changes and missing data:

The amount of Et₃N for the fs-TA in Fig S19 should be specified.

Response 2-6: Thanks a lot for the helpful comments. Per the advice, the extension and the detailed analyses regarding fs-TA spectra, the missing data such as the amount of Et₃N for the fs-TA in Fig S19 were described in Supplementary Experimental Methods and Analyses of Transient Absorption Spectra, Supplementary Information.

Reviewer #3 (Remarks to the Author):

The authors have prepared a much improved revised article and in my opinion I am satisfied that all of the reviewers comments have been addressed. I am, therefore, happy to recommend acceptance of this revised version.

Reply 3: We highly appreciated the reviewer for the acceptance of our manuscript. Thank you very much for your great effort in improving this paper.

Reviewer #4 (Remarks to the Author)

[The pdf version is also provided]

The new experimental results added to the revised manuscript by Duan and co-workers have indeed strengthened several claims and arguments that the authors would like to make. In particular, I appreciate seeing (i) the fsTA measurements, demonstrating the longer lifetime of the excited anion in Cd-SNDI, (ii) the reduced NDI•- absorption signals of Cd-SNDI upon addition of Ar-Cl, suggesting pre-association/electron transfer in the dark, (iii) radical trapping with TEMPO, providing the evidence of the intermediate, and (iv) the reactant-

size dependent photoreduction, suggesting the electron transfer events occurring within the porous structure of Cd-SNDI MOF. Despite the excellent results reported here, I still find a few issues that need to be addressed rigorously. Most of them are not about the experimental results, which are done very well, but related to the interpretation and discussion.

Response 4: we thank the reviewer for the recognition of our previous effort to enrich the experimental evidence, and it was also highly appreciated for the enlightening comment/guidance to improve the interpretation and discussion. In the following sections, we will answer to these comments in a point-to-point manner.

(1) The authors still present the idea of molecular segregation in a bizarre way. It feels like that they genuinely love this idea, so much so they want to relate the results to it regardless if it is appropriate. From the revised manuscript, it became clear now that the authors think that the SNDI units in Cd-SNDI stack into one column and amines or Ar-Cl “VIRTUALLY stack in situ” into another column; thus, two clusters of molecular units segregate. Two problems are associated with this proposition: (i) First, the authors do not have any structural proof of the location of amines and Ar-Cl. The changes in optical spectra do not imply their whereabouts. They may present/distribute in the MOF porous scaffold in a random and scattered manner (although being close to SNDI). The optical response also does not imply the relative arrangement of amines and Ar-Cl molecules. This proposition is doubtful, especially considering that not all SNDI units have become SNDI^{•-} (or excited SNDI^{•-}) and that the association of amines or Ar-Cl with SNDI^{•-} may be transient and dynamic. (ii) Second, if this idea of molecular segregation can not be substantiated, the materials design principle presented in this work reads weak and handwavy. In fact, for the ideas presented in Figure 1 or Scheme S2 to work, the amines and Ar-Cl molecules DO NOT have to assemble into columnar aggregates. In my opinion, this is really an unnecessary and unverified suggestion and would only make readers confused about the possible reaction mechanism.

Response 4-1: We thank the reviewer for this helpful discussion. We agreed that the association of Et₃N/Ar-Cl molecules with NDI⁰/NDI⁻ string of coordination polymer might be in a random, scattered/noncontinuous, and dynamic manner, which should be derived from the classical model of “segregated stacking”. Thus, as suggested by the referee, in the revised edition of the manuscript, the *in situ*-generated assembly of Et₃N···NDI string or Ar-Cl···NDI⁻ string was not regarded as “segregated stacking” anymore, and the related sections were rewritten.

(2) The quality of the Cd-SNDI crystal structure must be improved, as pointed out in the previous review. It is certainly common to observe non-planar polyaromatic hydrocarbons; all of these compounds have a smooth and curved structure, much like a twisted ribbon. However, the Cd-SNDI structure presented in the cif file is not sensible. The issue is not about being planar or not; the atomic arrangement of Cd-SNDI is simply unreasonable. Let me give one example to illustrate the problem: the improper dihedral angle about the NDI aromatic carbon atom attached with sulfur is 25–45° (depending on the sequence of atoms picked). This value seriously deviates from 0°, and thus this carbon is not an aromatic sp² carbon according to this cif file (for reference, the improper dihedral angle about the sp³ carbon in methane is ~35°). The quality of the data might be poor (high R_{int}) to start with, and/or the naphthalene diimide core units should be considered to be disordered in addition to the EtS part.

Response 4-2: Thanks to the reviewer to explain this concern so clearly and we are now aware of the unreasonable atomic arrangement of the original crystal structure. To fix this issue, we recollected the single-crystal X-ray diffraction data. This time, the R_{int} value decreased to 0.0459, and the naphthalene carbon atoms in the NDI core of the ligand showed improved coplanarity than the previous edition of data. In addition, the SEt branches of ligands were still disordered. The key parameters for this newly solved structure of Cd-SNDI have been updated in Table S2 of Supplementary Information, and the new cif file as well as the checkif file were also provided. All the crystal pictures of Cd-SNDI have been updated accordingly (Figs. 1, 7, S2, and S3).

(3) After sorting out the quality of the Cd-SNDI crystal structure, the authors could have better confidence in estimating the S...S distance. Concerning the distance measurement, the authors might consider the average of the two red distances in the figure below – if the improved/refined structure remained disordered.

Response 4-3: Per your advice, the average S...S distance in the newly-solved structure was 3.40 Å, which was calculated upon two possible combinations of disordered SEt groups with distances of 3.27 Å and 3.52 Å, respectively (Figs. 1b, 1c).

(4) Following (3), I would suggest the authors be cautious when describing the MOF structure using the term S...S or chalcogen-mediated interaction. In supramolecular chemistry as well as several statements in this manuscript, “S...S interaction” implies attractive stabilization energy between the two sulfur atoms. It is unlikely to be the case here. The proximity of sulfur orbitals may facilitate charge transport, but it does not necessarily imply stabilization and hence may not be the driver of the MOF structure. The positioning of atoms in the solid state (crystal) is often the consequence of many restraints imposed by other atoms; seeing any two units in proximity does not always imply interactions (attractive or repulsive alike). Again, if the authors firmly believe that there is “S...S interaction” driving the formation of the MOF structure, either a better proof or a literature reference should be provided, as pointed out in the previous review. (Ref 35 is relevant to the current study indeed, but it does NOT suggest that MOF structure therein is driven by chalcogen-mediated interaction.)

Response 4-4: Thanks for this helpful comment, and we have revised the related contents to clear the misleading descriptions.

The SEt branches of SNDI ligands were disordered in single-crystal chromatography (Figs. 1b, 1c, S2, and S3), and the proximal S...S contacts could be observed (Figs. 1b, 1c). An average S...S separation of 3.40 Å was calculated upon two possible combinations of disordered SEt groups, with distances of 3.27 Å and 3.52 Å, respectively (Fig. 1c). We held the same view that the S...S here was not the dominating driving force for the formation of Cd-SNDI. This can be clearly seen from the similar arrangement of NDI units either in the absence (Mg-NDI) or in the presence (Cd-SNDI) of proximal S...S distance. Whether the nature of S...S was attractive or repulsive could not be exclusively determined here, thus, we used a more “neutral” phrase “S...S contact” to replace the original “S...S interaction” in the revised manuscript, and the words like “chalcogen-linked/bridged interaction” would only be occasionally used in the concept design of Introduction Section.

The experimental results and theoretical calculations of Ref. 38 (previously cited as Ref. 35, *J. Am. Chem. Soc.* **137**, 1774-1777 (2015)) disclosed the remarkable correlation between the conductivities of coordination polymers/MOFs and the shortest S...S contact defined by neighbouring tetrathiafulvalene (TTF) cores within the framework. Thus, we cited it to support our hypothesis that the introduction of S-containing branches into the NDI core might bring proximal S...S contact into the coordination polymer to improve the charge-transfer ability of the framework.

Moreover, we also highly agreed with the reviewer’s suggestion that the proximal S...S between the neighbouring SNDI ligands could mediate the ground-state partial charge transfer and the excited-state photoinduced net charge transfer between neighbouring ligands. This was further evidenced by the reduced electrochemical impedance and enhanced photocurrent of Cd-SNDI (Figs. 3f, 8d), in comparison to the cases of free ligand H₄SNDI, isostructural coordination polymer Mg-NDI without S...S contact, and other NDI coordination polymers with different structures (Fig. 7).

(5) On page 9, it was suggested that the broad absorption at ~780 nm of the sample containing Cd-SNDI and Et₃N is a D₀ → D₁ transition (Fig 3a). Please clarify the molecular origin of the doublet states (Whose doublet?)

It is definitely not $\text{SNDI}^{\bullet-}$'s, as the blue trace in Fig 3a is different from the red one.). It might be informative to also comment on the fact that such 780-nm absorption was not observed for $\text{H}_4\text{SNDI}+\text{Et}_3\text{N}$ (Figure S5).

Response 4-5: Thank the reviewer for pointing out this wrong description. We agreed that this broad absorption with a peak of ~ 780 nm (blue trace in Fig. 3a) could not be assigned to the $D_0 \rightarrow D_1$ transition of the radical anion of Cd-SNDI.

When Et_3N was added to free ligand H_4SNDI , the broad charge-transfer (CT) band centred at 520 nm increased remarkably (Fig. 3b), indicating the partial electron transfer from Et_3N to NDI moiety. In comparison, when treating Cd-SNDI with Et_3N , a broad absorption band centred at red-shifted ~ 700 nm with increased intensity was detected (Fig. 3a).

Here, it might be helpful to cite Ref. 39 (*J. Am. Chem. Soc.* **141**, 17783-17795 (2019)) to understand these comparative results. In the case of improved intermolecular charge-transfer interaction between the electron-donating tetrathiafulvalene (TTF) and the supramolecular triangle assembly of electron-accepting NDI ($\text{NDI-}\Delta$), the red-shifted position, the broadened shape, and the enhanced density of charge-transfer band could be observed, which resembled our case of treating Cd-SNDI with Et_3N (Fig. 3a).

Then, from another perspective, the comparative EPR experiments were performed to track the electron transfer when adding the electron donor to ligand/coordination polymer. The charge-transfer (CT) band covering 400~600 nm of H_4SNDI reflected the intra-ligand partial electron transfer between SEt branch and NDI core, and the EPR spectrum of free ligand H_4SNDI showed a weak signal at the position of reported $\text{NDI}^{\bullet-}$ (Fig. S6a) (Ref. 39). In comparison, Cd-SNDI exhibited more remarkable signal of $\text{NDI}^{\bullet-}$ in EPR spectra (Fig. S6b), indicating the more intensive partial electron transfer which might be mediated by inter-ligand $\text{S}\cdots\text{S}$ contact in Cd-SNDI. After the addition of Et_3N , the treated coordination polymer Cd-SNDI also showed a stronger signal than the ligand H_4SNDI treated with Et_3N (Figs. S6c, S6d), suggesting a more effective partial electron transfer from Et_3N to Cd-SNDI than to the ligand. Furthermore, after shining light on the mixture of Et_3N and Cd-SNDI/ligand, very intensive EPR signals of $\text{NDI}^{\bullet-}$ were detected in both cases of coordination polymer and ligand (Figs. S6e, S6f), suggesting the photoinduced net electron transfer to generate the $\text{NDI}^{\bullet-}$ moieties after excitation.

The above-mentioned comparative results of UV-vis absorption spectra and EPR spectra indicated a stronger charge-transfer interaction between Et_3N and Cd-SNDI than in the case of Et_3N and H_4SNDI .

(6) The estimation of the photoreduction power of ScNDI and pyrene@ScNDI anion radicals is not reasonable. On pages 23-24 "If based upon the doublet-state $D_0 \rightarrow D_1$ transition peak at 495 nm, the maximum photoreducing powers ...-1.94 V": (i) 495 nm excitation is not the $D_0 \rightarrow D_1$ transition; it leads to higher doublet states. (ii) It is not obvious why the authors wanted to use the 495-nm transition for estimation instead of the lowest-energy band at ~ 780 nm. (iii) if an arbitrary absorption band and thus an arbitrary excited state is considered, -1.94 V should not be the "maximum"; any bluer-light excitation will give even higher energy. The same problem appears for the pyrene@ScNDI sample. Additionally, the word "redshifted" may be misleading, as the 495-nm excited state of ScNDI anion and the 505-nm excited state of pyrene@ScNDI anion are likely not the same excited state.

Response 4-6: Thank the reviewer for the suggestive comments.

(i) Sorry for the mistake; the 495 nm band was NOT the correct peak for the $D_0 \rightarrow D_1$ transition of radical anionic Sr-NDI.

(ii & iii) In the UV-vis absorption of $\text{NDI}^{\bullet-}$ moieties, the doublet-state transition bands were generally located in the range of 450~800 nm, typically shown as four groups of fingerprint peaks. Among them, the peak at 750~800 nm (such as the 780 nm in Sr-NDI) referred to $D_0 \rightarrow D_1$ with the lowest energy, and the peak at 450~500 nm (such as the 495 nm in Sr-NDI) referred to $D_0 \rightarrow D_n$ with the highest energy.

The reducing power of $(\text{NDI}^-)^*$ could be estimated according to the Rehm-Weller equation (Ref. 57: D. Rehm, A. Weller, *Isr. J. Chem.* **8**, 259–271 (1970)) based upon the doublet-state transition energy of NDI^- and the NDI^-/NDI redox potential. If the $D_0 \rightarrow D_n$ at 450–500 nm (such as the 495 nm in Sr–**NDI**) with the highest energy was used, the reducing power of $(\text{NDI}^-)^*$ might reach or exceed the reduction potentials of substituted aryl chlorides substrates in our manuscript. This excited-state energy estimation method was also used in the pioneering works (Ref. 8: *Science* **346**, 725–728 (2014)). Thus, in the photocatalytic set-up of this manuscript, a classic 455 nm LED was used to irradiate the $D_0 \rightarrow D_n$ band at 450–500 nm, aiming at reaching the competent reducing potential for reaction.

Compared with the fingerprint NDI^- peaks (495 nm, 610 nm, 688 nm, 780 nm) of radical anionic Sr–**NDI**, the NDI^- peaks of radical anionic pyrene@Sr–**NDI** were merged into two broad peaks centred at positions of 505 nm and 785 nm, respectively (Fig. S18). We highly agreed with the comment of the reviewer, that the 505-nm band of radical anionic pyrene@Sr–**NDI** cannot be simply regarded as being “red-shifted” from the 495-nm band of radical anionic Sr–**NDI**. Thus, we have corrected the related descriptions.

(7) As indicated in the previous review, the authors should clearly distinguish the idea of charge delocalization and transport. There is little evidence that the NDI^- anion is delocalized, given the observation of the characteristic peaks of radical anion isolated/monomeric NDI^- (Fig 3a, 3b, 3c). On the other hand, if there is some sort of rapid transport within the EPR timescale, the EPR line width may be reduced. However, it is not apparent based on the data presented.

Response 4-7: Thank the reviewer for this instructive comment.

Before answering this question, it should be helpful to revisit the charge-transfer (CT) interactions (from a structural perspective) and the related charge-transfer behaviours (from a functional perspective) in this manuscript. The charge-transfer (CT) interaction between the covalently or non-covalently linked D and A counterparts can mediate either the partial charge transfer at the ground state or the net charge transfer at the excited states. Several different kinds of charge-transfer interactions might be involved in this manuscript, such as the covalently linked intra-ligand SET- NDI^- and the non-covalently linked S \cdots S contact, and the pre-association counter pair of $\text{Et}_3\text{N}\cdots\text{Cd-SNDI}$ or the $\text{Ar-Cl}\cdots(\text{Cd-SNDI})^-$ also behaved like the non-covalently linked charge-transfer interaction.

In the previous edition, the “transportation” mainly referred to the “net” charge transfer, and the “delocalization” was related to both the “partial” charge transfer in the ground state and the “net” charge transfer in the excited state, of which the polysemy/equivocality possibly mislead the readers. Thus, in the revised edition, the use of those words was limited to the concept introduction, and they were mostly replaced by “charge transfer” with specific descriptions for disambiguation in the **Result** section of main text.

Regarding the central concern about charge “delocalization” (herein, we tentatively used this word to echo the referee comment) characteristics of NDI^- in Cd–**SNDI**, it might be helpful to interpret it in both cases of the ground and excited states.

We highly appreciated the reviewer’s advice for using EPR to track the ground-state charge transfer behaviour. It was noteworthy that the anisotropic fixation of ligands and the complex inter-ligand interactions in coordination polymers often caused the broadening of radical EPR signals (*Inorg. Chem.* **55**, 7270–7280 (2016)), disturbing the use of EPR as the probe of ground-state inter-ligand charge transfer of NDI^- in Cd–**SNDI**. We turned to UV-vis absorption for the possible clues of ground-state inter-ligand partial charge transfer. For the radical anionic samples of ligand and coordination polymers other than Cd–**SNDI**, the $D_0 \rightarrow D_1$ doublet-state transition located before 788 nm (Figs. 3b, 8b). For radical anionic Cd–**SNDI**, this band was located at 800 nm (Fig. 3a). This 12 nm red-shift suggested the lower doublet-state transition energy, indicating the inter-ligand partial charge transfer (or termed as “charge sharing”) mediated by S \cdots S contact in Cd–**SNDI**. A similar phenomenon was also disclosed by pioneering literature, the inter-ligand charge-transfer interaction in NDI^-

assembly gave rise to the obviously red-shifted doublet-state transition band of NDI^- (refer to Ref. 56: *Angew. Chem. Int. Ed.* **53**, 9476-9481 (2014)).

The transient photocurrent response was widely used to probe the photoinduced charge separation efficacy of coordination polymer/MOF-based material (refer to Ref. 55: *Chem. Eng. J.* **429**, 132377 (2022)). Under the ON/OFF photoirradiation and the negative bias potential, the 16-fold higher photocurrent of radical anionic Cd-SNDI than the respective radical anionic H_4SNDI indicated the remarkable contribution of excited-state inter-ligand charge transfer mediated by S \cdots S contact (Fig. 3f). Moreover, the comparative electrochemical impedance spectra (EIS) showed a smaller electronic impedance of radical anionic Cd-SNDI than the neutral one (Fig. 3e), reflecting the inter-ligand charge transfer of radical anionic Cd-SNDI in the presence of an induced electric field.

(8) A few experimental details need to be explained/justified more clearly. First, for the fsTA experiment, if the radical anion species are generated by photoreduction (using Et_3N as the electron donor), are all the SNDI units reduced? If this is not the case, the concentration of $\text{SNDI}^{\bullet-}$ anion may change during the course experiment, thus resulting in different concentrations of $[\text{SNDI}^{\bullet-}$ anion] and excited $^*[\text{SNDI}^{\bullet-}$ anion] at time = 0 of every 480 nm laser pulse, and therefore influencing the accuracy/credibility of kinetic analysis.

Response 4-8: Thank the reviewer to point this out. Before the fs-TA experiments, we added excess Et_3N and irradiated the resultant mixture for a long-enough time to make sure all the NDI units became the radical anions. The NDI^- was the dominative species in the samples of ligand and coordination polymer, and the presence of neutral NDI was negligible, which can be verified by the UV-vis absorption spectra shown in Figs. 3a, 3b.

Moreover, the fs-TA spectra of neutral samples for both ligand H_4SNDI and coordination polymer Cd-SNDI were collected in a comparative manner of radical anionic samples (Fig. 5). As shown in Figs. 5c-5f, the obvious differences of fs-TA spectra between neutral and radical anion samples for both ligand H_4SNDI and Cd-SNDI excluded the remarkable presence of neutral NDI species in radical anionic samples of ligand and coordination polymer.

(9) Second, it is sometimes unclear if the suspension (sonicated?) or the supernatant (after sonication) samples were used in the measurements (absorption, emission, and TA) and synthetic photochemical experiments. This should be clarified. For those using supernatant solutions, how do the authors confirm that the solution contains the pristine MOF samples instead of the structurally decomposed soluble fragments?

Response 4-9: Thank the reviewer for the notice. We used the suspension but not the supernatant for measurements (absorption, emission, and fs-TA) and synthetic photocatalytic experiments.

The ligand H_4SNDI and coordination polymer Cd-SNDI exhibited distinct spectra in the UV-vis absorption, emission, and fs-TA (Figs. 2, 3, 5), which demonstrated that it was the framework structures of Cd-SNDI but not the decomposed fragments that dominated those spectra. And we have done the PXRD before and after the synthetic photocatalytic experiments to verify the intact scaffold structure of coordination polymer/MOF samples (Fig. 2b).

(10) Third, the method for creating the difference spectra (e.g. Fig 3c) or normalized spectra (e.g. Fig 2c) should be clarified. For instance, it is not obvious what was normalized against in Fig 2c for the blue and red traces. In Fig 3c, what is the spectrum used for subtraction? If it were the neutral Cd-SNDI spectrum used for subtraction, why does the red trace in Fig 3a look different from the black trace in Fig 3c? Relatedly, it is unclear why Figure S17 was not presented as difference spectra (cf. Figures S15 or S16).

Response 4-10: Per your suggestion, detailed methods for creating the difference spectra were provided in the corresponding figure captions. In Fig. 2c, the fluorescence intensities were normalized against the maximal emission intensity of H₄SNDI, which was set to 1, to directly show the decreased fluorescence of NDI moiety after being incorporated into the coordination polymer Cd-SNDI.

In the case of blue and red traces in Fig. 3a, the spectrum for subtraction was the neutral sample Cd-SNDI. In comparison, when adding substrate Ar-Cl into radical anionic coordination polymer or ligand (such as the case of Cd-SNDI, Fig. 3c), the spectrum for subtraction was not the neutral sample like Cd-SNDI, but the mixture of neutral Cd-SNDI and Et₃N without irradiation. The rationalization was listed as follows: The weak charge-transfer interaction between Et₃N and Ar-Cl might disturb the judgement for the possibly existing pre-association of Ar-Cl⋯(NDI⁻), which would be a matter especially when the charge-transfer interaction-like pre-association was weak. Since our aim in Fig. 3c was to look for the minimal clues of possibly existing pre-association of Ar-Cl⋯(NDI⁻), the precise shape exhibition of NDI⁻ had to be compromised in this case. Thus, the difference between the red trace in Fig. 3a and the black trace in Fig. 3c was reasonable, and those details have been updated in the figure captions.

As shown in the previous edition of Figure S17, the UV-vis absorption of radical anionic Mg-SNDI was not remarkably changed upon the addition of substrate Ar-Cl or with successive photoirradiation. Per the advice of the reviewer, we changed Fig. S17 to the difference spectra to improve the readability.

(11) While I believe that H₄SNDI has been synthesized and very likely constitutes the ligand of Cd-SNDI, the ¹H and ¹³C NMR data and spectra of H₄SNDI are questionable. Numerous thiolated NDI derivatives have been reported in the literature; they all display “normal” spectra. The absence of EtS peaks and unusual chemical shifts cannot be attributed to the polar bonding between C-S (if so, the same phenomena would have been observed for other thiolated NDI). Please verify the structure/structural data. On a related note, the elemental analysis results of Cd-SNDI deviate noticeably from the expectation.

Response 4-11: we appreciated the reviewer bringing this question up.

The purity of ligand H₄SNDI was a vital factor for the quality of single crystal of coordination polymer Cd-SNDI. Owing to the very poor solubility of ligand H₄SNDI, it was not possible to purify H₄SNDI by column. It was reported that, for the NDI derivatives with poor solubility, forming the co-crystallization with carboxylic acid additive was a practical manner for purification/isomer resolution (*J. Am. Chem. Soc.* **137**, 11582-11585 (2015); *Angew. Chem. Int. Ed.* **55**, 4275-4279 (2016)). The purification of ligand in our case had to be done by recrystallization. To finely control the quality of ligand H₄SNDI for ensuring repeatability, many kinds of solvents were tested for recrystallization, and the microcrystal fine powder of ligand only could be obtained after recrystallization in acetic acid (HOAc), in the form of H₄SNDI•(CH₃CO₂H)₂, as depicted by ¹H-NMR. We verified the structure of H₄SNDI by the HRMS under the manner of matrix-assisted laser desorption ionization time-of-flight (MALDI-TOF), and experimental data 714.0625 matched well with the calculated value 714.0614 of single-molecular H₄SNDI (see: *Synthesis of H₄SNDI*, *Supplementary Information*). Although we failed to grow the single crystal of ligand H₄SNDI, the updated crystal structure of Cd-SNDI still clearly showed the substitution of the NDI core with SEt groups. Moreover, the element analysis (EA) was performed to examine the recrystallized ligand, confirming the phase purity, as shown below: Anal. Calcd (%) for H₄SNDI•(CH₃CO₂H)₂ (C₃₈H₃₀N₂O₁₆S₂): C, 54.68; H, 3.62; N, 3.36. Found: C, 54.60; H, 3.59; N, 3.32 (see: *Synthesis of H₄SNDI*, *Supplementary Information*).

Regarding the ¹H-NMR analysis of ligand H₄SNDI, the downfield-shifting and broadening of C-H of the NDI core were possibly due to the complexation of H₄SNDI with HOAc, and similar phenomena were also observed in reported examples of co-crystallization of NDI derivatives with carboxylic acid additives (*Angew. Chem. Int. Ed.* **55**, 4275-4279 (2016); *J. Am. Chem. Soc.* **137**, 11582-11585 (2015); see also: Asahi Research Center Co, Ltd. *Handbook of proton-NMR spectra and data*. 2015).

In the reported case of $n \rightarrow \pi$ charge-transfer interaction between the dimethylamine branch (D) and NDI (A), the electron transfer from dimethylamine moiety to NDI core could be triggered by the assistance of heat or photoirradiation to generate the radical anionic $\text{NDI}^{\cdot-}$ and the radical cationic $(\text{NMe}_2)^{\cdot+}$, which lead to the disappearance of NMR signals of both NDI core (A) and *N*-methyl of dimethylamine branch (D), and also resulted in the emergence of EPR signal of radical anionic $\text{NDI}^{\cdot-}$ (*J. Mater. Chem. C*, **7**, 4466–4474 (2019)). Those phenomena resembled the simultaneous disappearance of NMR signals of NDI core (A) and *S*-ethyl branch (D) (see: ^1H and ^{13}C NMR spectra of the isolated compounds, Supplementary Information) and the emergence of EPR signal of radical anionic $\text{NDI}^{\cdot-}$ (Fig. S6) for the ligand H_4SNDI obtained by recrystallization in HOAc under daylight (see: Synthesis of H_4SNDI , Supplementary Information).

Indeed, as commented by Reviewer #4, many reported thiolated NDI structures showed the “normal” NMR spectra (*Energy Environ. Sci.* **6**, 2172–2177 (2013); *Chem. Eur. J.* **21**, 6202–6207 (2015); *Chem. Eur. J.* **22**, 2648–2657 (2016)), indicating the not remarkable electron transfers from S-containing branches to NDI cores. Thus, it might be helpful to disclose the specific factors that spurred the electron transfer from SET branch to NDI core during the processes of preparation, workup, purification, and spectrum examination for the ligand. If we performed the procedures of -SEt decoration reaction, workup, recrystallization of ligand H_4SNDI , and handling the NMR sample under darkness, one side of SET group showed the “normal” status in ^1H -NMR spectra, and another SET of the two branches exhibited the remarkably broadening and desplitting characteristics in ^1H -NMR; and the corresponding CH_2 and CH_3 peaks of SET group were visible in ^{13}C -NMR (see: ^1H and ^{13}C NMR spectra of the isolated compounds, Supplementary Information). In sharp contrast, if those procedures were done as previously under the ambient daylight, both SET groups could not be seen in ^1H - and ^{13}C -NMR spectra (see: ^1H and ^{13}C NMR spectra of the isolated compounds, Supplementary Information), implying the possible presence of paramagnetic radical characteristics. Those results reflected that the photoirradiation from the ambient daylight possibly enhanced the partial electron transfer from the SET branch to the NDI core of the ligand.

Thus, considering the “presence” of SET-branch in the NMR spectra, the NMR data obtained under dark conditions were used to replace the original ones, and the relative information was described in Synthesis of H_4SNDI , Supplementary Information.

We speculated that some structural reasons might be correlated with the charge-transfer ability of ligand under the weak photoirradiation from ambient daylight. After the esterification of H_4SNDI with MeOH and the successive HOAc recrystallization in the presence of daylight, both two SET groups appeared in NMR, displaying the “normal” peaks (see: ^1H and ^{13}C NMR spectra of the isolated compounds, Supplementary Information). Furthermore, it was well-known that the anion $\rightarrow \pi$ charge-transfer interactions between the anion (F^- , RCO_2^- , I^- , CN^- , etc.) and the neutral NDI moiety could greatly enhance the electron transfer from electron source to NDI moiety (see: *J. Am. Chem. Soc.* **133**, 15256–15259 (2011); *Inorg. Chem.* **59**, 13371–13382 (2020); *J. Mater. Chem. C*, **7**, 4466–4474 (2019)). Although the deep discussion regarding this item might go beyond the bounds of this manuscript, still we made the deduction based upon the above-discussed results and pioneering literatures, that the carboxylic acid groups possibly played a role in enhancing the photosensitivity of intra-ligand charge transfer of H_4SNDI towards the weak irradiation such as the daylight.

Regarding the difference of elemental analysis results of Cd- SNDI from the expected value, it should be noted that the not precise match between the expected and experimental values of elemental analysis (EA) results for coordination polymer/MOF was often encountered in the pioneering works (*Nature* **402**, 276–279 (1999)), which might be caused by the loss or absorbance of the solvent molecule when transferring the single crystals out of mother liquid, and further get affected by ambient moisture or the testing conditions. This phenomenon was widely considered and somehow tolerated by pioneering research papers (such as the case of NDI-SEt coordination polymer/MOF: *Energy Environ. Sci.*, **6**, 2172–2177 (2013)). After careful comparison with the reported pioneering results, it was believed that the degree of difference between the expected and experimental EA values in the case of Cd- SNDI was still in the error tolerance scopes of those pioneering results.

(12) Nafion is strongly acidic and thus may affect the structure or structural integrity of MOF samples examined in this work. The use of it raises concerns about the impedance data.

Response 4-12: We were aware of the reviewer's concern and will try our best to solve it.

The use of Nafion as the dispersant reagent and the adhesive reagent was a common strategy for electrochemical analyses of hydrophobic coordination polymers/MOFs to improve their electronic conductivity in aqueous/protonic solvents and electrolytes (*Nat. Commun.* **13**, 4592 (2022); *J. Am. Chem. Soc.* **144**, 8676-8682 (2022)). The polymer dissolved in the aqueous solvent (like the case of polymer acidic Nafion) hardly diffused deeply into the pores of hydrophobic coordination polymer/MOF crystals, but majorly twined around the crystal surfaces (see the systematic pioneering works of Prof. Takashi Uemura, <https://uemuragroup.t.u-tokyo.ac.jp/en/>), which provided the possibility of maintaining the structure of inner phase of crystals.

To limit the possible influence of polymer acid Nafion on the structural integrity of coordination polymer, when preparing the working electrode, the dose of Nafion was small. For our samples, the volume ratio of Nafion to solvent was 3:50. After the impedance measurement of Cd-SNDI, the PXRD pattern of the recovered sample was examined, showing that the major framework scaffold was possibly maintained, as shown in Fig. S1a.

(13) Regarding “Considering the well-described charge delocalization ability of S···S mediated neutral or negatively charged NDI string,....” (page 11): (i) the statement “charge delocalization ...neutral NDI string...” does not make sense; and (ii) please provide references for the “well-described” charge delocalization.

Response 4-13: We thank the reviewer to point this issue out. In the previous edition, the statement “Considering the well-described charge delocalization ability of S···S mediated neutral or negatively charged NDI string,....” was the concomitant description of the original concept of “segregated stacking”. As suggested by the referee (**Response 4-1**), in the revised edition of the manuscript, the *in situ*-generated assembly of Et₃N···NDI string or Ar-Cl···NDI⁻ string was not regarded as “segregated stacking” any more, and the related sections were thoroughly rewritten. Thus, the original statement “Considering the well-described charge delocalization ability of S···S mediated neutral or negatively charged NDI string,....” was removed.

Regarding the concern about the experimental and literature evidence of “charge delocalization” or related words, we have made a thorough explanation in **Response 4-7**. To circumvent the possible repetition, it will be highly appreciated if the reviewer could refer to **Response 4-7**.

Reviewers' Comments:

Reviewer #2:

Remarks to the Author:

Revision is in an attached file.

Summary:

The manuscript describe the synthesis of the a coordination polymer using naphthalene diimide (NDI) substituted with SEt and Cd(II), achieving an S...S contact, which facilitated the electron mobility while simultaneously maintaining the reduced power of the ground state and the excited-state reducing power of its radical anionic species. This particular characteristic allows the authors to demonstrate the superior behavior of this heterogeneous photocatalyst in a consecutive photocatalysis (conPET) for the photoreduction of aryl halides including aryl chlorides with EWG. They show the benefits of the strategy comparing the performance to single-molecule dye photocatalysis and other coordination polymers. The photocatalytic system also demonstrates good performance in successive radical couplings allowing the formation of C-C, C-S, C-P, and C-B bonds by known procedures.

The authors claims that enhanced the electron delocalization within the polymer improve the PET from excited-state radical anion (NDI \cdot^-)* to aryl halides. The approach circumvents the short lifetime of excited states and also could avoid competing back electron transfer in the case of aryl halides. This strategy is interesting and I believe it would reach a broad audience in which ET processes from excited states are important and the work could contain the merits for publication in this journal.

I believe the authors have addressed most of the comments I had pointed out in the second revision. However, some particular corrections are still need that I described in the next section. In particular, the experimental details of some experiments are still missing which constitute a critical point for any scientific publication.

I recommend publication of the manuscript after these revisions have been done.

Comments and corrections:

Fig 1. Section d-f Fig 1 is confusing and misleading and should be modified.

Excitations events in Fig 1 are confusing and do not properly reflect the chemical catalytic cycle , in fact it is not clear in the figure which species are excited, which is fundamental for the process. Ground and excited states should be well defined in the descriptions of the figure, if I compare with Fig S1 it seems that “glow” structures are excited ones, but this does not correlate with Fig 1.

I believe the way of showing the excitation of molecules is better represented in Fig S1c and perhaps this could be translated to Fig 1.

- The $g \rightarrow d$ process implies an PET and consequently in Fig 1d a EtN $^{+\cdot}$ should be present.

- For $d \rightarrow e$ another photoexcitation is shown as a second PET with the formation of another radical anion, however in the legend of d the author mention “SS contact allowed the inter-ligand charge transfer and the transport of (NDI \cdot^-)*” but there is no representation of (NDI \cdot^-)* in Fig 1e, in fact (NDI \cdot^-)* is not represented in Fig1. If the light symbol represents more than one excitation is

not clear indicated in the description.

- For $e \rightarrow f$ another injection of light will imply PET or excitation but Fig 1f does not contain a difference with Fig 1e in this regard because excited states are not indicated.

- For $f \rightarrow g$ if NDI•- were excited in the previous event this process should not involve an excitation event, it should contain only the ET event to form ArX•-.

- It is not clear what the “glow” of NDI in Fig. 1D-f means.

Fig 2c. Experimental details are still missing (SI starts with Fig 3a).

- the sentence “a, Normalized absorption and emission spectra of Cd-SNDI and H4SNDI.” should be changed by “a, Normalized absorption and emission spectra of **suspension of Cd-SNDI and a solution of H4SNDI.**”

Fig 3c. Experimental details are still missing.

Lines 195-201, Fig 3c and Fig S16. Experimental details to reproduce this figure are missing. The interpretation of the intensity of the signals in this figure is critical in the analysis described which makes the condition of addition of substrate to the solution very important. In addition, I believe an “interaction” could be claimed but a “charge-transfer interaction” is more difficult to invoke, or a proper citation needs to be included. Experimental details of Figure 6e-f are missing.

Fig S15c. I am not sure what the point of this figure is. There is no need of a figure for this sentence “*These results **excluded** the existence of noticeable charge-transfer interaction or pre-association between the neutral Cd-SNDI and Ar-Cl substrate*” absence of interaction is better “show” without a Figure.

Line 229. “(D0 → Dn)” should be changed to “(D0 → D1)”

Line 232. Although ET and fragmentation could have a relation, in this case reduction potential values are only related to the electron transfer event and not to the fragmentation event. For these reason, the following sentence should be changed:

“...possessed sufficient driving forces for the cleavage of inert CAr-Halide bonds..” should be hanged by “...possessed sufficient driving forces for *the ET to the substrate allowing the cleavage of C-halide bonds*”

Line 232. I did not see the relevance of Ref. 62. Reference should be deleted or changed.

Fig 5. Experimental details are still missing. Preparation of solution or suspensions for mesurements is not indicated.

Line 311. “**much** shorter” should be changed by “shorter”

Lines 316-335

The analysis of fs-TA experiments have some points that need to be improved.

It should be a connection between the values of lifetimes of NDI^{•-} obtained at both excitation wavelength (630 nm and 480 nm). It is expected to see different processes but the D1 → D0 time should be associated with comparable values or its difference explained. In relation to this, I wonder why the traces of experiments at 630 nm were not adjusted to a double decay, this will probably give a better fit and relate better with the experiments at 480 nm.

Despite the previous comment, I believe the experiments at 630 nm shows a clear improvement in the polymer properties. On the other hand, experiments at 480 nm, although valuable, are much more complex since they involve more events and species. This opens the possibility to many valid interpretations that explain the results which may need more work to be confirmed or refuted. For these reasons, I would recommend keeping the more complicated interpretations of experiments at 480 nm in the SI, since they are a possible explanation that supports the 630 nm experiment more than a closed discussion. (See SI part for more comments)

Line 388-390 “despite the excited-state radical anionic forms (Fig. 8b) of both Mg–NDI and Cd–NDI having sufficient reducing potentials (estimated to be ca. -2.27 V and ca. -2.16 V, respectively, vs. SCE)” should be changed by “despite the excited-state radical anionic forms of both Mg–NDI and Cd–NDI having sufficient reducing potentials (estimated from Fig. 8b to be ca. -2.27 V and ca. -2.16 V, respectively, vs. SCE)”

Line 467 “peak at 495 nm” should be changed by “peak at 780 nm”

Line 468-469 “longer-wavelength 505 nm band” should be changed by “longer-wavelength 785 nm band”

Line 466-471 The values for E° of excite states should be revised. If the values are correct, they are too close, that is, these *estimated* reduction potential could not be considered really diferents. Anyway I believe the comparison between Sr–NDI and pyrene@Sr–NDI is not necessary to justify cAr –Cl results with these polymers, because none of them is reactive with this substrate. In this case again the comparison is with the reactive polymer, that is , Cd–SNDI vs pyrene@Sr–NDI (and Cd–SNDI vs Sr–NDI).

Supporting Information

Fig S6. Experimental details of Figure 6e-f are missing. Solution – suspension? Concentration? Solvent?

Fig S17. Scale in the y axis are missing, so the “negligible variations” cannot be judged.

Page S16

In the last two paragraphs the author discusses the disproportion reaction to form NDI²⁻ and NDI in the polymer. Although I agree with “possibly allowing the position shifting of (NDI 0)/(NDI •-)/(NDI 2-)” I am not sure about the following “and corresponding excited states”. Recombination

of NDI²⁻ and NDI have shown to recover NDI⁻ but not NDI^{-*}. In consequence, I recommend to change the sentence of this paragraph accordingly,

- “and corresponding excited states” should be deleted.

- “the regeneration of (NDI •⁻) * in coordination polymer” should be changed by “the regeneration of NDI •⁻ in coordination polymer”.

- “might regenerate (NDI •⁻) * at remote locations” should be changed by “might regenerate NDI •⁻ at remote locations”

Minor suggestions

Lines 102-105. In this sentence, an advantage is attributed to the presence of the sulphur substituent but Ref. 9 and 49 put the attention more to the material independent of the S-substituent. Citation should be changed or the sentence should not refer specifically to S-containing NDI materials.

Lines 208-212. I believe the attempt to resume two points makes the reading too heavy. I would recommend splitting the sentence in Et₃N and ArCl.

Alternative I would recommend to change “the collision and association of Et₃N/Ar–Cl with an arbitrary site of neutral **or** negatively charged NDI string would be possible to form the encounter pair for the PET process, and the separated charges would be carried away for the next rounds of encountering and PET events.” by “the collision and association of Et₃N (**or Ar–Cl**) with an arbitrary site of neutral (**or negatively charged**) NDI string would be possible to form the encounter pair for the PET process, and the separated charges would be carried away for the next rounds of encountering and PET events.”

Reviewer #4:

Remarks to the Author:

Most of the issues in the previous version have been addressed in this revised manuscript. I would like to recommend it for publication after minor clarification.

I still find a few plots of difference spectra strange looking. For instance, the red trace in Figure 3a seems to have a broad absorbance 'background' of 0.2 Δ OD from 500-1000 nm on top of the peaks originating from the radical anion of Cd-SNDI. Therefore, the red trace alone seems to represent more than one species (the radical anion); such a plot may cause confusion. I understand that improving the quality of this kind of spectra might not be very straightforward, so I would leave the authors to judge how much improvement can be reasonably made.

On the other hand, I believe the spectrum normalization issue should definitely be clarified. I suggest the authors review how the "normalized emission spectra" in Fig 2c were measured (although this plot is not an essential figure of this manuscript). The current representation suggests that the photoluminescence quantum yield of Cd-SNDI is roughly 75% smaller than that of H2SNDI (i.e. PLQY of CdSNDI \sim 0.25 x PLQY of H2SNDI). This comparison can only be made if the number of photons absorbed by the examined samples is identical (or, less ideally, similar). If this condition is not met, Fig 2c may seem misleading unless the authors explain how the plot was made. This issue was brought up in the previous review, but I probably did not make the point clearly -- if two measurements were not made on the same ground (in the current case, same absorbance), the process of normalization and comparison is not justified.

In a similar vein, the comparison of the EPR intensities from different samples (e.g. those in Fig 2 and Fig S6) may not be trivial as the actual amount of samples loaded into the EPR cavity can vary. However, the intensity increment after irradiation is significant most of times, so I am not too concerned about this comparison. Finally, I want to take this chance to point out that the so-called 'partial charge transfer' (a scenario often considered in, for example, intramolecular charge-transfer push-pull chromophores) should not give EPR response -- no matter how weak the signal intensity is. This last point is just a comment that I would like to share with the authors.

REVIEWER COMMENTS

Reviewer #2 (Remarks to the Author):

Revision is in an attached file.

Summary:

The manuscript describe the synthesis of the a coordination polymer using naphthalene diimide (NDI) substituted with SEt and Cd(II), achieving an S···S contact, which facilitated the electron mobility while simultaneously maintaining the reduced power of the ground state and the excitedstate reducing power of its radical anionic species. This particular characteristic allows the authors to demonstrate the superior behavior of this heterogeneous photocatalyst in a consecutive photocatalysis (conPET) for the photoreduction of aryl halides including aryl chlorides with EWG. They show the benefits of the strategy comparing the performance to single-molecule dye photocatalysis and other coordination polymers. The photocatalytic system also demonstrates good performance in successive radical couplings allowing the formation of C-C, C-S, C-P, and C-B bonds by known procedures.

The authors claims that enhanced the electron delocalization within the polymer improve the PET from excited-state radical anion (NDI^{•-})* to aryl halides. The approach circumvents the short lifetime of excited states and also could avoid competing back electron transfer in the case of aryl halides. This strategy is interesting and I believe it would reach a broad audience in which ET processes from excited states are important and the work could contain the merits for publication in this journal.

I believe the authors have addressed most of the comments I had pointed out in the second revision. However, some particular corrections are still need that I described in the next section. In particular, the experimental details of some experiments are still missing which constitute a critical point for any scientific publication.

I recommend publication of the manuscript after these revisions have been done.

Response 2-1: We highly appreciated the Reviewer for the positive comments about our previous revision, the instructive suggestions regarding the experimental details, and the precise interpretation of Figs. and texts. As shown below, the mentioned issues and items were revised and responded accordingly.

Comments and corrections:

Fig 1. Section d-f Fig 1 is confusing and misleading and should be modified.

Excitations events in Fig 1 are confusing and do not properly reflect the chemical catalytic cycle, in fact it is not clear in the figure which species are excited, which is fundamental for the process. Ground and excited states should be well defined in the descriptions of the figure, if I compare with Fig S1 it seems that “glow” structures are excited ones, but this does not correlate with Fig 1.

I believe the way of showing the excitation of molecules is better represented in Fig S1c and perhaps this could be translated to Fig 1.

- The $g \rightarrow d$ process implies an PET and consequently in Fig 1d a Et_3N^+ should be present.
- For $d \rightarrow e$ another photoexcitation is shown as a second PET with the formation of another radical anion, however in the legend of d the author mention “*SS contact allowed the inter-ligand charge transfer and the transport of $(\text{NDI}^{\bullet-})^*$* ” but there is no representation of $(\text{NDI}^{\bullet-})^*$ in Fig 1e, in fact $(\text{NDI}^{\bullet-})^*$ is not represented in Fig1. If the light symbol represents more than one excitation is not clear indicated in the description.
- For $e \rightarrow f$ another injection of light will imply PET or excitation but Fig 1f does not contain a difference with Fig 1e in this regard because excited states are not indicated.
- For $f \rightarrow g$ if $\text{NDI}^{\bullet-}$ were excited in the previous event this process should not involve an excitation event, it should contain only the ET event to form $\text{ArX}^{\cdot-}$.
- It is not clear what the “glow” of NDI in Fig. 1D-f means.

Response 2-2: We highly appreciated the Reviewer’s enlightening comments to improve the conceptual illustration of the manuscript.

After an overall consideration of the Reviewer’s advice, **Figs. 1d-1g** were thoroughly re-edited, and **Fig. S1** was also revised to improve the readability.

Fig. S1 was more like a macroscopic design and comparative illustration of concept, which focused on how the inter-ligand electronic communication within coordination polymer was expected to improve the accessibility of $(\text{NDI}^{\bullet-})^*$ towards the substrate, and to enhance the reactivity of $(\text{NDI}^{\bullet-})^*$ in photocatalytic Ar-X cleavage.

In comparison, **Figs. 1d-1g** was expected to show the structure-activity relationship of the major player catalyst, Cd-SNDI , more realistically and in more detail. **Figs. 1d-1g** included but was not limited to the following contents: (i) how the inter-ligand electronic communication through S...S contact could regenerate $\text{NDI}^{\bullet-}$ at the proximal position of Ar-X that diffused into the channel; (ii) how the interactions/associations such as neutral $\text{NDI}^{\bullet-}$... Et_3N and radical anionic $\text{NDI}^{\bullet-}$...Ar-X enhanced the formation of encounter pairs for PET events; (iii) how the $\text{NDI}^{\bullet-}$ that regenerated proximally to the one-electron reduced $[\text{Ar-X}]^{\cdot-}$ could alleviate the back-electron transfer from $[\text{Ar-X}]^{\cdot-}$ to the framework.

Regarding the concerns and suggestions of the Reviewer, the revised details of **Figs. 1d-1g** were listed below:

(1) About the presentation of excited-state neutral NDI and radical anionic $\text{NDI}^{\bullet-}$: since the neutral NDI and radical anionic $\text{NDI}^{\bullet-}$ were represented in different colours, the further use of “glow” for showing their excited states might be too complex and easily dazzle the readers. Thus, we no longer used “glow” in the revised **Figs. 1d-1g**. Instead, the “injection of light” symbols & affiliated “ $h\nu$ ” text marks and the “electron transfer” curve arrows & affiliated “e” text marks were employed at the same time to depict the excitation events of NDI or $\text{NDI}^{\bullet-}$ and the concomitant electron transfer behaviours. In case of **NO** above-mentioned symbols or curve arrows with the corresponding affiliated text marks, the neutral NDI or radical anionic $\text{NDI}^{\bullet-}$ should be regarded as being in the ground state.

(2) The Et_3N^+ was added to **Figs. 1d-1g** as the counter ion of $\text{NDI}^{\bullet-}$ to keep the charge balance.

(3) The S...S contacts were supplemented to the “edge” parts of each light grey watermark of **Figs. 1d-1g**, since the inter-ligand electron transfers might occur along S...S contacts and go beyond the boundaries of three NDI units shown in the light grey watermarks of **Figs. 1d-1g**.

Fig 2c. Experimental details are still missing (SI starts with Fig 3a).

- the sentence “a, Normalized absorption and emission spectra of Cd–SNDI and H4SNDI.” should be changed by “a, Normalized absorption and emission spectra of **suspension of Cd–SNDI and a solution of H4SNDI.**”

Response 2-3: Per your advice, the experimental details for Fig. 2c were added in **Experimental Details for Figures in Manuscript, Supplementary Information**, and the mentioned caption of Fig. 2a has been revised as suggested by the Reviewer.

Fig 3c. Experimental details are still missing.

Lines 195-201, Fig 3c and Fig S16. Experimental details to reproduce this figure are missing. The interpretation of the intensity of the signals in this figure is critical in the analysis described which makes the condition of addition of substrate to the solution very important. In addition, I believe an “interaction” could be claimed but a “charge transfer interaction” is more difficult to invoke, or a proper citation needs to be included. Experimental details of **Figure S6e-f** are missing.

Response 2-4: We highly agreed with the Reviewer that the detailed descriptions of experimental conditions were critical.

For Fig. 3c, we have added the experimental details in **Experimental Details for Figures in Manuscript, Supplementary Information**.

For Fig. S16, the experimental details were added to the figure caption.

Besides, the “charge transfer interaction” between electron-deficient $C_{Ar}-Cl$ and NDI^- , which we stated in the previous edition, has been changed to the “interaction” per the Reviewer’s suggestion.

For **Figs. S6e and S6f**, to compare the radical anionic EPR signals of **H4SNDI** and **Cd–SNDI** more directly, the normalized spectra of the previous edition were changed to the original curves with intensities in the y-axis in this new edition, and the figure captions were revised to show the experimental details and improve the readability.

Fig S15c. I am not sure what the point of this figure is. There is no need of a figure for this sentence “*These results excluded the existence of noticeable charge-transfer interaction or pre-association between the neutral Cd–SNDI and Ar–Cl substrate*” absence of interaction is better “show” without a Figure.

Response 2-5: We thank the Reviewer for this suggestion, and have deleted **Fig. S15c**.

Line 229. “(D0 → Dn)” should be changed to “(D0 → D1)”

Line 232. Although ET and fragmentation could have a relation, in this case reduction potential values are only related to the electron transfer event and not to the fragmentation event. For these reason, the following sentence should be changed:

“...possessed sufficient driving forces for the cleavage of inert CAr–Halide bonds..” should be changed by “...possessed sufficient driving forces for the ET to the substrate allowing the cleavage of C-halide bonds”

Line 232. I did not see the relevance of Ref. 62. Reference should be deleted or changed.

Fig 5. Experimental details are still missing. Preparation of solution or suspensions for measurements is not indicated.

Line 311. “much shorter” should be changed by “shorter”

Response 2-6: We thank the Reviewer for all these detailed comments and have revised the corresponding points accordingly. The mentioned mistakes or extra words have been changed.

The original Ref. 62 was about the measurement setup for fs-TA analysis, which was removed from the main text and only kept in the Method section of the Manuscript.

Experimental details for Fig. 5 were provided in the second paragraph of Supplementary Experimental Methods and Analyses of Transient Absorption Spectra in Supplementary Information, and we have added a brief explanatory note in the caption of Fig. 5 to improve the readability.

Lines 316-335 The analysis of fs-TA experiments have some points that need to be improved.

It should be a connection between the values of lifetimes of NDI^{•-}* obtained at both excitation wavelength (630 nm and 480 nm). It is expected to see different processes but the D₁→D₀ time should be associated with comparable values or its difference explained. In relation to this, I wonder why the traces of experiments at 630 nm were not adjusted to a double decay, this will probably give a better fit and relate better with the experiments at 480 nm.

Despite the previous comment, I believe the experiments at 630 nm shows a clear improvement in the polymer properties. On the other hand, experiments at 480 nm, although valuable, are much more complex since they involve more events and species. This opens the possibility to many valid interpretations that explain the results which may need more work to be confirmed or refuted. For these reasons, I would recommend keeping the more complicated interpretations of experiments at 480 nm in the SI, since they are a possible explanation that supports the 630 nm experiment more than a closed discussion. (See SI part for more comments)

Response 2-7: We thank the Reviewer for giving us these instructive comments and the opportunity for improvement.

It was known that the excitation of aromatic diimides at different wavelengths can lead to different excited-state behaviors (*Chem. Sci.* **2017**, *8*, 3821–3831). For example, in Ref. 45 (*J. Phys. Chem. B* **2019**, *123*, 7731–7739), a single excited-state decay with a lifetime of 142 ps was obtained for D₁→D₀ transition of (NDI^{•-})^{*} under 780 nm excitation. When the excitation laser wavelength was changed to a higher energetic wavelength of 605 nm, a two-component kinetic model was required to fit the data, of which the lifetime of 103 ps was attributed to D₁→D₀ transition and another lifetime of 0.9 ps was assigned to the decay of a higher-lying excited state. Moreover, the different experimental setups also possibly affected the fitting of decay processes; it was reported that the 605 nm excitation on NDI^{•-} can also give the decay lifetimes with the single-exponential time constants (*J. Phys. Chem. A* **2000**, *104*, 6545–6551).

Regarding the decay for the trace of 630 nm experiment in our case, we have tried the single- or double-exponential decay fitting, and the better fit was the single-exponential decay

(Figs. 5b, 5d), which might be correlated to $D_1 \rightarrow D_0$ transition as depicted by the Reviewer. The window range of 350~500 nm seemed proper to exhibit the excitation of NDI^- , without severe disturbance/overlap from the signals of intra- or inter-ligand charge separations after excitation of NDI^- (Figs. 5a, 5c).

When irradiating the radical anionic Cd-SNDI by 480 nm laser, the window range of 500~750 nm was more suitable to observe the events and species after excitation of NDI^- , such as the intra- and inter-ligand charge separations, of which the ns-scaled decay lifetimes were significantly different from $D_1 \rightarrow D_0$ transitions of $(\text{NDI}^-)^*$ (Fig. S19).

Based upon the above-mentioned literature information and experimental results, we highly agreed with the comments of the Reviewer that “the experiments at 630 nm shows a clear improvement in the polymer properties. On the other hand, experiments at 480 nm, although valuable, are much more complex since they involve more events and species”. Thus, we shifted the major part of the results and explanations for 480 nm laser irradiation experiments from the main text to **Supplementary Information, Supplementary Experimental Methods and Analyses of Transient Absorption Spectra**.

Line 388-390 “despite the excited-state radical anionic forms (Fig. 8b) of both Mg-NDI and Cd-NDI having sufficient reducing potentials (estimated to be ca. -2.27 V and ca. -2.16 V, respectively, vs. SCE)” should be changed by “despite the excited-state radical anionic forms of both Mg-NDI and Cd-NDI having sufficient reducing potentials (estimated from Fig. 8b to be ca. -2.27 V and ca. -2.16 V, respectively, vs. SCE)”

Response 2-8: We have revised this sentence per the Reviewer’s suggestion.

Line 467 “peak at 495 nm” should be changed by “peak at 780 nm”

Line 468-469 “longer-wavelength 505 nm band” should be changed by “longer-wavelength 785 nm band”

Line 466-471 The values for E° of excite states should be revised. If the values are correct, they are too close, that is, these *estimated* reduction potential could not be considered really diferents. Anyway I believe the comparison between Sr-NDI and pyrene@Sr-NDI is not necessary to justify CAr –Cl results with these polymers, because none of them is reactive with this substrate. In this case again the comparison is with the reactive polymer, that is, Cd-SNDI vs pyrene@Sr-NDI (and Cd-SNDI vs Sr-NDI).

Response 2-9: We thank the Reviewer for these helpful comments. The relative wavelengths have been modified, and the values of E° were corrected. We agreed that the photoreducing power of radical anionic Sr-NDI and pyrene@Sr-NDI were very close, and the comparison should be conducted for Cd-SNDI vs. pyrene@Sr-NDI (and Cd-SNDI vs. Sr-NDI). Those related contents have been modified accordingly.

Supporting Information

Fig S6. Experimental details of Figure S6e-f are missing. Solution – suspension? Concentration? Solvent?

Fig S17. Scale in the y axis are missing, so the “negligible variations” cannot be judged.

Response 2-10: We thank the Reviewer for these detailed comments. As requested, we provided the experimental details of Figs. S6e-f in the caption of Fig. S6, and also added the scale to the y-axis of Fig. S17.

Page S16

In the last two paragraphs the author discusses the disproportion reaction to form NDI²⁻ and NDI in the polymer. Although I agree with “possibly allowing the position shifting of (NDI⁰)/(NDI^{•-})/(NDI²⁻)” I am not sure about the following “and corresponding excited states”. Recombination of NDI²⁻ and NDI have shown to recover NDI^{•-} but not NDI^{•*}. In consequence, I recommend to change the sentence of this paragraph accordingly,

- “and corresponding excited states” should be deleted.
- “the regeneration of (NDI^{•-})^{*} in coordination polymer” should be changed by “the regeneration of NDI^{•-} in coordination polymer”.
- “might regenerate (NDI^{•-})^{*} at remote locations” should be changed by “might regenerate NDI^{•-} at remote locations”

Response 2-11: We thank the Reviewer for these important comments to help us remove the possible misleading descriptions. As depicted by the Reviewer, the recombination of NDI²⁻ and NDI could only generate NDI^{•-} while the generation of (NDI^{•-})^{*} required further excitation. Accordingly, we revised the above items and the whole paragraph to improve the demonstration.

Minor suggestions

Lines 102-105. In this sentence, an advantage is attributed to the presence of the sulphur substituent but Ref. 9 and 49 put the attention more to the material independent of the S-substituent. Citation should be changed or the sentence should not refer specifically to S-containing NDI materials.

Response 2-12: We truly appreciated the Reviewer’s helpful comments. The previous As stated by the Reviewer, the original Refs. 9 and 46 were related to the excited state of the NDI radical anion. Whereas, the description here needed the reference to state the benefit of sulfur substitutes in the charge-transfer interaction with electron-deficient aryl radical precursor. Thus, we deleted the original Ref. 9 and replaced the original Ref. 46 by the new one which showed the sulfur substitutes of anionic aryl compound to assist the formation of a charge-transfer complex with electron-deficient aryl halides (new Ref. 46: *J. Am. Chem. Soc.* **2017**, *139*, 13616-13619).

Lines 208-212. I believe the attempt to resume two points makes the reading too heavy. I would recommend splitting the sentence in Et₃N and ArCl.

Alternative I would recommend to change “the collision and association of Et₃N/Ar–Cl with an arbitrary site of neutral or negatively charged NDI string would be possible to form the encounter pair for the PET process, and the separated charges would be carried away for the next rounds of encountering and PET events.” by “the collision and association of Et₃N (or Ar–Cl) with an arbitrary site of neutral (or negatively charged) NDI string would be possible to

form the encounter pair for the PET process, and the separated charges would be carried away for the next rounds of encountering and PET events.”

Response 2-13: Per your good suggestion, we have revised the corresponding descriptions.

Reviewer #4 (Remarks to the Author):

Most of the issues in the previous version have been addressed in this revised manuscript. I would like to recommend it for publication after minor clarification.

Response 4-1: We thank the Reviewer for recognizing our previous efforts to improve our work. In the following, we responded to the comments in a one-by-one manner.

I still find a few plots of difference spectra strange looking. For instance, the red trace in Figure 3a seems to have a broad absorbance ‘background’ of 0.2 Δ OD from 500-1000 nm on top of the peaks originating from the radical anion of Cd-SNDI. Therefore, the red trace alone seems to represent more than one species (the radical anion); such a plot may cause confusion. I understand that improving the quality of this kind of spectra might not be very straightforward, so I would leave the authors to judge how much improvement can be reasonably made.

Response 4-2: We thank the Reviewer for allowing us to explain this ‘strange looking’ of the previous version of Fig. 3a and the possible improvement.

Different from the smooth absorbance curves of the solution, the UV-vis absorption spectra of suspension often suffered from the scattering phenomena which might be caused by the size, degree of dispersion/aggregation, degree of solvation, and other factors of particles in the suspension, easily leading to the fluctuation or shift of baseline (see: *Ultraviolet and visible spectroscopy*, By: Gauglitz, Gunter; Edited by: Guenzler, Helmut; Williams, Alex; *Handbook of Analytical Techniques*, 2001, 1, 419-463).

In the previous version of Fig. 3a, the broad band of Et₃N...Cd-SNDI charge-transfer interaction and four finger-print peaks of radical anionic Cd-SNDI overlapped with the up-shifted and fluctuated baselines. It was speculated that the big particle size of the old batch of Cd-SNDI suspension possibly contributed to the “strange looking” of spectra. To circumvent this issue and improve the demonstration, the freshly prepared crystals of Cd-SNDI were finely ground in DMF before UV-vis examination, and the resulting UV-vis absorbance looked more “normal” than the previous case, which was regarded as Fig. 3b in the new edition. Although the very strong scattering band of particles in DMF dominated the section lower than 350 nm, we have to admit that it was rather difficult to find a perfect solution, at least the more informative section above 400 nm of the improved spectra was not suffered from the up-shifted baseline, and the absorbance variations during Et₃N addition and successive photoirradiation were more identifiable than the previous edition.

Moreover, to compare the absorbance variations of ligand H₄SNDI and coordination polymer Cd-SNDI more directly, the original curves but not the spectrum subtractions were exhibited in the revised spectra to improve the readability (Figs. 3a, 3b in the new edition).

On the other hand, I believe the spectrum normalization issue should definitely be clarified. I suggest the authors review how the “normalized emission spectra” in Fig 2c were measured (although this plot is not an essential figure of this manuscript). The current representation suggests that the photoluminescence quantum yield of Cd-SNDI is roughly 75% smaller than that of H₂SNDI (i.e. PLQY of CdSNDI \sim 0.25 x PLQY of H₂SNDI). This comparison can only be made if the number of photons absorbed by the examined samples is identical (or, less ideally, similar). If this condition is not met, Fig 2c may seem misleading unless the authors explain

how the plot was made. This issue was brought up in the previous review, but I probably did not make the point clearly — if two measurements were not made on the same ground (in the current case, same absorbance), the process of normalization and comparison is not justified.

Response 4-3: We appreciated this helpful comment. The UV-vis spectra of the suspension of Cd-SNDI and the solution of H₄SNDI were measured under similar conditions (Fig. 2c).

The stock solution of H₄SNDI and suspension of Cd-SNDI were prepared similarly as depicted in the procedures of Figs. 3a and 3b (See: Experimental Details for Figures in Manuscript, Supplementary Information). After several iterative rounds of tentatively scanning the emission and excitation spectra, 405 nm was chosen as the common excitation wavelength for both the H₄SNDI solution and the Cd-SNDI suspension. By diluting the stock solution of H₄SNDI and the suspension of Cd-SNDI, their absorbances at *ca.* 405 nm were adjusted to the same levels before the comparative photoluminescence analyses. Thus, it was believed that the photoluminescence of Cd-SNDI suspension and H₄SNDI solution were reasonably compared on similar ground.

Moreover, the corresponding experimental details of Fig. 2c were added to Experimental Details for Figures in Manuscript, Supplementary Information.

In a similar vein, the comparison of the EPR intensities from different samples (e.g. those in Fig 2 and Fig S6) may not be trivial as the actual amount of samples loaded into the EPR cavity can vary. However, the intensity increment after irradiation is significant most of times, so I am not too concerned about this comparison. Finally, I want to take this chance to point out that the so-called ‘partial charge transfer’ (a scenario often considered in, for example, intramolecular charge-transfer push-pull chromophores) should not give EPR response — no matter how weak the signal intensity is. This last point is just a comment that I would like to share with the authors.

Response 4-4: Thanks to the reviewer for this enlightened comment. We agreed that merely the “partial charge transfer” or “intramolecular charge-transfer push-pull chromophores” could not give rise to the EPR signal that needed the NET charge separation.

In the previous revision (for details, see Synthesis of H₄SNDI, Supplementary Information and Response 4-11 in the previous response letter), the comparative NMR experiments disclosed that the ligand H₄SNDI tended to occur intra-ligand charge separation under the weak photoirradiation from ambient daylight.

Similarly, the charge separation tendencies of ligand H₄SNDI and coordination polymer Cd-SNDI under weak photoirradiation from ambient daylight might also be detectable by the EPR examinations.

In the comparative EPR experiments, solid samples of H₄SNDI and Cd-SNDI containing the same molar amounts of NDI moieties were employed to ensure comparability. The samples were mounted in the 0.5 mm capillaries for the sake of photo-permeability, and then sealed in EPR tubes under an inert N₂ atmosphere.

In the absence of LED irradiation (the so-called “dark” condition), the sample H₄SNDI showed a minimal EPR signal at the position of reported NDI⁻ (Fig. S6a), which might be attributed to the slight charge separation under the photoirradiation from ambient daylight. In comparison (Fig. S6b), the coordination polymer Cd-SNDI exhibited an increased and more identifiable EPR signal under photoirradiation from ambient daylight. It was believed that the inter-ligand S...S contact of Cd-SNDI shifted the intra-ligand separated charges to inter-ligand

separated ones, helping to retard the charge recombination and to maintain the EPR signal strength.

Moreover, as requested by the Reviewer, the experimental details were added to the caption of **Fig. S6**.

Reviewers' Comments:

Reviewer #2:

Remarks to the Author:

Revision is in an attached file.

Reviewer #4:

Remarks to the Author:

The provision of the measurement details in the SI for each figure is highly appreciated. I want to suggest, as a minor point, that the decay kinetics in Figures 5c and 5d should deserve a better analysis and discussion, as these data are now explicitly presented and discussed in the main text. The transient signal at 480 nm does not follow a mono-exponential decay (evidenced by the poor fit quality in Figure 5d; Figure 5b is not great, but reasonable). Instead, at least it should be described with multi-exponentials or including a long-time offset. This re-analysis will likely result in the primary decay time constant becoming short than 164 ps. Furthermore, the "persistent" signal (lasting >1.5 ns) and some similarity between the spectra at 1.5 ns and those at <20 ps may have mechanistic implications. Nonetheless, I would like to recommend the manuscript for publication.

REVIEWER COMMENTS

Reviewer #2 (Remarks to the Author):

Revision is in an attached file.

Summary:

The manuscript describe the synthesis of the a coordination polymer using naphthalene diimide (NDI) substituted with SEt and Cd(II), achieving an S···S contact, which facilitated the electron mobility while simultaneously maintaining the reduced power of the ground state and the excitedstate reducing power of its radical anionic species. This particular characteristic allows the authors to demonstrate the superior behavior of this heterogeneous photocatalyst in a consecutive photocatalysis (conPET) for the photoreduction of aryl halides including aryl chlorides with EWG. They show the benefits of the strategy comparing the performance to single-molecule dye photocatalysis and also to other coordination polymers which highlight the unique advantage of SS-bridged dye stacking. The photocatalytic system demonstrates good performance in successive radical couplings allowing the formation of C-C, C-S, C-P, and C-B bonds by known procedures.

The authors claims that enhanced the electron delocalization within the polymer improve the PET from excited-state radical anion (NDI \cdot^-)* to aryl halides. The approach circumvents the short lifetime of excited states and also could avoid competing back electron transfer in the case of aryl halides. This strategy is interesting and I believe it would reach a broad audience in which ET processes from excited states are important and the work could contain the merits for publication in this journal.

The authors have addressed all the comments I had pointed out in the previous revision. I recommend publication of the manuscript after two minor revisions.

Response 2-1: We highly appreciated the Reviewer for the positive comments about our previous revision, and thanks the Reviewer for all the efforts to help us to improve this work. The mentioned two minor issues were revised accordingly.

Minor corrections:

line 329 “up to 8.2 ns” should be deleted. This value without a comparison is misleading.

line 357 “Fig. S29” should be changed by “Fig. S29-30”

Response 2-2: Per your advice, we have deleted “up to 8.2 ns” and changed the original “Fig. S29” to updated “Supplementary Figs. 31 and 32”.

Reviewer #4 (Remarks to the Author):

The provision of the measurement details in the SI for each figure is highly appreciated.

Response 4-1: We thank the Reviewer for recognizing our previous efforts to improve our work as well as the help for us to improve this work. In the following, we responded to the comments in a one-by-one manner.

I want to suggest, as a minor point, that the decay kinetics in Figures 5c and 5d should deserve a better analysis and discussion, as these data are now explicitly presented and discussed in the main text. The transient signal at 480 nm does not follow a mono-exponential decay (evidenced by the poor fit quality in Figure 5d; Figure 5b is not great, but reasonable). Instead, at least it should be described with multi-exponentials or including a long-time offset. This re-analysis will likely result in the primary decay time constant becoming short than 164 ps.

Response 4-2: We thank the Reviewer for this kind suggestion. Indeed, Reviewer #2 mentioned the similar items in the previous round of peer-review. As we responded to Reviewer #2 in the former response letter (see **Response 2-7**), we tried to analyze **Figure 5d** as a multi-exponential decay, but it was found that the better fit was the single-exponential decay. The poor fit quality of **Figure 5d** than that of **Figure 5b** might be correlated with the uneven dispersion of Cd-SNDI particles in the suspension sample.

On the other hand, the different excited behaviors such as different decay model for $(\text{NDI}^-)^*$ has been constantly reported even for solution samples of NDI derivatives. For example, in **Ref. 45** (*J. Phys. Chem. B* **2019**, *123*, 7731–7739), a single excited-state decay with a lifetime of 142 ps was obtained for $D_1 \rightarrow D_0$ transition of $(\text{NDI}^-)^*$ under 780 nm excitation. When the excitation laser wavelength was changed to a higher energetic wavelength of 605 nm, a two-component kinetic model was required to fit the data, of which the lifetime of 103 ps was attributed to $D_1 \rightarrow D_0$ transition and another lifetime of 0.9 ps was assigned to the decay of a higher-lying excited state. Moreover, the different experimental setups also possibly affected the fitting of decay processes; it was reported that the 605 nm excitation on NDI^- can also give the decay lifetimes with the single-exponential time constants (*J. Phys. Chem. A* **2000**, *104*, 6545–6551).

Furthermore, the “persistent” signal (lasting >1.5 ns) and some similarity between the spectra at 1.5 ns and those at <20 ps may have mechanistic implications. Nonetheless, I would like to recommend the manuscript for publication.

Response 4-3: Thanks to the reviewer for this enlightened comment. We agreed that the “persistent” signals (lasting >1.5 ns) and the spectra at <20 ps may have mechanistic implications.

These information actually also triggered interests and raised up the key comments of Reviewer #2 in the previous two rounds of peer review, and we responded accordingly in very details. Briefly speaking, the spectra shown at <20 ps mostly presented the excited-state absorption (ESA) bands, and the curves might start to exhibit the characteristics of charge-separation/charge-transfer states after several dozens of ps, and the spectra reflected more clues of charge-separation/charge-transfer states after several hundreds of ps. Moreover, the comparative studies on fs-TA of ligand and coordination polymer Cd-SNDI under different excitation laser wavelengths comprehensively revealed the tendency of inter-ligand charge transfer within Cd-SNDI to provide potential merits for understanding the structure-activity relationship (SAR) and mechanism.

Therefore, we have provided the detailed phenomena of the related photochemical/physical processes and the detailed mechanistic speculations in the supplementary information file, please refer to the section of **Supplementary Experimental Methods and Analyses of Transient Absorption Spectra**.